# Cholesterol-lowering effects of oats induced by microbially produced phenolic metabolites in metabolic syndrome: a randomized controlled trial

Linda Klümpen[1], Aakash Mantri [1,2], Maren Philipps [3,4], Waldemar Seel[1], Laura Schlautmann[3], Mohamed H. Yaghmour [3], Verena Wiemann[5], Birgit Stoffel-Wagner[6], Martin Coenen [7], Leonie Weinhold[8], Jan Hasenauer [3,4], Thomas Fließwasser [5,9], Sven Burgdorf [3], Christoph Thiele [3], Peter Stehle[10] & Marie-Christine Simon [1] ✉

Oats have various positive effects on human health, but the underlying mechanisms are not fully understood. To identify oat-microbiome-host interactions contributing to metabolic improvements, we conducted two randomized controlled dietary interventions in parallel-design in individuals with metabolic syndrome, comparing a short-term, high-dose and a six-week, moderate oat intake with respective controls (DRKS00022169). Both oat diets lead to an increase in plasma ferulic acid (0.64 [0.26, 1.02], $P = 0.002$; 0.55 [0.21, 0.89], $P = 0.003$), while the high-dose oat-diet also increased dihydroferulic acid (1.23 [0.44, 2.01], $P = 0.003$). Here we show that microbial phenolic metabolites are driving factors for the cholesterol-lowering effect of oats, which might be of relevance since short-term, high-dose oat-diet is a suitable approach to alleviate obesity-related lipid disorders.

Metabolic syndrome (MetS), characterized by the co-occurrence of central obesity, dyslipidemia, elevated blood pressure (BP), and dysglycemia[1], is a significant risk factor for the development of type 2 diabetes mellitus (T2DM)[2] and cardiovascular disease (CVD)[3]. In addition, MetS is linked to an aberrant gut microbiota[4]. The prevalence of MetS has increased dramatically in recent decades and is considered a major public health challenge in both developed and developing countries. Studies indicate that up to 31% of the world's population is affected[5]. As the underlying pathophysiology of MetS is complex and cannot be attributed to a single mechanism, lifestyle changes offer a fundamental treatment option[6]. Among dietary modifications, fiber intake plays an important role in managing MetS, since the consumption of dietary fiber is associated with metabolic health[7].

Oats offer an interesting and promising approach for treating MetS due to their unique composition characterized by a high fiber content, especially β-glucan, essential minerals and vitamins, and various bioactive substances, including phenols which exert antioxidant and anti-inflammatory effects that may improve metabolic function[8]. Furthermore, oats are an accessible and sustainable food item. The positive effects of oats on glucose metabolism were first

[1]Institute of Nutrition and Food Science, Nutrition and Microbiota, University of Bonn, Bonn, Germany. [2]Institute for Genomic Statistics and Bioinformatics, University Hospital Bonn, Bonn, Germany. [3]Life and Medical Sciences Institute (LIMES), University of Bonn, Bonn, Germany. [4]Bonn Center for Mathematical Life Sciences, University of Bonn, Bonn, Germany. [5]Institute for Pharmaceutical Microbiology, University Hospital Bonn, University of Bonn, Bonn, Germany. [6]Institute of Clinical Chemistry and Clinical Pharmacology, Central Laboratory, University Hospital Bonn, Bonn, Germany. [7]Institute of Clinical Chemistry and Clinical Pharmacology, University Hospital Bonn, Bonn, Germany. [8]Institute of Medical Biometry, Informatics and Epidemiology, University Hospital Bonn, Bonn, Germany. [9]German Center for Infection Research (DZIF), Partner Site Bonn-Cologne, Bonn, Germany. [10]Institute of Nutrition and Food Science, Nutritional Physiology, University of Bonn, Bonn, Germany. ✉e-mail: mcsimon@uni-bonn.de

described by Carl von Noorden, a German diabetologist who developed a special oat cure for the treatment of diabetes at the beginning of the 20th century[9]. A few years later, de Groot et al. observed that oats could also improve lipid metabolism by reducing serum cholesterol[10]. Since then, various clinical studies have confirmed these beneficial effects of oats, and health claims have been approved for the cholesterol-lowering properties and reduction in postprandial glycemic response of oat β-glucan[11,12]. However, the underlying mechanisms of oat-induced improvements in metabolic health are not fully understood. Recent studies suggest the ability of oats to modulate the gut microbiota as a mechanism for its health-promoting effects[13]. In particular, microbially produced metabolites were ascribed a decisive role in nutrition-induced metabolic improvement[14]. However, it is still unknown whether metabolites produced by the microbial degradation of oats, such as phenolic compounds, play an important health-promoting role alongside short-chain fatty acids (SCFAs)[15] produced by the bacterial fermentation of β-glucan. We hypothesized that the beneficial effects of oats on metabolism might be influenced by host-microbiome interactions, leading to changes in phenolic compounds such as ferulic acid (FA), dihydroferulic acid (DHFA), 2-aminophenol sulfate, and 2-acetamidophenol sulfate, which have been indicated in recent studies as relevant (microbial) metabolites from whole grains such as oats with different biological functions[16,17]. The aim of this randomized controlled trial (RCT) was to investigate the effects of a short-term, high-dose oat diet (a modified version of the original oat cure by Carl von Noorden[9]) and a six-week, moderate oat diet compared with a macronutrient-adapted control diet and a Western diet, respectively, on lipid metabolism, the gut microbiota, and global metabolomic profiles, in particular the phenolic compounds FA and DHFA, in 68 individuals with MetS ($n = 17$ participants/group) and to elucidate the underlying mechanism using an integrative multi-omics analysis.

In this work, our data demonstrate the superiority of the oat diets in increasing plasma levels of phenolic compounds, such as FA, DHFA, 2-aminophenol sulfate, and 2-acetamidophenol sulfate compared with the respective controls. In addition, the short-term, high-dose oat diet improves lipid metabolism by lowering serum cholesterol levels compared with the control diet. The observed associations between the cholesterol reduction and the alterations in metabolomic profiles indicate that phenolic compounds, especially their microbial degradation products, might be driving factors for the cholesterol-lowering effect of oats, besides the known mechanism of β-glucan.

## Results

### Impact of the two different oat diets on human metabolism

To determine the effects of a short-term, high-dose oat diet under hypocaloric conditions and a six-week, moderate oat diet under isocaloric conditions on metabolism, the gut microbiota, and global metabolomic profiles, in particular FA and DHFA, in individuals with MetS compared with corresponding oat-free control diets, two randomized controlled nutritional intervention studies were conducted in parallel-design with 34 subjects each ($n = 17$ per group) (Fig. 1a, b) and completed in July 2022.

In the short-term intervention study, participants assigned to the oat group (OG) consumed three oat meals daily for two days instead of their habitual Western diet. Each oat meal comprised $100 \times g$ of rolled oat flakes (Demeterhof Schwab GmbH & Co. KG, Windsbach, Germany) boiled in water. To ascertain potential long-term effects, the two-day intervention period was followed by a six-week follow-up period during which the participants returned to their habitual diet without oats. Subjects assigned to the control group (CG) consumed three standardized control meals without oats on each intervention day, which were macronutrient-adapted to the OG, instead of their habitual Western diet.

In the six-week intervention study, participants in the oat group (OG$^{6w}$) replaced one habitual meal per day with an oatmeal comprising $80 \times g$ of rolled oat flakes (Demeterhof Schwab GmbH & Co. KG), while maintaining their habitual Western diet. Participants in the corresponding control group (CG$^{6w}$) maintained their habitual Western diet and remained abstinent from oats during the six-week study period according to the inclusion criteria.

During the short-term intervention study, two male participants withdrew from the study for personal reasons (Fig. 1c). Thus, 32 participants completed the study and were considered for the statistical analysis (OG: $n = 17$, CG: $n = 15$) which included 15 males and 17 females aged $58.9 \pm 7.4$ years (mean ± standard deviation [SD]) with a body mass index (BMI) of $32.0 \pm 3.3$ kg/m² (Table 1). In the CG, 60% of participants were men, while in the OG 35% were men. All participants had central obesity (100%) and at least two further metabolic syndrome traits, including increased BP (100%), impaired glucose metabolism (75%), and dyslipidemia (63%).

In the six-week intervention study, all 34 participants, including 15 males and 19 females aged $59.7 \pm 7.6$ years with a BMI of $31.6 \pm 3.2$ kg/m² (Table 1), completed the study and were considered in the statistical analysis (Fig. 1d). In the CG$^{6w}$, 47% of participants were men, while in the OG$^{6w}$ 41% were men. According to the inclusion criteria, all participants had central obesity (100%) and at least two further metabolic syndrome traits. Thus, at baseline, 97% of the subjects had elevated BP, 76% impaired glucose metabolism, and 59% dyslipidemia.

Further baseline characteristics of the participants and detailed information on their habitual dietary behavior are provided in Supplementary Data 1.

### Compliance and adverse events

Adherence to the study diets were recorded, showing a compliancy rate of 99% (short-term intervention study) and 89% (six-week intervention study). Tolerability assessment indicated that the diets were well tolerated overall, with no severe side effects related to the diets.

### Oat diets increased plasma (dihydro)ferulic acid levels

According to our primary hypothesis, significant higher plasma levels of FA (0.64 [0.26, 1.02] (β estimate [95% confidence interval (CI)]), $P = 0.002$; Fig. 2a) and DHFA (1.23 [0.44, 2.01], $P = 0.003$; Fig. 2b), determined by liquid chromatography with tandem mass spectrometry (LC-MS/MS)[18], were observed in OG compared to CG after the two-day intervention period, using linear regression adjusted for the baseline value of the response, age, sex, and BMI (LnR$^{adj.}$). Consistent with this, an increase in plasma FA level was observed in OG$^{6w}$ compared with CG$^{6w}$ after the six-week intervention period (0.55 [0.21, 0.89], $P = 0.003$; Fig. 2c); however, no difference in DHFA level was found (0.63 [−0.12, 1.37], $P = 0.096$; Fig. 2d). Our results therefore suggest that both the short-term, high-dose and the six-week, moderate oat diet have a systemic effect. This effect is more pronounced and potentially stronger if related to the gut microbiota as shown in the short-term intervention due to the increase in DHFA, a major microbial metabolite of FA[19].

### Oat diet reduced serum cholesterol levels

Investigating the diet-induced modulation of the metabolism using sparse partial least squares-discriminant analysis (sPLS-DA)[20,21], a significant difference between OG and CG was observed after the two-day intervention period (area under the curve (AUC) = 0.88 ($P = 1.99 \times 10^{-4}$), balanced error rate (BER) = 0.154; Fig. 3a), with low-density lipoprotein cholesterol (LDL-C), total cholesterol (TC), and diastolic BP being the most important distinguishing features (Fig. 3b). A significant reduction in LDL-C (−16.26 [−23.60, −10.32] mg/dL, $P^{adj.} = 0.011$ (Bonferroni−Holm)) and TC (−15.61 [−24.17, −7.80] mg/dL, $P^{adj.} = 0.02$) levels in OG compared with CG were also identified using LnR$^{adj.}$

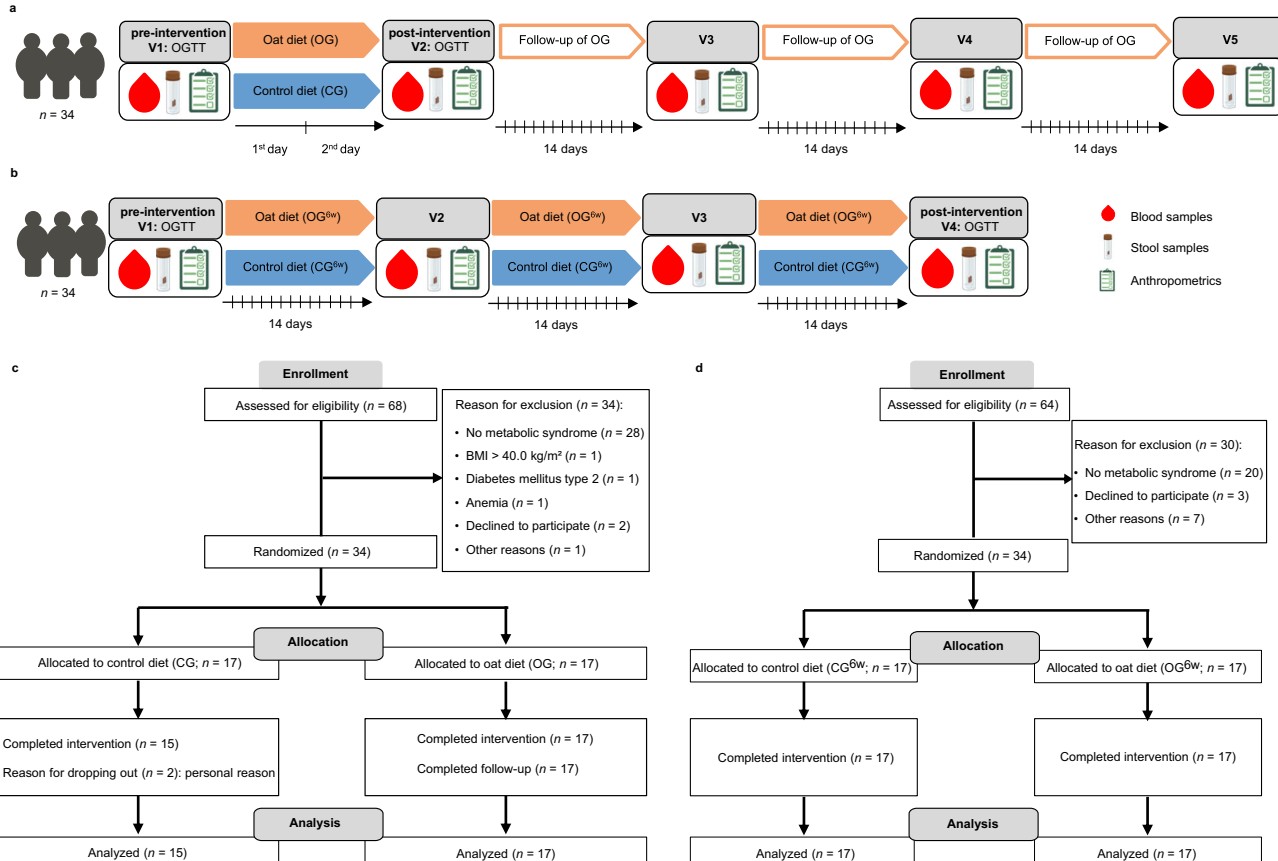

**Fig. 1 | Study scheme and participants flow diagram. a** The short-term intervention study included two clinical visits, before (V1) and after the two-day intervention period (V2), followed by three clinical visits during a six-week follow-up period within the oat group (OG; V3–V5). On V1 and V2, blood samples were taken during a 3-h oral glucose tolerance test (OGTT), fecal samples were collected, and detailed anthropometric measurements and lifestyle assessments were performed. Additionally, fecal samples were collected after the first intervention day. During the follow-up period (V3–V5), fasting blood and fecal samples were collected and detailed anthropometric measurements were performed. **b** The six-week intervention study included four clinical visits in two-week intervals with the first before (V1) and the last after the six-weeks intervention period (V4). On V1 and V4, blood samples were taken during a 3-h oral glucose tolerance test (OGTT), fecal samples were collected and detailed anthropometric measurements and lifestyle assessments were performed. At the visits during the intervention period (V2, V3), fasting blood samples, fecal samples, and anthropometric data were collected.
**c** Participants flow diagram of the short-term intervention study according CONSORT flow diagram. Out of 68 expressions of interest, 34 participants met the criteria of MetS per definition and were randomly assigned 1:1 to the study groups. A total of 32 participants completed the study with two dropouts (both males) in CG for personal reasons and were included in the final analysis. **d** Participants flow diagram of the six-week intervention study according CONSORT flow diagram. Out of 64 expressions of interest, 34 participants met the criteria of MetS per definition and were randomly assigned 1:1 to the study groups. All participants completed the study and were included in the final analysis. BMI body mass index, CG control group, CG[6w] six-week control group, OG oat group, OG[6w] six-week oat group, OGTT oral glucose tolerance test, V visit. Created in BioRender. Simon, M. (2026) https://BioRender.com/lkjxcl7[116].

(Fig. 3b, c). Since cholesterol levels tended to remain below baseline during the six-week, oat-free follow-up period, persistent effects on lipid metabolism might be assumed (Fig. 3d). This assumption is further supported by the high compliance observed during the follow-up period, as all participants abstained from oat consumption and returned to their habitual Western diet, with no significant differences compared to their pre-study dietary patterns (Supplementary Data 2). Thus, our results indicate clearly that a high-dose oat diet improves lipid metabolism by decreasing serum TC and LDL-C levels, even after two days, which is consistent with the known cholesterol-lowering effect of oats[22]. In addition, beneficial effects on anthropometrics and glucose metabolism were observed within each diet group (Supplementary Data 2), which we attribute to the diet-related calorie restriction[23].

In the six-week intervention study, sPLS-DA revealed no differences in the diet-induced modulation of metabolism between OG[6w] and CG[6w] (AUC = 0.58 ($P$ = 0.399), BER = 0.412; Fig. 3e, f). Participants' metabolic status, including TC and LDL-C levels, remained largely stable (Fig. 3g, h and Supplementary Data 2), indicating that the

consumption of a single oatmeal daily, integrated into a Western diet under isocaloric conditions, has a rather mild effect on metabolism. Thus, along the known positive effects of oats on lipid metabolism[22] observed in OG, the six-week, moderate oat diet stabilized metabolic markers.

## Oat diet enhanced microbially produced phenols in plasma

After the two-day intervention period, a strong separation between OG and CG in the non-targeted global plasma metabolomic profile, which was generated by ultra-high-performance liquid chromatography coupled with tandem high-resolution mass spectrometry (UPLC-MS/MS)[24,25], was observed using sPLS-DA (AUC = 0.99 ($P$ = 1.47 × 10$^{-6}$, BER = 0.04; Fig. 4a). Phenolic compounds, including 2-acetamidophenol sulfate, 2-aminophenol sulfate, 4-hydroxyhippurate, 4-vinylphenol sulfate, and vanilloylglycine, were among the most distinguishing metabolites (Fig. 4b) and showed a significant increase in OG compared with CG (LnR$^{adj.}$: false discovery rate (FDR) corrected $q$ < 0.05; Fig. 4b, c). Thus, our results suggest that a short-term, high-dose oat diet leads to a distinct shift in the plasma

**Table 1 | Baseline characteristics of the participants**

| | Short-term dietary intervention study | | | Six-week dietary intervention study | | |
|---|---|---|---|---|---|---|
| | Total (n = 32) | CG (n = 15) | OG (n = 17) | Total (n = 34) | CG⁶ʷ (n = 17) | OG⁶ʷ (n = 17) |
| Age (years) | 58.9 ± 7.4 | 59.5 ± 7.5 | 58.4 ± 7.5 | 59.7 ± 7.6 | 60.7 ± 7.2 | 58.7 ± 8.0 |
| Sex (% males) | 47 | 60 | 35 | 44 | 47 | 41 |
| BMI, kg/m² | 32.0 ± 3.3 | 32.8 ± 3.6 | 31.3 ± 2.9 | 31.6 ± 3.2 | 31.4 ± 3.6 | 31.9 ± 2.8 |
| WC, cm | 106.6 ± 9.4 | 111.0 ± 8.8 | 102.8 ± 8.3 | 106.4 (8.4) | 105.8 (7.5) | 108.0 (9.0) |
| SBP, mmHg | 143.3 ± 15.0 | 148.5 ± 18.3 | 138.7 ± 9.7 | 136.1 ± 12.1 | 137.2 ± 11.5 | 134.9 ± 12.8 |
| DBP, mmHg | 93.9 ± 12.1 | 93.9 ± 16.2 | 93.8 ± 7.2 | 87.3 ± 9.6 | 86.5 ± 10.0 | 88.0 ± 9.4 |
| MET score | 1413 (1242)ᵃ | 1025 (1078)ᵃ | 1605 (846)ᵃ | 1519 (1779) | 1544 (1269) | 1038 (2091) |
| MSFsc | 3.9 ± 0.9ᵇ | 3.5 ± 0.8ᵇ | 4.2 ± 0.8ᵇ | 3.8 (1.3)ᶜ | 4.1 (0.9)ᶜ | 3.4 (0.9)ᶜ |
| TG, mg/dL | 152.5 (65.0) | 153.0 (97.0) | 152.0 (49.0) | 137.0 (69.0) | 147.0 (88.5) | 135.0 (56.5) |
| HDL-C, mg/dL | 50.3 ± 9.8 | 48.9 ± 7.9 | 51.5 ± 11.3 | 49.0 (20.8) | 49.0 (23.5) | 47.0 (16.0) |
| Glucose, mg/dL | 98.7 ± 10.0 | 99.5 ± 10.4 | 98.0 ± 9.8 | 97.5 (13.5) | 96.0 (10.5) | 103.0 (32.0) |
| HOMA-IR | 3.9 (3.1) | 4.1 (2.3) | 3.3 (3.7) | 3.6 (2.3) | 3.1 (2.1) | 3.9 (2.4) |

Data are presented as means ± SD or medians (IQR) for continuous variables, and as frequencies for count data. *BMI* body mass index, *CG* control group, *CG⁶ʷ* six-week control group, *DBP* diastolic blood pressure, *HDL-C* high-density lipoprotein cholesterol, *HOMA-IR* homeostasis model assessment-estimated insulin resistance, *MET* metabolic equivalent, *MSFsc* corrected mid sleep on free days (chronotype), *OG* oat group, *OG⁶ʷ* six-week oat group, *SBP* systolic blood pressure, *TG* triglycerides, *WC* waist circumference. Source data are provided as a Source Data file.
ᵃ*n* = 31 (CG: *n* = 14, OG: *n* = 17).
ᵇ*n* = 25 (CG: *n* = 10, OG: *n* = 15).
ᶜ*n* = 30 (CG⁶ʷ: *n* = 15, OG⁶ʷ: *n* = 15).

metabolomic profile, particularly characterized by an increase in microbially produced phenolic compounds.

A clear difference in the plasma metabolomic profile was also observed between OG⁶ʷ and CG⁶ʷ after the six-week intervention period (AUC = 0.927 ($P = 2.1 \times 10^{-5}$), BER = 0.176; Fig. 4d). The phenolic compounds 2-acetamidophenol sulfate and 2-aminophenol sulfate were among the most important distinguishing metabolites and showed a significant increase in OG⁶ʷ compared with CG⁶ʷ (LnRᵃᵈʲ·: $q < 0.05$; Fig. 4e, f). Notably, an increase in these phenols was also detected in the OG, supporting our assumption that this increase is related to oats. Our results thus indicate that a six-week, moderate oat diet leads to shifts in the global plasma metabolomic profile that are consistent with those of a short-term, high-dose oat diet, although the effects are less pronounced.

**Oat diets modulated fecal amino acid and lipid metabolites**
When investigating the diet-induced modulation of the non-targeted global metabolomic profiles in feces[24,25], sPLS-DA revealed a strong separation between OG and CG after the two-day intervention period (AUC = 0.93 ($P = 1.16 \times 10^{-4}$), BER = 0.177; Fig. 5a). Amino acids and their (microbial) degradation products, including 8-methoxykynurenate, indolepropionate, 3-hydroxy-phenylacetate, and N-acetyltryptophan, as well as pyridoxate, a metabolite of the vitamin B6 metabolism, were among the most influential compounds for differentiating the diet groups (Fig. 5b). 8-methoxykynurenate and pyridoxate increased in OG compared to CG, while indolepropionate and 3-hydroxyphenylacetate decreased (LnRᵃᵈʲ·: $q < 0.05$; Fig. 5b, c). Shifts in the fecal metabolomic profile induced by the short-term, high-dose oat diet thus appear to be mainly linked to amino acid-related pathways, which have a significant relevance for cholesterol metabolism[26].

After the six-week intervention period, different shifts in the fecal metabolomic profile were also revealed between OG⁶ʷ and CG⁶ʷ (AUC = 0.75 ($P = 0.014$), BER = 0.353; Fig. 5d). The most influential metabolites contributing to group differentiation were lipid metabolites; however, these metabolites did not differ significantly between the diet groups according to LnRᵃᵈʲ· ($q > 0.05$; Fig. 5e). Thus, our results suggest that a six-week, moderate oat diet has less pronounced effects on the global fecal metabolomic profile compared with a short-term, high-dose oat diet and might lead to a shift in the lipid metabolite pattern rather than to changes in individual lipids.

**Oat diets modulated gut microbiota composition**
Investigating dietary modulation of the gut microbiota composition based on 16S rRNA gene sequencing using sPLS-DA, a significant difference was observed between OG and CG after the two-day intervention period (AUC = 0.83 ($P = 2.96 \times 10^{-3}$), BER = 0.219; Fig. 6a). Erysipelotrichaceae UCG-003 was the most important genus for differentiating the diet groups (weight loading component 1: −1.0) and showed a significant increase in OG compared with CG according to LinDA[27] (log₂ fold change = 1.01, $q = 0.025$; Fig. 6b). This indicates that the increase is related to oats, which is further supported by the visible reduction during the oat-free follow-up (Fig. 6c).

After the six-week intervention period, differences in the modulation of the microbial composition were also observed between OG⁶ʷ and CG⁶ʷ (AUC = 0.79 ($P = 0.004$), BER = 0.265; Fig. 6d). *Ruminococcus torques* group, a mucolytic bacterium that has been linked to negative health outcomes[28], was most relevant for differentiating the diet groups (weight loading component 1: −1.0) and tended to decrease in OG⁶ʷ compared with CG⁶ʷ (log₂ fold change = −1.58, $q = 0.07$; Fig. 6e, f).

Our results suggest that both a short-term, high-dose and a six-week, moderate oat diet led to specific microbial shifts while maintaining the core microbiome composition with a stable overall ecological structure and diversity (Supplementary Fig. 1).

**Modulated microbial functional capacity**
Whether microbial functional capacity changed in response to the dietary interventions was investigated based on the Kyoto Encyclopedia of Genes and Genomes (KEGG) database[29] by quantifying pathway abundance using the Phylogenetic Investigation of Communities by Reconstruction of Unobserved States 2 (PICRUSt2)[30]. According to sPLS-DA, OG and CG modulated the microbial functions differently but not statistically significantly (AUC = 0.69 ($P = 0.09$), BER = 0.396; Fig. 6g); even though, several pathways seemed to be of relevance for differentiation, including carbohydrate digestion and absorption, aminobenzoate degradation, selenocompound metabolism, naphthalene degradation, and phosphotransferase system (PTS) (Fig. 6h). An increase in aminobenzoate degradation and PTS as well as a decrease in naphthalene degradation and carbohydrate digestion and absorption was identified using LMMᵃᵈʲ· ($P < 0.05$; Fig. 6h), suggesting that

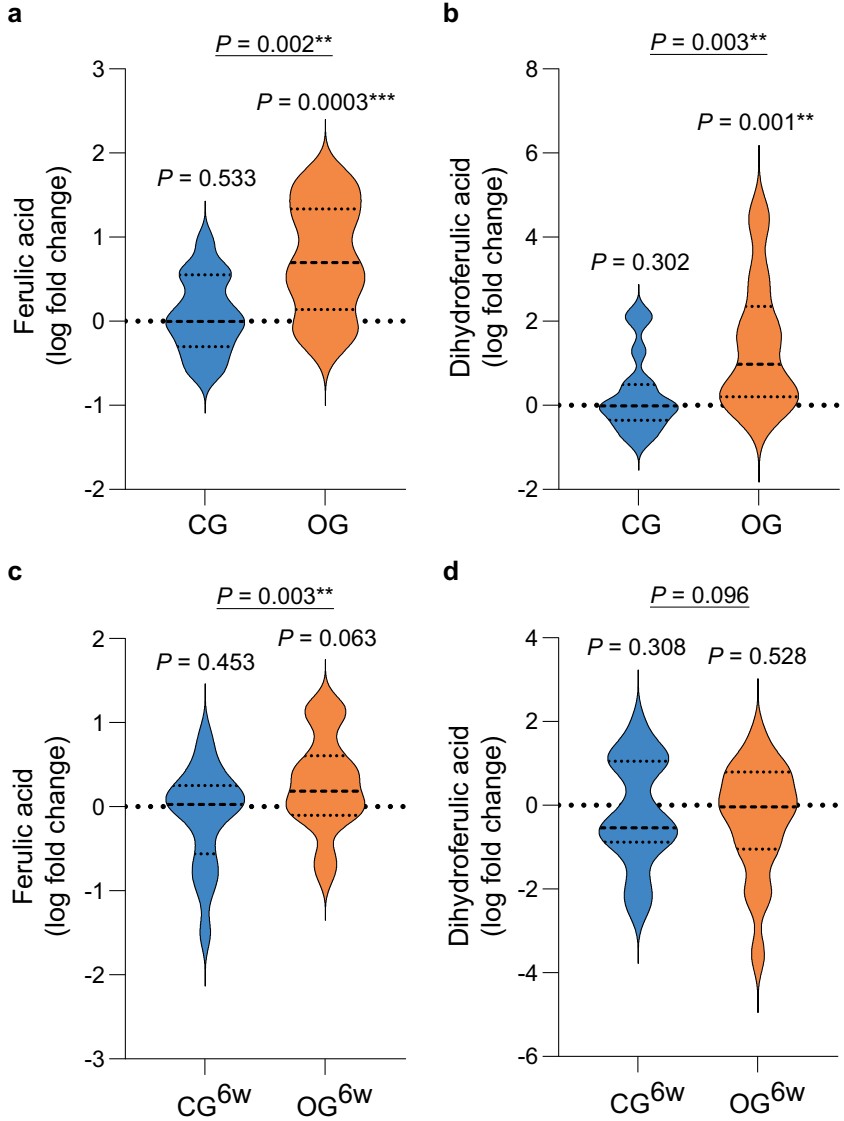

**Fig. 2 | Oat diets increased plasma (dihydro)ferulic acid levels.** Diet-induced changes in (**a**) plasma ferulic acid and (**b**) plasma dihydroferulic acid levels in OG ($n = 17$) and CG ($n = 15$). Diet-induced changes in (**c**) plasma ferulic acid and (**d**) plasma dihydroferulic acid levels in OG[6w] ($n = 17$) and CG[6w] ($n = 17$). **a**–**d** Violin plots show the log fold change (dashed center line: median; dotted line: upper and lower quartiles). Within-group differences were analyzed using two-sided paired Student's *t*-test. Details on the test statistics are provided in Supplementary Data 2. Between-group differences were analyzed using linear regression models adjusted for baseline concentration, age, sex, BMI (LnR[adj.]). CG control group, CG[6w] six-week control group, OG oat group, OG[6w] six-week oat group. Source data are provided as a Source Data file.

benzoate and related pathways might be of relevance and potentially associated with the observed increase in the phenolic metabolites in plasma[31].

Furthermore, a different modulation of the microbial functional capacity between the diet groups of the six-week intervention study was revealed, although not statistically significant (AUC = 0.68 ($P = 0.07$), BER = 0.353; Fig. 6i), with phenylalanine metabolism, proximal tubule bicarbonate reclamation, beta-lactam resistance, fructose and mannose metabolism, and PTS seemed to be the most relevant pathways for differentiation between OG[6w] and CG[6w] (Fig. 6j). An increase in OG compared to CG was shown in phenylalanine metabolism and proximal tubule bicarbonate reclamation, while a decrease was observed in the other pathways using LMM[adj.] ($P < 0.05$; Fig. 6j). Thus, our results suggest that a the six-week, moderate oat diet might modulate a variety of different microbial functions, which is in line with the heterogenous metabolic response.

**Cholesterol reduction by oats linked to increase in phenols**

Data Integration Analysis for Biomarker discovery using Latent cOmponents (DIABLO[32]) revealed strong correlations between the changes in metabolism, plasma and fecal metabolites, as well as microbial composition (model 1.1: AUC = 0.94 ($P = 5.03 \times 10^{-3}$), BER = 0.07; Fig. 7a) and functional capacity (model 2.1: AUC = 0.92 ($P = 5.11 \times 10^{-3}$), BER = 0.04; Fig. 7b) induced by the short-term, high-dose oat diet. Notably, the reduction in serum cholesterol levels was associated with an increase in several plasma phenolic compounds including FA, DHFA, 2-acetamidophenol sulfate, 2-aminophenol sulfate, 4-hydroxyhippurate, vanilloylglycine, 4-vinylphenol sulfate, 2-methoxy-hydroquinone sulfate (1), and 3-methoxycatechol sulfate (2). These inverse associations were additionally largely confirmed by pairwise correlation analyses ($P < 0.05$; Fig. 7c). In addition, PLS regression models revealed that the change in plasma phenolic compounds solely predicted 13.6% of the variation in cholesterol levels

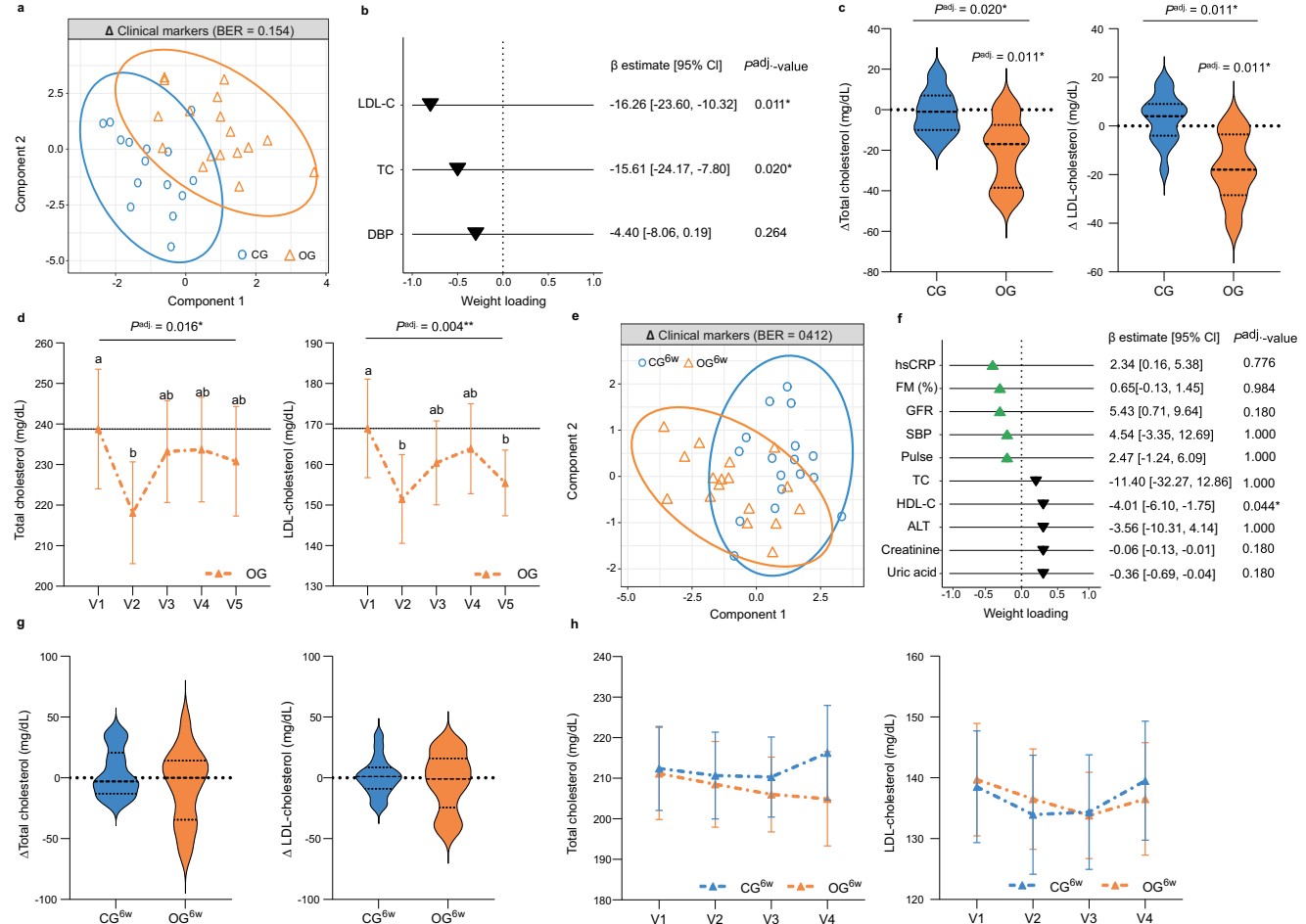

**Fig. 3 | Oat diet reduced serum cholesterol levels. a** Two-component sample plot of the sPLS-DA on metabolic changes in OG and CG. **b** Weight loadings of selected clinical markers in component 1 of the sPLS-DA (green or black triangle: increase or decrease in OG vs. CG). Coefficient and 95% CI were calculated using linear regression models adjusted for baseline concentration, age, sex, BMI (LnR$^{adj.}$). **c** Violin plots showing absolute change in total cholesterol and LDL-cholesterol (dashed center line: median; dotted line: upper and lower quartiles). **d** Longitudinal course of total cholesterol and LDL-cholesterol levels during the six-week follow-up period. Data are presented as means ± SEM. Differences over time were analyzed using LMM adjusted for age, sex, BMI (LMM$^{adj.}$). Different letters indicate significant difference between time points. **e** Two-component sample plot of the sPLS-DA on metabolic changes in OG$^{6w}$ and CG$^{6w}$. **f** Weight loadings of the top 10 selected clinical markers in component 1 of the sPLS-DA (green or black triangle: increase or decrease in OG$^{6w}$ vs. CG$^{6w}$). Coefficient and 95% CI were calculated using linear regression models adjusted for baseline concentration, age, sex, BMI (LnR$^{adj.}$). **g** Violin plots showing absolute change in total cholesterol and LDL-cholesterol

(dashed center line: median; dotted line: upper and lower quartiles). **h** Longitudinal course of total cholesterol and LDL-cholesterol levels during the six-week intervention period. Data are presented as means ± SEM. Between-groups differences were analyzed using LMM adjusted for age, sex, BMI (LMM$^{adj.}$) with the interaction term visit*group. All *P*-values were adjusted for multiple testing using Bonferroni–Holm method. **c** + **g** Within-group differences were analyzed using two-sided paired Student's *t*-test. Between-group differences were analyzed using linear regression models adjusted for baseline concentration, age, sex, BMI (LnR$^{adj.}$). **a**–**e** OG: $n = 17$, CG: $n = 15$. **e**–**h** OG$^{6w}$: $n = 17$, CG$^{6w}$: $n = 17$. ALT alanine aminotransferase, BER balanced error rate, CG control group, CG$^{6w}$ six-week control group, CI confidence interval, DBP diastolic blood pressure, FM fat mass, GFR glomerular filtration rate, HDL high-density lipoprotein cholesterol, hsCRP high-sensitivity C-reactive protein, LDL-C low-density lipoprotein cholesterol, OG oat group, OG$^{6w}$ six-week oat group, SBP systolic blood pressure, SEM standard error of the mean, TC total cholesterol. Source data are provided as a Source Data file.

($Q^2 = 0.136$) and explained 13.5% of the variance in TC ($R^2 = 0.135$) and 19.3% of the variance in LDL-C ($R^2 = 0.193$). Thus, our results suggest that phenolic compounds play an important role in the cholesterol-lowering effect of oats.

Moreover, associations of microbial phenol degradation products including DHFA and 4-hydroxyhippurate with genera such as Erysipelotrichaceae UCG-003 and microbial pathways such as aminobenzoate and naphthalene degradation were found (see below), indicating that the increase in plasma phenols is associated with oat-induced changes in the gut microbiota. Shifts in Erysipelotrichaceae UCG-003, *Marvinbryantia* and naphthalene degradation were associated with changes in circulating cholesterol levels (see below), with the inverse association between Erysipelotrichaceae UCG-003 and cholesterol levels also identified by pairwise correlation analysis

($P < 0.05$; Fig. 7d). In addition, these changes in the gut microbiome solely predicted 13.5% of the variation in cholesterol levels ($Q^2 = 0.135$) and explained 11.1% of the variance in TC ($R^2 = 0.111$) and 19.6% of the variance in LDL-C ($R^2 = 0.196$). This supports our assumption that alterations in microbial composition and functional capacity may contribute to lowering cholesterol levels.

Besides the phenolic compounds, various plasma amino acid metabolites such as 3-methylhistidine, N-acetyl-3-methylhistidine, and 1-methyl-5-imidazolelactate were positively associated with the oat-induced change in circulating cholesterol levels (see below, Supplementary Data 3), which was also shown by pairwise correlations ($P < 0.05$; see below). These metabolites were additionally linked to shifts in Erysipelotrichaceae UCG-003 and microbial naphthalene degradation (see below, Supplementary Data 3), indicating that a

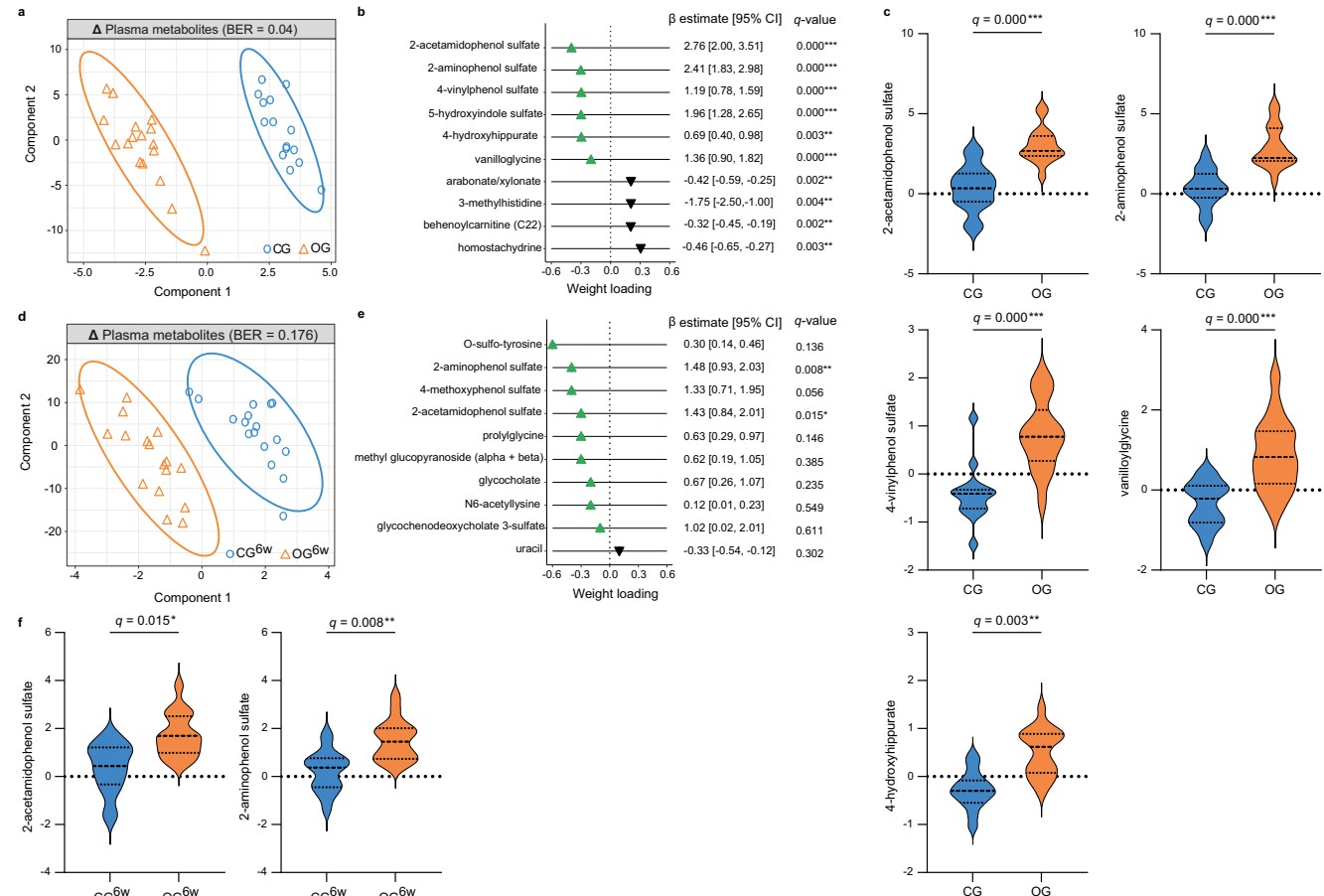

**Fig. 4 | Oat diet enhanced microbially produced phenols in plasma. a** Two-component sample plot of the sPLS-DA on the changes in the global plasma metabolomic profiles in OG and CG. **b** Weight loadings of the top 10 selected plasma metabolites in component 1 of the sPLS-DA (green or black triangle: increase or decrease in OG vs. CG). Coefficient and 95% CI were calculated using linear regression models adjusted for baseline concentration, age, sex, BMI (LnR$^{adj.}$). **c** Violin plots showing log fold change of selected phenolic metabolites (dashed center line: median; dotted line: upper and lower quartiles). Between-group differences were analyzed using linear regression models adjusted for baseline concentration, age, sex, BMI (LnR$^{adj.}$). **d** Two-component sample plot of the sPLS-DA on the changes in the global plasma metabolomic profiles in OG$^{6w}$ and CG$^{6w}$. **e** Weight loadings of the top 10 selected plasma metabolites in component 1 of the sPLS-DA (green or black triangle: increase or decrease in OG$^{6w}$ vs. CG$^{6w}$). Coefficient and 95% CI were calculated using linear regression models adjusted for baseline concentration, age, sex, BMI (LnR$^{adj.}$). **f** Violin plots showing log fold change of selected phenolic metabolites (dashed center line: median; dotted line: upper and lower quartiles). Between-group differences were analyzed using linear regression models adjusted for baseline concentration, age, sex, BMI (LnR$^{adj.}$). All $P$-values were FDR-corrected based on Benjamini/Hochberg method ($q$-value). **a**–**c** OG: $n = 17$, CG: $n = 15$. **d**–**f** OG$^{6w}$: $n = 17$, CG$^{6w}$: $n = 17$. BER balanced error rate, CG control group, CG$^{6w}$ six-week control group, CI confidence interval, OG oat group, OG$^{6w}$ six-week oat group. Source data are provided as a Source Data file.

decrease in histidine metabolites due to oat-induced shifts in microbial composition and functional capacity may contribute to lowering cholesterol levels; however, the observed associations were not as strong as for the phenolic compounds.

Moreover, the changes in cholesterol levels were positively associated with fatty acids, including plasma 2S,3R-dihydroxybutyrate, fecal branched-chain 14:0 dicarboxylic acid, and fecal isocaproate (i6:0), and negatively associated particularly with fecal endocannabinoids, which have been linked to anti-inflammatory effects[33] (see below). These results were largely confirmed by additional pairwise correlation analyses ($P < 0.05$; Fig. 7c, e). Notable, shifts in Erysipelotrichaceae UCG-003 and microbial naphthalene degradation were associated with many of the above-mentioned metabolites (see below; Supplementary Data 3). This suggests that changes in the lipidome contribute to the reduction in circulating cholesterol levels potentially via an oat-induced modulation of the gut microbiota; however, the correlations were also less pronounced compared to those with the phenolic compounds. Detailed information on the pairwise correlations is provided in Supplementary Data 4.

In the six-week intervention study, DIABLO revealed several correlations between changes in metabolism, plasma and fecal metabolites, as well as the microbial composition (model 1.2: AUC = 0.85 ($P = 4.42 \times 10^{-3}$), BER = 0.32; Supplementary Fig. 2a) and functional capacity (model 2.2: AUC = 0.81 ($P = 0.01$), BER = 0.32; Supplementary Fig. 2b). However, most of the selected features did not differ between OG$^{6w}$ and CG$^{6w}$, indicating that the identified correlations might not be related to the moderate oat diet and should be interpreted with caution. Information on all correlations identified is provided in the supplement (Supplementary Data 3).

## DHFA altered cholesterol metabolism in vitro

In peripheral blood mononuclear cells (PBMCs) fed with alkyne cholesterol, DHFA led to a significant decrease in the molar fraction of alkyne cholesterol ($-15.71 \pm 6.33$ mol% (mean ± SEM), $P = 0.042$), while the total lipidome remained stable (Fig. 8a). In addition, in metabolically healthy controls (MHCs), the molar fraction of alkyne cholesterol esters tended to be lower in the PBMCs treated with DHFA ($-5.54 \pm 1.94$ mol%, $P = 0.065$; Fig. 8a). These results support that

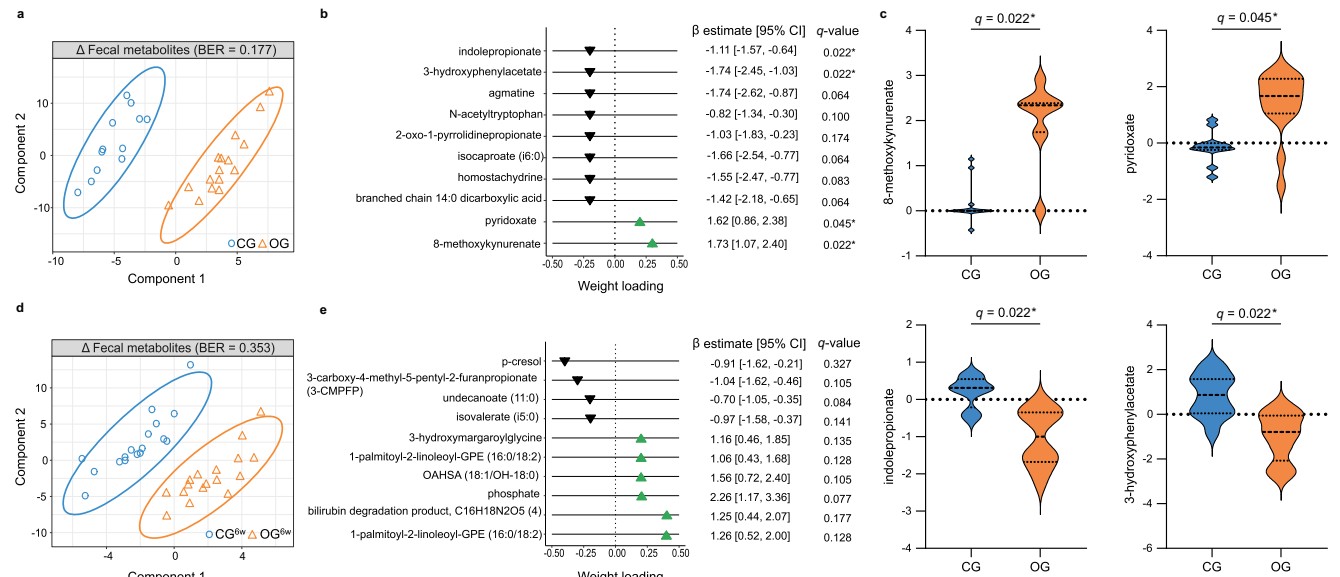

**Fig. 5 | Oat diets modulated fecal amino acid and lipid metabolites. a** Two-component sample plot of the sPLS-DA on the changes in the global fecal metabolomic profiles in OG and CG. **b** Weight loadings of the top 10 selected fecal metabolites in component 1 (green or black triangle: increase or decrease in OG vs. CG). Coefficient and 95% CI were calculated using linear regression models adjusted for baseline concentration, age, sex, BMI (LnR^adj.). **c** Violin plots showing log fold change of selected metabolites (dashed center line: median; dotted line: upper and lower quartiles). Between-group differences were analyzed using linear regression models adjusted for baseline concentration, age, sex, BMI (LnR^adj.). **d** Two-component sample plot of the sPLS-DA on the changes in the global fecal metabolomic profiles in OG^6w and CG^6w. **e** Weight loadings of the top 10 selected fecal metabolites in component 1 (green or black triangle: increase or decrease in OG^6w vs. CG^6w). Coefficient and 95% CI were calculated using linear regression models adjusted for baseline concentration, age, sex, BMI (LnR^adj.). All P-values were FDR-corrected based on Benjamini/Hochberg method (q-value). **a–c** OG: n = 16, CG: n = 12. **d–f** OG^6w: n = 17, CG^6w: n = 17. BER balanced error rate, CG control group, CG^6w six-week control group, CI confidence interval, OG oat group, OG^6w six-week oat group. Source data are provided as a Source Data file.

microbially produced phenolic compounds such as DHFA mediate the oats' cholesterol-lowering effects by reducing alkyne cholesterol and cholesterol esters incorporation into PBMCs relative to the total lipidome.

In HuH7 cells fed with $^{13}$C labeled acetate and alkyne fatty acid 182, DHFA seemed to decrease the molar fraction of $^{13}$C labeled-alkyne cholesterol esters and alkyne cholesterol esters of the total lipidome, while the total lipidome itself increased by trend (Fig. 8b). Consistent with this, a decrease in the molar fraction of the unlabeled cholesterol esters tended to take place (Supplementary Fig. 3a). No differences in total lipidome, the molar fraction of alkyne cholesterol esters and alkyne cholesterol were observed between HuH7 cells either treated with or without DHFA and additionally fed with alkyne cholesterol (Supplementary Fig. 3b). Our results indicate that DHFA might impact different effects i) the totality of lipids, ii) the de novo synthesis of cholesterol esters and iii) the esterification of cholesterol in HuH7 cells, supporting that phenolic compounds such as DHFA might contribute to the cholesterol-lowering properties of oats.

### Microbial oat metabolization produced phenolics in vitro

To prove whether the identified phenolic compounds stem from the microbial metabolization of oats, anaerobic fecal batch culture fermentations of in vitro digested oat flakes were conducted. This demonstrated that the gut microbes are capable of producing a variety of metabolites, including phenolic compounds, by directly interacting with the provided oats. In particular, a significant increase in 2-aminophenol was shown in both oat groups (physiological oat group (POG) vs. physiological control group (PCG): P < 0.001 (6 h, 36 h), P < 0.01 (24 h), starving oat group (SOG) vs. starving control group (SCG): P < 0.01 (24 h, 36 h); Fig. 8c), which is in line with the observed increase in the plasma level of 2-aminophenol sulfate in our RCTs. In addition, a rapid production of microbial products such as DHFA and vanillic acid were observed in both oat groups by a simultaneous

reduction of oat phenolic compounds such as FA and vanillin mainly within 6 h (POG vs. PCG: P < 0.01 (DHFA, FA, vanillin), P < 0.05 (vanillic acid); SOG vs. SCG: P < 0.05 (DHFA, FA, vanillin); Supplementary Fig. 3c), emphasizing that the microbial activity results in the production of phenolic oat-derived metabolites.

Moreover, as expected, shifts in the concentrations of various SCFAs, in particular an increase in acetic acid (POG vs. PCG: all P < 0.05) and succinate (SOG vs. SCG: P = 0.033 (36 h)) and a decrease in iso-butyric acid (POG vs. PCG: P = 0.002 (24 h), SOG vs. SCG: P = 0.001 (6 h), P = 0.047 (36 h); Fig. 8c), were found. Furthermore, significantly higher abundances of known SCFA-producing bacteria such as *Faecalibacterium*, *Fusicatenibacter*, *Bifidobacterium*, *Blautia*, and *Roseburia* were observed in POG compared with PCG, especially after 36 h (all P < 0.05; Supplementary Fig. 3d). In addition, *Erysipelotrichaceae* UCG-003 seemed to increase in POG compared to PCG after 36 h of incubation (P = 0.172; Fig. 8d), supporting that the identified increase in the abundance of *Erysipelotrichaceae* UCG-003 in our RCT is induced by the short-term, high-dose oat intake.

## Discussion

Our results suggest that the phenolic compounds in oats, and particularly their microbial degradation products, are driving factors for the oat-induced decrease in serum cholesterol levels. Moreover, changes in a number of metabolites of lipid and amino acid metabolism might contribute to the cholesterol-lowering effect via oat-induced shifts in the gut microbiome composition and function. As observed in the short-term intervention study, microbially produced metabolites, especially phenolic compounds such as DHFA, may play a more relevant role in oat-induced metabolic improvements than bioactive compounds originating from oats. In the moderate six-week intervention study, however, individual differences in lipid metabolism and microbial composition and functional capacity might be more important. (Fig. 9) The observed differences in the health effects of the

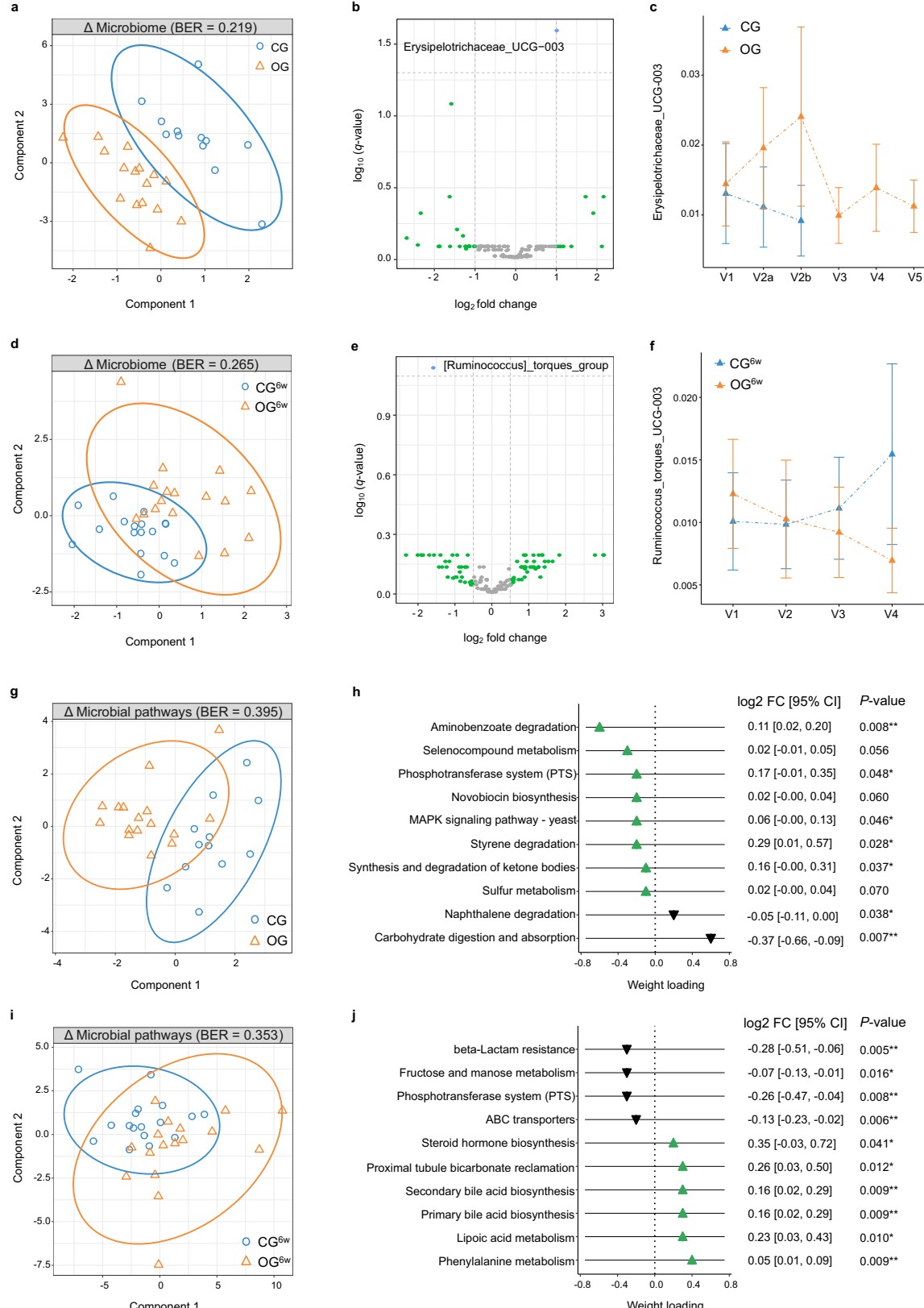

two oat interventions may be explained by different underlying mechanisms such as calorie restriction, dose-dependent exposure to the bioactive compounds and potential synergistic effects.

As the gut microbiota is able to metabolize dietary phenolic compounds into more biologically active metabolites, it is thought to play crucial roles in the digestion and absorption of these compounds[34]. Indeed, in our in vitro batch culture experiment, we demonstrated a rapid microbial production of phenolic compounds from oat fermentation. Accordingly, we observed a significant increase in plasma FA levels after both oat diets and an increase in plasma DHFA concentrations, one of the main microbial derivatives of FA, after the short-term, high-dose oat diet compared with the respective controls.

**Fig. 6 | Oat diets modulated gut microbiota composition and functional capacity. a** Two-component sample plot of the sPLS-DA on the shifts in microbial composition in OG and CG. **b** Volcano plot showing the changes in microbial composition on genus level identified using LinDA, presented as log2 fold change of each genus against its statistical significance, reported as negative log10-transformed $q$-value (FDR-corrected $P$-value), for the interaction term visit*group. **c** Relative abundance of Erysipelotrichaceae UCG-003 (mean ± 95% CI) before (V1), during (V2a) and after the two-day intervention (V2b) as well as during the six-week follow-up period (V3–V5) in OG and CG. **d** Two-component sample plot of the sPLS-DA on the shifts in microbial composition in OG⁶ʷ and CG⁶ʷ. **e** Volcano plot showing the changes in microbial composition on genus level identified using LinDA, presented as log2 fold change of each genus against its statistical significance, reported as negative log10-transformed $q$-value (FDR-corrected $P$-value), for the interaction term visit*group. **f** Relative abundance of *Ruminococcus torques* group

(mean ± 95% CI) before (V1), during (V2, V3) and after the six-week intervention period (V4) in OG⁶ʷ and CG⁶ʷ. **g** Two-component sample plot of the sPLS-DA on the changes in microbial functional capacity in OG and CG. **h** Weight loadings of the top 10 selected microbial pathways in component 1 (green or black triangle: increase or decrease in OG vs. CG). Log2 fold change and 95% CI were calculated using linear mixed models adjusted for baseline concentration, age, sex, BMI (LMMᵃᵈʲ·). **i** Two-component sample plot of the sPLS-DA on the changes in microbial functional capacity in OG⁶ʷ and CG⁶ʷ. **j** Weight loadings of the top 10 selected microbial pathways in component 1 (green or black triangle: increase or decrease in OG⁶ʷ vs. CG⁶ʷ). Log2 fold change and 95% CI were calculated using linear mixed models adjusted for baseline concentration, age, sex, BMI (LMMᵃᵈʲ·). **a**–**c**, **g** + **h** OG: $n = 16$, CG: $n = 12$. **d**–**f**, **i** + **j** OG⁶ʷ: $n = 17$, CG⁶ʷ: $n = 17$. BER balanced error rate, CG control group, CG⁶ʷ six-week control group, CI confidence interval, OG oat group, OG⁶ʷ six-week oat group. Source data are provided as a Source Data file.

These increases were associated with decreased TC and LDL-C levels, demonstrating that phenolic metabolites derived from oats, such as FA, can decrease cholesterol levels[35]. The proposed mechanism underlying the cholesterol-lowering properties of FA is the inhibition of 3-hydroxy-3-methyl-glutaryl-coenzyme A reductase (HMGCR), which controls cholesterol synthesis and modulates lipogenic gene expression in the liver[36]. However, the potential of FA to improve lipid metabolism has to date mainly been investigated in animal studies[36,37], while the potential of DHFA and other (microbially produced) phenolic metabolites to improve lipid metabolism has not yet been investigated. Therefore, our finding that DHFA also mediates the cholesterol-lowering effect by modulating cholesterol metabolism in PBMCs and tends to affect total lipids, de novo synthesis of cholesterol esters, and esterification of cholesterol in HuH7 cells is quite remarkable and offers new therapeutic approaches.

Moreover, we found elevated levels of several other phenolic compounds related to oats and metabolites associated with the microbial degradation of phenolic compounds, including 4-hydroxyhippurate[38], 2-hydroxyhippurate[39], 2-acetamidophenol sulfate[17], 2-aminophenol sulfate[17], 3-methoxycatechol sulfate[40], vanilloylglycine[41], 4-vinylphenol sulfate[42], and 2-methoxyhydroquinone sulfate[43]. To the best of our knowledge, such a connection between these metabolites and cholesterol reduction, as observed in the short-term intervention study, has not yet been demonstrated. Thus, our results indicate that a short-term, high-dose oat diet and, to a lesser extent, the six-week, moderate oat diet significantly increase the plasma concentration of phenolic compounds[44] and their microbial degradation products[39], thereby confirming the bioavailability of these metabolites and contributing to cholesterol reduction.

The attenuated effect of the six-week oat intervention on plasma phenolic metabolite levels may be attributed to the lower oat consumption ($80 \times g$/day) when integrating a single oat meal into the habitual diet, compared to the short-term high-dose oat diet ($300 \times g$/day) where three oat meals replaced the habitual diet entirely. The single meal leads to a relatively lower intake of oat bioactive compounds, including phenols, which may impact their biological effects, such as their hypolipidemic activity[36]. In addition, maintaining the habitual Western dietary pattern alongside a single oat meal introduced greater variability in foods and nutrient exposure, potentially covering intervention efficacy. We therefore assume that the impact of a single oat meal may not be strong enough to compensate for the inter-individual differences in habitual food intake during the study period. Heterogeneity of oat meal preparation[45] as recommended in the OG⁶ʷ in combination with inter-individual differences in the response to oats based on the gut microbiome and the host genetic phenotype may also contribute to the moderate effect in the six-week intervention[46]. These observations suggest that personalized nutrition strategies may optimize oat interventions at moderate doses, where individual parameters seem to exert greater influence compared to a

high-dose protocol. Despite the high individual variability, we could explain almost 20% of the variance in LDL-cholesterol in the short-term study solely by the changes in the plasma phenolic compounds. Furthermore, it is worth noting that the participants in the short-term intervention study additionally underwent calorie restriction, while the dietary intervention in the six-week study was isocaloric. The observed differences in the health effects of the two oat interventions may therefore be explained by different underlying mechanisms such as calorie restriction[47], dose-dependent exposure to the bioactive compounds, and potential synergistic effects.

Notably, in the short-term, high-dose oat diet, the increase in phenolic compounds was associated with a specific oat-induced modulation of the gut microbiota, characterized by an increase in Erysipelotrichaceae UCG-003, which is associated with a healthy aging process[48]. Because of the strong positive correlation between this genus and various phenolic compounds and its inverse correlation with cholesterol levels, we propose that Erysipelotrichaceae UCG-003 may metabolize the phenolic compounds in oats and thus contribute to lowering cholesterol levels. This oat-induced increase in Erysipelotrichaceae UCG-003 and its inverse correlation with cholesterol levels in humans, however, should be confirmed in further clinical studies. Presently this bacterium is an uncultivated genus whose physiological properties have not been described. However, the KEGG metabolic pathways for the family Erysipelotrichaceae include pathways for the degradation of aromatic compounds, specifically benzoate degradation[49]. We observed an oat-induced increase in aminobenzoate degradation, which is in line with a previous study[13] showing the upregulation of pathways associated with the bacterial degradation of aromatic compounds[50]. This supports our finding of a positive correlation between aminobenzoate degradation and DHFA levels. Aminobenzoate degradation is linked to cardio-metabolic health owing to its inverse correlation with body weight[51]. Therefore, an oat-induced increase in this specific pathway may be considered beneficial, particularly in individuals with MetS, as investigated.

Previous studies have suggested that dietary L-histidine can induce hypercholesterolemia[52]. Therefore, the changes in histidine metabolism observed in the short-term, high-dose oat diet are notable, as a reduction in dietary histidine intake resulted in lower levels of 3-methylhistidine in plasma[53]. These alterations may have contributed to the oat-induced cholesterol reduction. In addition, metabolites produced by microbes through the degradation of histidine, specifically imidazole propionate, have negative effects on human health by impairing glucose tolerance and insulin signaling[54,55]. A reduction in potentially harmful microbially produced metabolites of histidine, such as 1-methyl-5-imidazolelactate and 1-methyl-5-imidazoleacetate, may contribute to the cholesterol-lowering effect, as observed in our short-term, high-dose oat diet.

To date, the properties of oats to improve lipid metabolism have mainly been attributed to β-glucan, which can reduce the intestinal absorption of dietary cholesterol by forming a viscous gel in aqueous

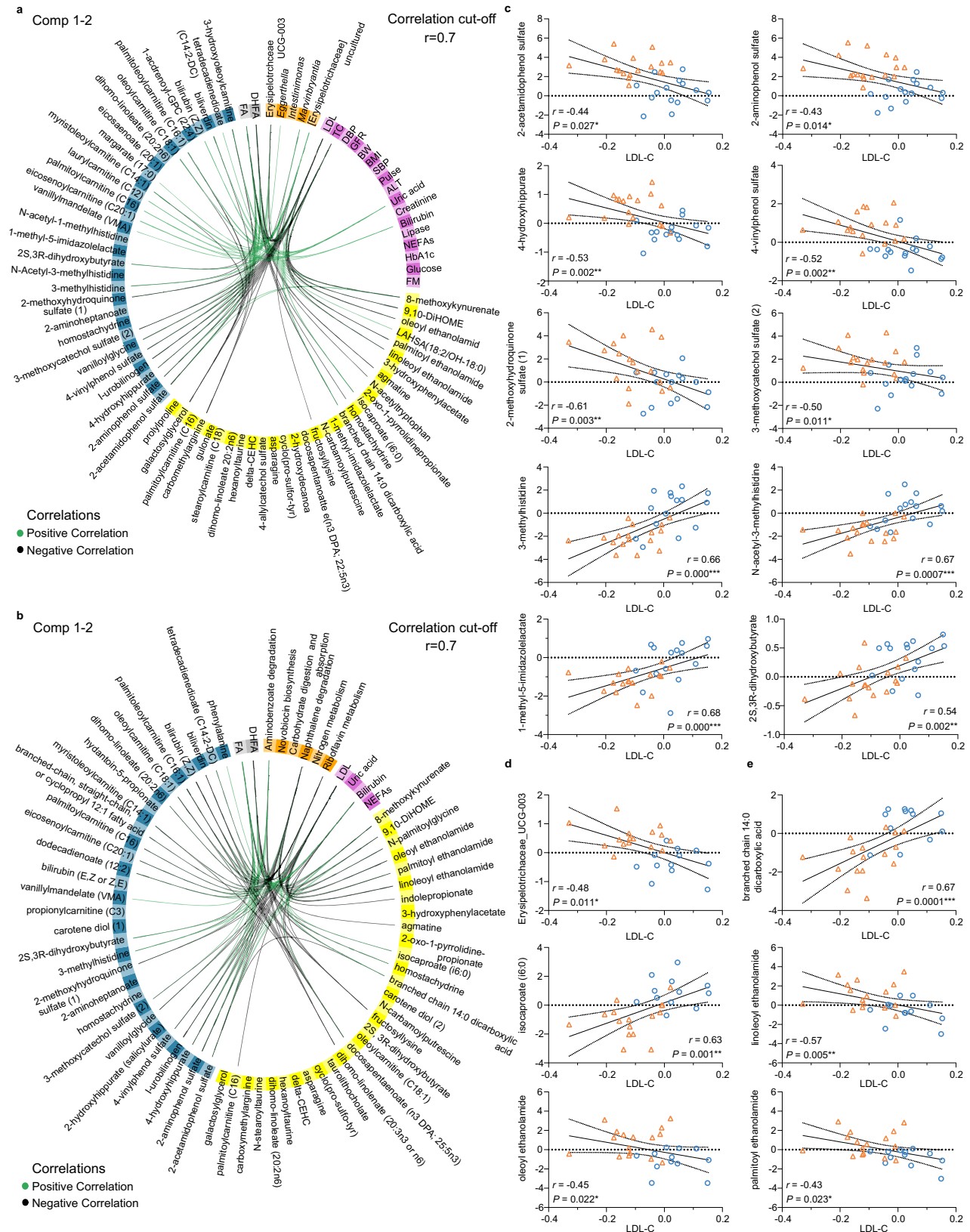

solutions and inhibit the enterohepatic circulation of bile acids by increasing their fecal excretion[56,57]. Furthermore, the effects of SCFAs produced by the bacterial fermentation of dietary fiber on lipid metabolism have already been studied in detail[58,59]; however, little is known about the role of other bioactive substances contained in oats, such as phenolic compounds. In this present work, we demonstrate

that even a two-day high-dose oat diet improved lipid metabolism by lowering serum TC (−8%) and LDL-C concentrations (−10%), confirming the long-postulated cholesterol-lowering effect of oats[22,60] and highlighting the role of oat-derived phenolic compounds and their respective microbial metabolites as possible contributing factors. So far, little is known about the effects of short-term, high-dose oat diets[61].

**Fig. 7 | Oat-derived bioactive phenolic compounds reduced cholesterol in association with the gut microbiome.** Circos plot shows positive (green) and negative (black) correlations (cutoff $r = \pm 0.7$) between the selected variables in the five data sets along component 1 and 2 derived from the DIABLO evaluations ("models") ($n = 28$). **a** Model 1.1 includes the data sets gut microbiome composition (orange), clinical markers (violet), fecal metabolites (yellow), plasma metabolites (blue), and targeted plasma metabolomic profile (DHFA, FA) (grey). **b** Model 2.1 includes the data sets microbial pathways (orange), clinical markers (violet), fecal metabolites (yellow), plasma metabolites (blue), and targeted plasma metabolomic profile (DHFA, FA) (grey). **a + b** Intra-block correlations are not presented. **c–e** Scatter plots show the associations between the log fold changes in LDL-cholesterol

and (**c**) selected plasma metabolites, (**d**) Erysipelotrichaceae UCG-003, (**e**) selected fecal metabolites in OG (presented as orange triangles) and CG (presented as blue circles). Pairwise correlations were assessed using Spearman's rank correlation coefficient. **a, b, d, e** $n = 28$ (OG: $n = 16$, CG: $n = 12$). **c** $n = 32$ (OG: $n = 17$, CG: $n = 15$). ALT alanine aminotransferase, BMI body mass index, BER balanced error rate, BW body weight, CG control group, Comp component, DHFA dihydroferulic acid, DBP diastolic blood pressure, GFR glomerular filtration rate, FA ferulic acid, FM fat mass, LDL low-density lipoprotein cholesterol, NEFAs non-esterified fatty acids, OG oat group, SBP systolic blood pressure, TC total cholesterol. Source data are provided as a Source Data file.

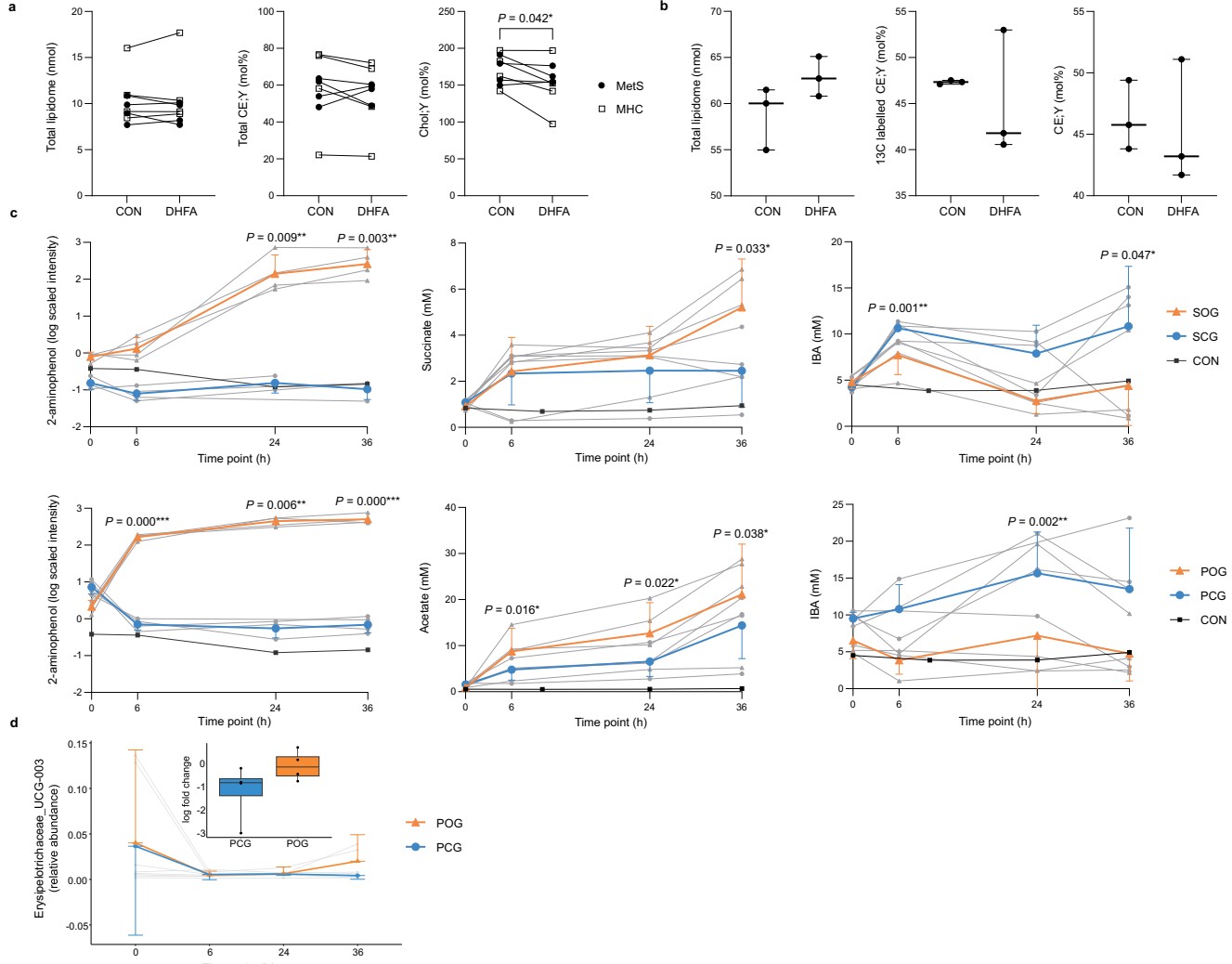

**Fig. 8 | Effects of microbially produced phenolic compounds on cholesterol metabolism in vitro. a** Impact of DHFA on the total lipidome and the molar fraction (mol%) of alkyne cholesterol esters and alkyne cholesterol of the total lipidome of PBMCs fed with alkyne cholesterol. Individual data points are shown for $n = 8$ participants (black circles: individuals with MetS, white squares: MHCs). Differences between the treatment groups (DHFA vs. CON) were analyzed using two-sided paired Student's $t$-test. **b** Impact of DHFA on the total labeled lipidome and the molar fraction (mol%) of $^{13}C$ labeled-alkyne cholesterol esters and alkyne cholesterol esters of HuH7 cells fed with $^{13}C$ labeled acetate and alkyne fatty acid 182. Samples are triplicate for each condition ($n = 3$) and presented as median and range. Differences between the treatment groups (DHFA vs. CON) were analyzed using two-sided unpaired Student's $t$-test. **c** Microbially induced changes in 2-aminophenol (log fold change) and selected SCFAs (mM) in vitro over 36 h. Data are shown as individuals data points (grey triangles: SOG or POG, grey dots: SCG or

PCG, $n = 4$ each; black squares: control) and as mean $\pm$ SD (orange: SOG or POG, blue: SCG or PCG). Differences between the treatment groups were analyzed using two-sided paired Student's $t$-test with the log fold change as input. **d** Shifts in the relative abundance of Erysipelotrichaceae UCG-003 in vitro over 36 h. Data are shown as individuals data points (grey triangles: POG, grey dots: PCG, $n = 4$ each) and as mean $\pm$ SD (orange: POG, blue: PCG). Boxplots show the corresponding log fold change at 36 h (dashed center line: median; whiskers: min and max; dots: individual data points, $n = 4$ each). Differences between the groups were analyzed using paired two-sided Student's $t$-test with the fold change of the CLR-transformed count data as input (relative abundance threshold: 0.01%); $P = 0.172$. CON control treatment (without DHFA), DHFA dihydroferulic acid, IBA isobutyric acid, MHCs metabolically healthy controls, MetS metabolic syndrome, PCG physiological control group, POG physiological oat group, SCG starving control group, SOG starving oat group. Source data are provided as a Source Data file.

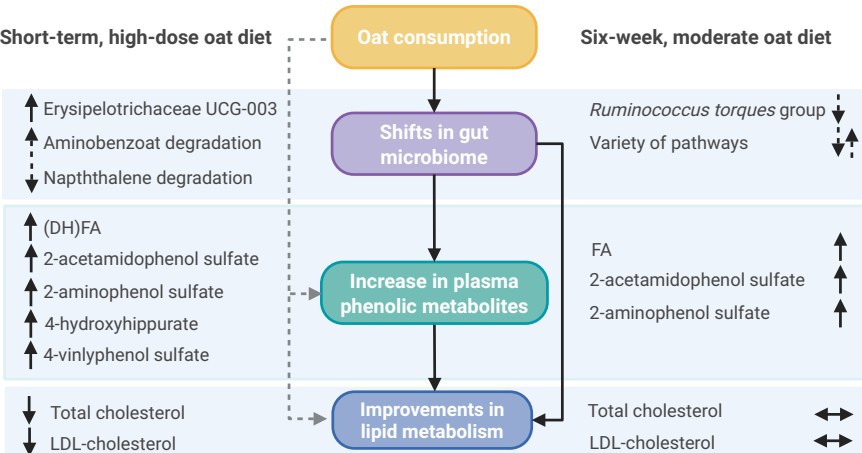

**Fig. 9 | Summary of the key findings: cholesterol-lowering effects of oats by microbially produced phenolic metabolites.** solid arrow: significant change; dashed arrow: trend; ↑ increase; ↓ decrease; ↔ no significant change. DHFA dihydroferulic acid, FA ferulic acid, LDL low-density lipoprotein. Created in Bio-Render. Simon, M. (2026) https://BioRender.com/6tkzoib[117].

Since nutritional intervention, particularly a two-day treatment, is expected to have only subtle treatment effects, the significant reduction in cholesterol, as shown in our study, is quite notable. Compared to the mean effect size of cholesterol-lowering drugs, e.g., statins, which cause an average relative reduction in LDL-C of 15% to 58%, depending on the specific product, dose (5–80 mg) and the continuous intake[62], the clinical implications of the oat diet can be considered as relevant. Especially since cholesterol levels tended to remain below the baseline value during the 6-week follow-up phase of the short-term intervention study, it would be advisable to consume a high-dose oat diet intermittently. Given the increasing prevalence of MetS, it is particularly important to identify cost-effective, easy-to-implement (e.g., concepts of intermittent fasting[63]), and sustainable strategies such as a short-term, high-dose oat diet to improve metabolism.

Our study has several strengths. We conducted two well-designed randomized controlled trials in parallel-design and confirmed our results in three different in vitro experiments. In addition, our study groups—including males and females—were characterized in depth and adherence to the respective diets was rigorously monitored using three complementary methods. By integrating multiple data sets—including various clinical markers related to MetS, metabolomic profiles in plasma and feces, as well as the gut microbiome—we provide a comprehensive understanding of the intervention effects and underlying mechanisms. However, our findings remain constrained by the limited sample size typical in clinical studies, making it susceptible to baseline imbalances, despite the randomized group allocation. As a result, modest effects of the nutritional interventions may have gone undetected due to large inter-individual variability and potential confounding factors. Furthermore, nutritional interventions in general might potentially have a stronger impact on higher or even pathological baseline level of clinical markers compared to physiological levels, which may have implications when comparing and interpreting the results of these studies. In addition, adherence to the study instruction during the follow-up period was self-reported, which is common practice in nutritional studies, though potential overestimation should be considered compared to the objective biomarker measures, which were conducted before and after the short-term and six-week intervention. Finally, the unbalanced sex distribution between the study groups may have influenced the results. To confirm our results based on an RCT and to evaluate potential sex-specific differences in the response to the intervention, further studies are needed. For example, a multicenter RCT with a sample size calculation may provide the next level of evidence in clinical research as it allows for sex-specific analysis and better generalizability across different populations and reduce a potential center-specific bias.

In conclusion, oat consumption, particularly a short-term, high-dose oat diet, provides metabolic health benefits for individuals with MetS by increasing circulating microbially produced phenolic metabolites, thereby lowering serum cholesterol levels. This identified mechanism shows that phenolic metabolites besides SCFAs are driving factors for the cholesterol-lowering properties of oats and emphasizes that interactions between oats and the gut microbiome play an important role in this health-promoting effect. The mild effects of the six-week oat diet indicate that a more personalized strategy is needed to increase the health benefits of such moderate interventions, since inter-individual differences in response to oats based on person-specific characteristics will be considered. Our results offer great potential since oat-based interventions, especially a short-term, high-dose oat diet, are a fast and effective approach to alleviate obesity-related lipid disorders, and they open new avenues for microbiota-targeted nutritional therapies, considering microbially produced phenolic metabolites as potential agents to improve metabolic disorders linked to obesity. However, whether the identified phenolic metabolites from oats contribute to lowering cholesterol levels in a specific pattern and across different populations needs to be further investigated.

## Methods

### Ethics approval and participants' consent

This study complied with all relevant ethical regulations and was conducted in accordance with the principles of the Declaration of Helsinki and its subsequent amendments, approved by the ethics committee of the Medical Faculty, University of Bonn (Approval Number: 212/20, approval date: 03 June 2020), and prospectively registered at the German Clinical Trials Register under identifier DRKS00022169. Written informed consent was obtained from all the participants.

### Study design

This study comprised two 1:1 randomized, controlled, prospective dietary interventions, each with a parallel design, conducted between September 2020 and July 2022 at the Department of Nutrition and Microbiota, University of Bonn, Germany. The short-term intervention study was completed on schedule, while the six-week intervention study was delayed due to the COVID-19 pandemic.

Potential subjects completed an initial telephone pre-screening followed by an in-person screening visit, which included anthropometric measurements and a detailed blood analysis to confirm eligibility and a detailed explanation of all study procedures and requirements. In the short-term intervention study, eligible participants were invited for two clinical visits, the first before (V1) and the second directly after the two-day intervention period (V2), followed by three clinic visits at two-week intervals during a six-week follow-up period (V3–V5) to determine potential long-term effects of the oat diet (Fig. 1a). Therefore, the follow-up period was conducted to evaluate the durability of the responses, allowing conclusions to be drawn about the optimal intervals at which a short-term, high-dose oat diet might be implemented on a regular basis. During the six-weeks, intervention study, eligible participants were invited for four clinical visits at two-weeks intervals with the first before (V1) and the last after the six-week intervention period (V4) (Fig. 1b).

After the first visit (V1), participants were randomly assigned to the experimental group (OG or OG⁶ʷ) or the control group (CG or CG⁶ʷ) of each study using computer-generated randomization tables in a block format with variable block length generated by a researcher not clinically involved in the study. These tables were concealed from the researchers and participants until the interventions were assigned and participants allocated to the respective intervention group by study personnel not involved in final data analysis. Blinding after allocation to the interventions was not feasible.

At V1 and the clinical visit directly after each intervention period (V2 or V4), anthropometric data, fecal samples as well as fasting and postprandial blood samples, taken during an oral glucose tolerance test (OGTT), were collected. Office BP and resting energy expenditure were measured using a semiautomatic BP measurement device and indirect calorimetry, respectively. In addition, an in-depth nutritional and lifestyle assessment was conducted at V1. At the clinical visits during the six-week intervention study (V2 + V3), anthropometric and BP measurements were performed and fasting blood and fecal samples were collected. These also took place at the clinical visits during the follow-up period of the short-term intervention study (V3 – V5). In addition, a fecal sample was collected after the first intervention day of the short-term intervention. Interim contact with the study coordinator (via telephone or e-mail) was made available to all participants on each intervention day.

## Participants

Participants were recruited through newspaper advertisements, flyers, and social media in the surrounding area of Bonn, Germany (first participant enrolment: September 8, 2020; last participant enrolment: May 30, 2022). The study subjects included female and male adults of Caucasian descent with overweight or obesity (BMI 27–39.9 kg/m²) and MetS aged 45–70 years. Sex and gender of each participant were determined based on self-reported information (questionnaire). MetS was primarily diagnosed based on the global consensus definition of the International Diabetes Federation[1] as central obesity measured by waist circumference (≥ 80 cm for women and ≥ 94 cm for men) and at least two of the following five criteria: (1) prehypertension (≥ 120–139 mmHg systolic and/or ≥ 80–89 mmHg diastolic)[64] or stage 1 hypertension (≥ 140–159 mmHg systolic and/or ≥ 90–99 mmHg diastolic); (2) triglyceridemia (≥ 150 mg/dL); (3) decreased fasting serum HDL-C (< 40 mg/dL for men and <50 mg/dl for women); (4) hyperglycemia (≥ 100 mg/dL); (5) indication of insulin resistance (HOMA-IR index > 2.5) (adapted to the WHO definition[65]). Further inclusion criteria were (6) non-smoker (defined as abstinence during the past 3 years prior to study enrollment[66,67]) and (7) written informed consent for participation in the study. The specific threshold values for BP were chosen because prehypertension (≥ 120 mmHg systolic and/or ≥80 mmHg diastolic) has been associated with a significantly increased risk of cardiovascular disease[68–71] and therefore represents a stage in

which preventive dietary interventions may be more effective. The fifth criterion was added after the initial study registration in order to attach more importance to impaired glucose metabolism as a characteristic of MetS and to counteract recruitment difficulties due to the COVID-19 pandemic. In addition, the subjects' habitual diet corresponded to a Western dietary pattern without regular oat consumption. Exclusion criteria were as follows: (1) regular oat consumption, defined as at least one usual portion per week; (2) chronic use of medication that affects glucose metabolism (e.g., cortisone); (3) diagnosed type 2 Diabetes mellitus, chronic liver diseases (transaminases > twice the normal value), gastrointestinal diseases, chronic inflammatory diseases (hsCRP > 30 mg/L) including rheumatoid arthritis, previous cardiovascular events, thyroid disease (e.g., untreated hypo- or hyperthyroidism), tumor diseases, acute illnesses or recent surgeries; (4) anemia; (5) pregnancy, lactation; (6) allergies and intolerances to components of the oat products; (7) antibiotic therapy within three months prior to study inclusion; (8) alcohol abuse (defined according to ICD-10/DSM-5 within the last three years, or self-reported problematic alcohol consumption requiring treatment with less than 12 months abstinence before study enrollment), medication abuse and/or drug abuse (BTMG or other psychotropic substances); (9) regular intake of dietary supplements (especially n-3 fatty acids, vitamin E, magnesium, calcium, iron, zinc); (10) planned change in lifestyle, especially participation in a weight loss program; (11) vegetarian or vegan diet; (12) participation in another clinical trial at the same time or within the last 30 days; (13) other exclusion criteria at the discretion of the doctor/investigator.

## Dietary interventions

During the short-term intervention period, the participants followed a two-day, high-dose oat diet or a macronutrient-adapted control diet, both hypocaloric and high in fiber (1100–1200 kcal/d; carbohydrates (67 energy% (E%), of which fiber > 15%, β-glucan: $13.5 \times g$ vs. $0 \times g$), protein (15 E%), and fat (17 E%)), depending on their allocation (OG vs. CG). Participants in the OG consumed three oat meals per day instead of their habitual diet. Each meal consisted of $100 \times g$ rolled oat flakes (Demeterhof Schwab GmbH & Co. KG, Windsbach, Germany) and has been consumed as porridge, prepared with water. Fruits restricted to apples, pears, and berries and vegetables restricted to spinach and leeks were used as additives and added to the meal plans of the CG in the same amount. No salt, sugar, or sweeteners were added. Participants in the CG consumed two meals per day comprising bread and raw vegetables (breakfast and dinner) and one warm meal per day (lunch) instead of their habitual diet. There was a time interval of four hours between meals. During this time, participants were required to consume only unsweetened beverages. For standardization, the participants received a recommendation for dinner the evening before the start of the intervention (carbohydrate-rich bread meal with raw vegetables). During the six-week follow-up period, participants of the OG adhered to their habitual diet without consuming oats.

During the six-week dietary intervention, participants in the OG⁶ʷ replaced one meal per day with an oat meal consisting of $80 \times g$ oat flakes (Demeterhof Schwab GmbH & Co. KG) which corresponds to $8 \times g$ dietary fiber ($3.5 \times g$ β-glucan), while maintaining their habitual diet under isocaloric conditions. For the preparation of the oat meal, the participants received a collection of recipes that included classic porridges, overnight oats, smoothies, and baked goods and was adapted to the daily calorie requirements of each participant individually, measured using indirect calorimetry. The participants in the CG⁶ʷ maintained their habitual dietary pattern unchanged and did not consume any oats.

In both the short-term and six-week intervention period, the meals were prepared independently by the subjects at home using detailed instructions. Participants were instructed to avoid any other oat products (other than the oat flakes provided) for at least two weeks before and during the study. The nutrient composition of all test meals

was calculated using the computer-based nutrient calculation program EBISpro, based on the German nutrient database Bundeslebensmittelschlüssel, version 2016 (Max Rubner-Institut, Karlsruhe, Germany).

## Primary and secondary outcomes

All outcomes reported in the article are listed in the study protocol and registry and thus have been defined prior to the start of the study.

The primary outcome was the plasma concentration of (DH)FA and secondary outcomes were parameters of glucose (plasma glucose, serum insulin, HbA1c, glucagon) and lipid metabolism (lipids and lipoproteins (TC, HDL, triglycerides (TG)), free fatty acids, serum lipase); parameters of kidney (uric acid, urea, creatinine) and liver function (GOT, GPT, GGT, bilirubin) which, together with a small blood count, were also used as safety parameters; anthropometric parameters, resting energy expenditure and blood pressure which, together with energy and nutrient intake, were also used as control parameters; gut microbiota composition and diversity based on fecal samples; avenanthramides concentrations in plasma which served as a compliance marker; as well as global metabolomic profiles in plasma and feces which have been considered as optional secondary outcomes in the study protocol, with the objective to investigate the effects of a short-term, high-dose oat diet and a six-week, moderate oat diet compared with a macronutrient-adapted control diet and a Western diet, respectively, on lipid metabolism, the gut microbiota, and metabolomic profiles, in particular the phenolic compounds FA and DHFA, in individuals with MetS and to elucidate the underlying mechanism using an integrative multi-omics analysis.

Further secondary outcomes listed in the study protocol and registry were not included in the article for the following reasons: (i) they could not be assessed due to the COVID-19 pandemic and/or resulting budget constraints, time delays, and loss of cooperation partners (concentrations of AVA in urine samples, glucagon, GLP-1, GLP-2, GIP, tumor necrosis factor α (TNF-α), neurofilament light (NFL), parameters of antioxidant status (plasma β-carotene, α-tocopherol, vitamin C, vitamin A, oxidized LDL-C), ghrelin, leptin and adiponectin (optional), DNA methylation and SNP genotyping, and—originally only planned for a subgroup of the six-week, intervention study—inflammation of the subcutaneous adipose tissue (adipocytes), browning of adipocytes, fatty acid composition in the erythrocytes and expression of the G-protein coupled receptors (GPR) in the adipose and intestinal tissue); (ii) they are outside the scope of the main research question and would exceed the manuscript's focus (amino acids (tryptophan, tyrosine), serum electrolytes (potassium, sodium, magnesium), neurotransmitters (serotonin)); or (iii) they were analyzed as part of a secondary subgroup analysis, which is reported in a separate manuscript (short-chain fatty acids (SCFAs), zonulin, interleukin-6 (IL-6) and lipopolysaccharides (LPS)).

## Compliance

Adherence to the treatment protocol was assessed based on three independent criteria. Firstly, the concentration of the oat-specific biomarker AVAs[72,73] was quantified as an objective blood marker. Plasma AVAs concentration was determined using LC-MS/MS by Metabolon Inc. (Morrisville) (Metabolon Method TAM223: "LC-MS/MS Quantitation of Dihydroferulic Acid and Three Avenanthramide Compounds in Human Plasma") based on a previously described method[73] (see below). Secondly, the participants received the exact number of oat packages required for each intervention period and were asked to return all empty and unemptied packages after completion of the respective intervention period. Thirdly, the participants who were assigned to the oat diets (OG or OG⁶ʷ) or the control diet of the short-term intervention (CG) completed a detailed checklist on each intervention day to record precise information on the selection and preparation of the respective test meals.

During the six-week follow-up period in the oat group of the short-term intervention (OG), compliance with the study instruction (i.e., refraining from oat consumption and returning to habitual Western diet) was assessed using two complementary methods: (1) specific query by the study staff about abstinence of oat meals as consumed during intervention and any other oat products, and (2) 3-day dietary records completed by the participants at baseline (V1) and at the end of the follow-up period (V5).

## Anthropometrics

Anthropometric measurements were performed following a standard operative procedure[74]. Body weight was measured in light clothing with an empty bladder, using electronic column scales with an accuracy of up to $100 \times g$ (seca scale 704, seca GmbH and Co. KG, Hamburg, Germany). Body height was determined to the nearest of 0.1 cm using a stadiometer (seca scale 704, seca GmbH and Co. KG, Hamburg, Germany). The BMI was calculated using the following formula: BMI = weight [kg]/(height [m])$^2$. Waist circumference was measured midway between the lowest rib and iliac crest at maximal exhalation to the nearest of 0.1 cm in duplicates. Body composition (fat mass and fat-free mass) was determined by air-displacement plethysmography using a BOD-POD body composition system (Cosmed, Firdolfingen, Germany).

## Office blood pressure and heart rate

Office blood pressure and heart rate were measured twice in a seated position using a semiautomatic blood pressure measurement device (Boso Carat Professional, Bosch + Son GmbH and Co. KG, Juningen, Germany) under standardized conditions in accordance with European and American guidelines[75,76].

## Indirect calorimetry

For determination of whole-body resting energy expenditure (REE) and substrate oxidation via measurement of carbon dioxide production and oxygen consumption, indirect calorimetry was performed, using Quark-RMR® (Cosmed, Fridolfing, Germany) with a coefficient of variation of <10%[77]. REE was multiplied by 1.5 (physical activity level, PAL) to calculate the total energy expenditure.

## Nutrition and lifestyle assessments

The habitual diet was characterized using a Food Frequency Questionnaire (FFQ) over 12 months. Physical activity (7-day total metabolic equivalent of task score) was assessed at baseline using the short version of the validated international physical activity questionnaire (IPAQ)[78,79]. For the evaluation of the sleeping behavior (circadian rhythm), the validated Munich chronotype questionnaire (MCTQ) was used[80,81]. Socio-economic status was obtained using a questionnaire based on the parent questionnaire to measure socioeconomic status in the study on the health of children and adolescents in Germany (KiGGS) Wave 2[82].

## Collection of blood and stool samples

Fasting venous blood samples were collected in the morning between 8:00 a.m. and 10:00 a.m. after a 12-h overnight fast. Postprandial blood samples were collected during a 3-h OGTT. All samples were taken under standardized conditions using tubes containing EDTA, fluoride, or a coagulation activator (S-Monovette, Sarstedt, Germany). Plasma and serum supernatants were obtained by centrifugation at $3000 \times g$ for 15 min at 8 °C after complete coagulation (only serum) and immediately frozen in cryovials at −80 °C until further analysis. Fecal samples were collected before each clinic visit according to a standard operation procedure and immediately stored at −80 °C until further analysis[83].

## Oral glucose tolerance test

Oral glucose tolerance test was performed after an overnight fast with $75 \times g$ glucose and 300 ml water (Dextrose O.G-T.; Roche Diagnostics,

GmbH, Mannheim, Germany). Venous blood samples were collected via a venous catheter before ingestion and at 15, 30, 45, 60, 120, and 180 min postprandially to analyze the time course of plasma glucose, serum insulin, serum TG and serum non-esterified fatty acid (NEFA) concentrations.

Various indices were calculated to assess the diet-induced effects on glucose control and insulin resistance. Insulin sensitivity was evaluated using the oral glucose insulin sensitivity index (OGIS)[84,85] and the insulin sensitivity index (ISI), which provide a measurement of insulin-mediated glucose clearance[86]. β-Cell function was assessed using the insulinogenic index[84,85], disposition index[87–89] and HOMA-β[90]. In addition, insulin resistance was evaluated using the quantitative insulin sensitivity check index (QUICKI)[91,92] and the triglyceride-glucose index (TyG)[93,94] based on fasting concentrations. Moreover, the area under the curve (AUC) of the glucose and insulin concentration over 180 min, calculated using the trapezoidal rule and considering only complete data sets, was used to measure differences in glucose tolerance upon intervention. The AUCs of the NEFA and TG concentration were calculated in the same way over 120 min. Furthermore, the presence of metabolic dysfunction-associated steatotic liver disease (MASLD) was estimated using the fatty liver index (FLI ≥ 60)[95,96].

## Routine laboratory blood analyses

Routine laboratory analyses of serum included fasting and postprandial TG and insulin levels, fasting TC, LDL-C, HDL-C, and high-sensitivity C-reactive protein (hsCRP) levels, and clinical biochemistry. Routine laboratory analyses of plasma included fasting and postprandial glucose, fasting HbA1c, and hematology parameters. All parameters were measured in a certified medical laboratory (Central Laboratory of the Institute of Clinical Chemistry and Clinical Pharmacology at the University Hospital Bonn, Germany) within 4 h of blood sampling under standardized conditions using the Roche/Hitachi Cobas c system (Roche Diagnostics, Mannheim, Germany). Methods specifications are available online (https://www.ukbonn.de/ikckp/zentrallabor/leistungsverzeichnis/). The HOMA index was calculated as follows: HOMA-IR = [insulin (mU/L) × glucose (mg/dL)] ÷ 405[90]. Insulin resistance was assumed if the HOMA index was > 2.5[97].

## Analysis of non-esterified fatty acids

Fasting and postprandial serum NEFA concentrations were analyzed pre- and post-intervention using an in vitro enzymatic colorimetric method assay (NEFA-HR(2), Wako Diagnostics, Mountain View, CA, USA) following the manufacturer's instructions and the recommended quality control procedure (inter-assay/intra-assay variability 5.4%/4.8%) at the Institute of Nutrition and Food Science, University Bonn, Germany[98].

## In vitro cell culture experiments

To confirm that the identified phenolic compounds mediate the beneficial effect of oats on cholesterol metabolism, we performed in vitro cell culture experiments. Briefly, blood was collected from participants with MetS ($n = 4$; 65 ± 9 y, 29.3 ± 2.2 kg/m²) and MHCs ($n = 4$; 33 ± 11 y, 25.1 ± 4.9 kg/m²) in EDTA-containing cuvettes to isolate peripheral blood mononuclear cells (PBMCs). After pre-processing, PBMCs were exposed to medium with or without 10 μM DHFA (Sigma-Aldrich, #17803-5 G) that additionally contained 20 μg/mL alkyne cholesterol. In addition, HuH7 cells (JCRB0403) were treated with the same medium with or without 10 μM DHFA after 48 h of cultivation in growth medium. The medium was additionally supplemented either with 20 μg/mL alkyne cholesterol or 5 mM 1,2-$^{13}C_2$ sodium acetate (CIL, #CLM-440-1) and 100 μM alkyne fatty acid 182. After 24 h of incubation, the samples were subjected to lipid extraction and dissolved lipids were analyzed by MS.

In detail, PBMCs were isolated by density gradient centrifugation with Pancoll (density 1.077 g/mL, PAN Biotech, # P04-601000). After washing of the cells with PBS and removal of platelets, PBMCs were plated in RPMI Medium (PAN Biotech, # P04-22100) supplemented with Penicillin/Streptomycin and 10% Fetal Bovine Serum with a density of 2 × 10⁶/mL into a 96 well round bottom plate (Sarstedt, # 83.3925). After 2 h incubation, PBMCs were exposed to medium with or without 10 μM DHFA (Sigma-Aldrich, #17803-5 G). Additionally, the medium was supplemented with 20 μg/mL alkyne cholesterol. After 24 h, PBMCs were washed first with 200 μL ice-cold 0.5% DL-BSA (Carl Roth, #0052.3) in PBS and then with 200 μL ice-cold 155 mM Ammonium acetate in $H_2O$. Between every washing step, plates were centrifuged at 610 × g for 5 min and supernatant was decanted. Subsequently, 250 μL of PBMC extraction mix (230 μl MeOH/CHCl₃ 5/1, 20 μl internal standard mix containing alkyne-labeled lipids: 103 pmol TG 48:3-d8, 14.6 pmol DG 32:3-d8, 26 pmol CE 17:2-d7, 15.8 pmol dh-alkyne-Cer15:1-d8, 13 pmol PA 32:3-d8, 113 pmol PC 32:3-d8, 36 pmol PE 32:3-d8, 30 pmol PI 32:3-d8, 34 pmol PS 32:3-d8 and 20 pmol of double-labeled TG 49:5-d8 and nonalkyne labeled lipids: 250 pmol PE 33:1-d7, 472 pmol PC 33:1-d7, 98 pmol PS 31:1, 84 pmol PI 34:0, 56 pmol PA 31:1, 51 pmol PG 28:0, 39 pmol LPA 17:0, 35 pmol LPC 17:1, 38 pmol LPE 17:0, 32 pmol Cer 17:0, 240 pmol, SM 18:1-d9, 55 pmol GlcCer 12:0, 339 pmol TG 50:1-d4, 111 pmol, CE 18:1-d6 and 64 pmol DG 31:1) was added to each well and plates were placed into an ultrasound bath for 20 s. Samples were collected and then subjected to lipid extraction.

HuH-7 cells (JCRB0403) were cultivated in growth medium (DMEM 4.5 g l⁻¹ glucose, 10% FCS plus penicillin/streptomycin), seeded into 12-well plates and allowed to grow for 48 h. HuH-7 cells were exposed to medium with or without 10 μM DHFA (Sigma-Aldrich, #17803-5 G). Additionally, the medium contained either 20 μg/mL Alkyne cholesterol or 5 mM 1,2-13C2 sodium acetate (CIL, #CLM-440-1) and 100 μM alkyne fatty acid 182, and incubated for 24 h. Cells on 12-well plate were washed once with ice-cold 0.5% DL-BSA in PBS and quickly once with 155 mM ammonium acetate, taking care to remove the liquid after the last wash as complete as possible. To each well, 500 μl of extraction mix (480 μl MeOH/CHCl3 5/1, 20 μl internal standard mix containing alkyne-labeled lipids: 206 pmol TG 48:3-d8, 29.2 pmol DG 32:3-d8, 52 pmol CE 17:2-d7, 31.6 pmol dh-alkyne-Cer15:1-d8, 26 pmol PA 32:3-d8, 226 pmol PC 32:3-d8, 72 pmol PE 32:3-d8, 60 pmol PI 32:3-d8, 68 pmol PS 32:3-d8 and 41 pmol of double-labeled TG 49:5-d8 and nonalkyne labeled lipids: 500 pmol PE 33:1-d7, 944 pmol PC 33:1-d7, 196 pmol PS 31:1, 168 pmol PI 34:0, 112 pmol PA 31:1, 102 pmol PG 28:0, 78 pmol LPA 17:0, 70 pmol LPC 17:1, 76 pmol LPE 17:0, 64 pmol Cer 17:0, 480 pmol, SM 18:1-d9, 110 pmol GlcCer 12:0, 678 pmol TG 50:1-d4, 222 pmol, CE 18:1-d6 and 128 pmol DG 31:1) was added and the entire plate was sonicated for 1 min in a bath sonicator. The extracts including the cell remnants were collected into 1.5-ml original Eppendorf tubes and centrifuged at 20,000 × g for 5 min to pellet protein. The supernatants were transferred into fresh tubes. After addition of 400 μl of CHCl₃ and 600 μl of water, samples were shaken for 30 s and centrifuged for 5 min at 20,000 × g. The upper phase was removed, and the lower phase transferred into a fresh tube and dried for 20 min at 45 °C in a speed-vak. CHCl₃ (8 μl) was added and the tubes briefly vortexed. To each tube, 40 μl of Click mix was added (prepared by mixing 20 μl of 50 mM C171 in 50% MeOH (stored as aliquots at −80 °C) with 200 μl of 5 mM Cu(I)AcCN₄BF₄ in AcCN and 800 μl of ethanol), followed by sonication for 5 min and incubation at 40 °C for 16 h. 250 μl of CHCl₃ and 600 μl of water were added per sample and briefly shaken and centrifuged for 2 min at 20,000 × g. The upper phase was removed and the lower phase dried in a speed-vac as above. 1000 μl of spray buffer (2-propanol/methanol/water 8:5:1 + 10 mM ammonium acetate) was added, the tubes sonicated for 5 min and the dissolved lipids analyzed by MS.

Mass spectra were recorded on a Thermo Q-Exactive Plus spectrometer (Instrument Version 2.8 Build 280502, Software Version 2.13 SP1 Build 3170) equipped with a standard HESI ion source using direct injection from a Thermo Dionex AS-AP autosampler driven by an AXP-

MS pump under the control of Xcalibur software (Version 4.7.69.37). Samples were sprayed at flow rates of 10 µl min⁻¹ in spray buffer with the following parameters: sheath gas 6, aux gas 2, sweep gas 0, gas heating off, spray voltage positive mode 4.1 kV and ion transfer capillary temperature 280 °C. MS1 spectra (resolution 280,000) were recorded in 100 m/z windows from 250 to 1200 m/z (positive mode), followed by recording MS/MS spectra (resolution 280,000) by data independent acquisition in 1 m/z windows from 250 to 1200 m/z (positive mode). Typical scan parameters were for MS1 scans: automatic gain control target $5 \times 10^5$, maximum ion time 200 ms, resolution 280,000, peak mode centroid, for MS2 scans: automatic gain control target $2 \times 10^5$, maximum ion time 100 ms, resolution 280,000, no spectral multiplexing, fixed first mass at 100 m/z, isolation window $m/z$ 0.7, stepped normalized collision energy (NCE) 10, 35, 40, spectrum data type centroid. The raw data were converted to .mzML files using MSConvert (Version 3.0.24326-c89bdab) and analyzed with LipidXplorer software (Version 1.2.8.1). For identification and quantification of labeled lipids, molecular fragment query language (.mfql) files were written that identify the species by the presence of a peak corresponding to the expected masses of the lipid class. For further analysis, absolute amounts were calculated using internal standard intensities, followed by the calculation of the molar fraction for the identified lipids.

### Fecal batch culture fermentation of in vitro digested oats

To evaluate if the metabolism of the microbiome results in the production of the identified oat-derived metabolites, in particular the phenolic compounds, we conducted an anaerobic in vitro fecal batch culture experiment. In brief, aliquots of homogenized baseline stool samples from four representative participants of the short-term intervention study were used to inoculate sterile anoxic basal medium either with or without in vitro digested oats in a starving group (anoxic basal buffer; SOG vs. SCG) or a physiological group (Brain Heart Infusion medium; POG vs. PCG). Samples for global metabolomic profiling including phenolic compounds, SCFAs and microbiome analysis were taken directly after inoculation, after 6, 24h, and 36 h of incubation time.

In detail, oat flakes (Demeterhof Schwab GmbH & Co. KG, Windsbach, Germany) were digested in vitro according to a modified method described by Kristek et al.[99]. Oat flakes ($60 \times g$) were homogenized in 150 ml of sterile distilled water and microwaved for one minute at 950 W. Subsequently, 20 mg α-Amylase (from *Aspergillus oryzae*, Sigma-Aldrich; dissolved in 6,25 ml of 1 mM CaCl₂) was added and the mixture incubated for 30 min at 37 °C and 200 rpm on an incubation shaker. After incubation, the pH of the solution was adjusted to 2 using 6 M HCl, and $2.7 \times g$ Pepsin (Carl Roth) in 25 ml of 0.1 M HCl was added, followed by 2 h of incubation at 37 °C and 200 rpm. From there, 560 mg of pancreatin (from porcine pancreas, Sigma-Aldrich), $3.5 \times g$ bile salts (Sigma-Aldrich), and 125 ml of 0.5 M NaHCO₃ were added, and the pH was adjusted to 7.0 with 5 M NaOH. After 3 h of further incubation, the solution was transferred into a regenerated cellulose dialysis tubing (Spectra/Por® 7, Carl Roth) with 1 kDa MWCO and dialyzed against 0.01 M NaCl at 5 °C for 15 h. Subsequently, the dialysis fluid was exchanged, and dialysis continued for 2 h. Finally, the dialyzed solution was lyophilized for 5 days to remove any fluid content.

For the fecal batch culture experiments, 500 mg of stool samples that were taken each from four representative participants prior to the short-term intervention study were homogenized in 5 ml of sterile, anoxic basal buffer (0.5 g/l bile salts (Sigma-Aldrich), 0.01 g/l CaCl₂ × 2 H₂O, 0.04 g/l K₂HPO₄, 0.01 g/l MgSO₄ × 7 H₂O, 0.04 g/l KH₂PO₄, 0.1 g/l NaCl, 2 g/l NaHCO₃, flushed with 90% N₂/10% CO₂). Subsequently, the slurry was divided into 2.5 ml aliquots until further use. To mimic a fasting period for the fecal microbiota in comparison to physiological conditions, one aliquot per participant was used to inoculate either

50 ml of sterile, anoxic basal buffer (starvation group) or 50 ml of sterile anoxic Brain Heart Infusion medium (BHI, flushed with 90% N₂/10% CO₂; physiological group) in serum flasks sealed with butyl rubber stoppers. After incubation overnight at 37 °C, the starvation and the physiological groups were each used to inoculate ($3\%_{v/v}$) 50 ml of sterile, anoxic basal medium (see Kristek et al.[99]; flushed with 90% N₂/10% CO₂) either with or without $3.3 \times g$ of anoxic, sterile in vitro digested oat flakes (from 66 g/l stock solution in 5 mM potassium phosphate buffer, pH 7.5, flushed with N₂; corresponding to an oat meal based on a colon volume of 1.5 l) in serum flasks sealed with butyl rubber stoppers. Finally, the cultures were incubated at 37 °C over 36 h. Samples for global metabolomic profiling, including phenolic compounds, SCFAs, and microbiome analysis, were taken directly after inoculation, after 6, 24, and 36 h of incubation time.

**SCFA profiling.** The concentrations of SCFAs were determined using isocratic HPLC (Agilent 1260 Infinity II System equipped with a VWD dual wavelength detector, Agilent Technologies). Briefly, samples were centrifuged for 2 min at $12,000 \times g$ and the supernatants were sterile filtered (Filtropur S, pore size: 0.2 µm, Sarstedt). The filtrate was diluted 1:5 with 5 mM TFA and finally analyzed on an Aminex HPX-87H $300 \times 7.8$ mm column (Bio-Rad Laboratories) at 60 °C with 5 mM TFA as the mobile phase (flow rate of 0.5 ml/min). To determine the absolute concentrations of every SCFA species in each sample, UV absorption at 210 nm was quantified against respective standard curves using OpenLab CDS software version 2.6.

### Targeted plasma metabolomic profile (primary outcome)

Plasma dihydroferulic acid concentration, the primary outcome of this study, and ferulic acid values (peak areas) before and after each intervention period were generated by Metabolon Inc. (Morrisville) using LC-MS/MS according to Metabolon Method TAM223 ("LC-MS/MS Quantitation of Dihydroferulic Acid and Three Avenanthramide Compounds in Human Plasma")[18].

In detail, human plasma samples were spiked with stable labeled internal standards and then incubated with glucuronidase and sulfatase enzymes to convert conjugated metabolites to their free form. After incubation, the samples were subjected to protein precipitation with acetonitrile. Next, the samples were centrifuged, and the supernatant dried down under nitrogen and reconstituted with water containing formic acid. The samples were mixed and injected onto an Agilent 1290/AB Sciex QTRAP 7500 LC-MS/MS system equipped with a BEH C18 reversed phase UHPLC column. The flow rate was 625 µL/min. The mass spectrometer was operated in negative mode using electrospray ionization (ESI).

The peak areas of the respective analyte product ions were measured against the peak area of the corresponding internal standard product ions (e.g., dihydroferulic acid, avenanthramides). Quantitation was performed using a linear regression with a 1/× weighting generated from fortified calibration standards prepared immediately prior to each run. LC-MS/MS raw data were collected and processed using AB SCIEX software Analyst 1.6.3 and processed using SCIEX OS-MQ software v1.7. Data reduction was performed using Microsoft Excel for Office 365 v.16.

### Global metabolomic profiling

For both fasting plasma and fecal samples, non-targeted global metabolomic profiles before and after each intervention period were generated by Metabolon Inc. (Research Triangle) using UPLC-MS/MS[24,25] (Supplementary Data 6). The metabolomic dataset of both intervention studies comprised a total of 1149 and 1082 metabolites in plasma and feces, respectively. In the supernatant of the batch culture experiment a total of 995 metabolites was identified. This includes amino acids, peptides, carbohydrates, energy intermediates, lipids, nucleotides, cofactors and vitamins, xenobiotics, and partially

characterized molecules of both endogenous and an established microbial origin[100]. After removing drug-associated metabolites, 1128 plasma metabolites and 1054 fecal metabolites (applies to both interventions) were included in the analyses.

A detailed description of the procedure (Metabolon Platform) is provided below.

**Sample accessioning.** Following receipt, samples were inventoried and immediately stored at −80 °C. Each sample received was accessioned into the Metabolon LIMS system and was assigned by the LIMS a unique identifier that was associated with the original source identifier only. This identifier was used to track all sample handling, tasks, results, etc. The samples (and all derived aliquots) were tracked by the LIMS system. All portions of any sample were automatically assigned their own unique identifiers by the LIMS when a new task was created; the relationship of these samples was also tracked. All samples were maintained at −80 °C until processed.

**Sample preparation.** Samples were prepared using the automated MicroLab STAR® system from Hamilton Company. Several recovery standards were added prior to the first step in the extraction process for QC purposes. To remove protein, dissociate small molecules bound to protein or trapped in the precipitated protein matrix, and to recover chemically diverse metabolites, proteins were precipitated with methanol under vigorous shaking for 2 min (Glen Mills GenoGrinder 2000) followed by centrifugation. The resulting extract was divided into multiple fractions: two for analysis by two separate reverse phase (RP)/UPLC-MS/MS methods with positive ion mode electrospray ionization (ESI), one for analysis by RP/UPLC-MS/MS with negative ion mode ESI, one for analysis by HILIC/UPLC-MS/MS with negative ion mode ESI, while the remaining fractions were reserved for backup. Samples were placed briefly on a TurboVap® (Zymark) to remove the organic solvent. The sample extracts were stored overnight under nitrogen before preparation for analysis.

**QA/QC.** Several types of controls were analyzed in concert with the experimental samples: a pooled matrix sample generated by taking a small volume of each experimental sample (or alternatively, use of a pool of well-characterized human plasma) served as a technical replicate throughout the data set; extracted water samples served as process blanks; and a cocktail of QC standards that were carefully chosen not to interfere with the measurement of endogenous compounds were spiked into every analyzed sample, allowed instrument performance monitoring and aided chromatographic alignment. Specifically, the Metabolon platform includes three distinct types of Metabolon QC samples and two different Metabolon QC standards: (i) the QC sample "MTRX" is a large pool of human plasma maintained by Metabolon that has been characterized extensively and used to assure that all aspects of the Metabolon process are operating within specifications; (ii) the QC sample "CMTRX" is a pool created by taking a small aliquot from every participant sample to assess the effect of a non-plasma matrix on the Metabolon process and distinguish biological variability from process variability; (iii) the QC sample "PRCS", an aliquot of ultra-pure water, serves as a process blank to assess the contribution to compound signals from the process; (iv) the recovery standard (RS) is used to assess variability and verify performance of extraction and instrumentation; and v) the internal standard (IS) assesses variability and performance of instrument.

Instrument variability was determined by calculating the median relative standard deviation (RSD) for the standards that were added to each sample prior to injection into the mass spectrometers. Overall process variability for feces was determined by calculating the median RSD for all endogenous metabolites (i.e., non-instrument standards) present in 100% of the Client Matrix samples, which are technical replicates of pooled client samples. Overall process variability for plasma was determined by calculating the median RSD for all endogenous metabolites (i.e., non-instrument standards) present in the MTRX7 technical replicates. Values for instrument and process variability meet Metabolon's acceptance criteria (median RSD internal standards: 6% (plasma) vs. 7% (feces), median RSD endogenous biochemicals: 11% (plasma) vs. 12% (feces)). Experimental samples were randomized across the platform run with QC samples spaced evenly among the injections.

**Ultrahigh performance liquid chromatography-tandem mass spectroscopy (UPLC-MS/MS).** All methods utilized a Waters ACQUITY ultra-performance liquid chromatography (UPLC) and a Thermo Scientific Q-Exactive high resolution/accurate mass spectrometer interfaced with a heated electrospray ionization (HESI-II) source and Orbitrap mass analyzer operated at 35,000 mass resolution[24]. The dried sample extract was then reconstituted in solvents compatible to each of the four methods. Each reconstitution solvent contained a series of standards at fixed concentrations to ensure injection and chromatographic consistency. One aliquot was analyzed using acidic positive ion conditions, chromatographically optimized for more hydrophilic compounds (PosEarly). In this method, the extract was gradient eluted from a C18 column (Waters UPLC BEH C18-2.1 × 100 mm, 1.7 μm) using water and methanol, containing 0.05% perfluoropentanoic acid (PFPA) and 0.1% formic acid (FA). Another aliquot was also analyzed using acidic positive ion conditions; however, it was chromatographically optimized for more hydrophobic compounds (PosLate). In this method, the extract was gradient eluted from the same aforementioned C18 column using methanol, acetonitrile, water, 0.05% PFPA and 0.01% FA and was operated at an overall higher organic content. Another aliquot was analyzed using basic negative ion optimized conditions using a separate dedicated C18 column (Neg). The basic extracts were gradient eluted from the column using methanol and water, however, with 6.5 mM Ammonium Bicarbonate at pH 8. The fourth aliquot was analyzed via negative ionization following elution from a HILIC column (Waters UPLC BEH Amide 2.1 × 150 mm, 1.7 μm) using a gradient consisting of water and acetonitrile with 10 mM Ammonium Formate, pH 10.8 (HILIC). The MS analysis alternated between MS and data-dependent $MS^n$ scans using dynamic exclusion. The scan range varied slightly between methods but covered 70–1000 m/z. Raw data files are archived and extracted as described below.

**Bioinformatics.** The informatics system consisted of four major components, the Laboratory Information Management System (LIMS), the data extraction and peak-identification software, data processing tools for QC and compound identification, and a collection of information interpretation and visualization tools for use by data analysts. The hardware and software foundations for these informatics components were the LAN backbone, and a database server running Oracle 10.2.0.1 Enterprise Edition.

**LIMS.** The purpose of the Metabolon LIMS system was to enable fully auditable laboratory automation through a secure, easy to use, and highly specialized system. The scope of the Metabolon LIMS system encompasses sample accessioning, sample preparation and instrumental analysis and reporting and advanced data analysis. All of the subsequent software systems are grounded in the LIMS data structures. It has been modified to leverage and interface with the in-house information extraction and data visualization systems, as well as third party instrumentation and data analysis software.

**Data extraction and compound identification.** Raw data was extracted, peak-identified, and QC processed using a combination of Metabolon developed software services (applications). Each of these services perform a specific task independently, and they

communicate/coordinate with each other using industry-standard protocols. Compounds were identified by comparison to library entries of purified standards or recurrent unknown entities. Metabolon maintains a library based on authenticated standards that contains the retention time/index (RI), mass to charge ratio (*m/z*), and fragmentation data on all molecules present in the library. Furthermore, biochemical identifications are based on three criteria: retention index within a narrow RI window of the proposed identification, accurate mass match to the library +/− 10 ppm, and the MS/MS forward and reverse scores between the experimental data and authentic standards. The MS/MS scores are based on a comparison of the ions present in the experimental spectrum to the ions present in the library spectrum. While there may be similarities between molecules based on one of these factors, the use of all three data points is utilized to distinguish and differentiate biochemicals. More than 5,400 commercially available purified or in-house synthesized standard compounds have been acquired and analyzed on all platforms for determination of their analytical characteristics. An additional 7000 mass spectral entries have been created for structurally unnamed biochemicals, which have been identified by virtue of their recurrent nature (both chromatographic and mass spectral). These compounds have the potential to be identified by future acquisition of a matching purified standard or by classical structural analysis. Metabolon continuously adds biologically-relevant compounds to its chemical library to further enhance its level of Tier 1 metabolite identifications.

**Compound quality control.** A variety of curation procedures were carried out to ensure that a high-quality data set was made available for statistical analysis and data interpretation. The QC and curation processes were designed to ensure accurate and consistent identification of true chemical entities, and to remove or correct those representing system artifacts, mis-assignments, mis-integration, and background noise. Metabolon data analysts use proprietary visualization and interpretation software to confirm the consistency of peak identification and integration among the various samples.

**Metabolite quantification and data normalization.** Peaks were quantified using area-under-the-curve. For studies spanning multiple days, a data normalization step was performed to correct variation resulting from instrument inter-day tuning differences. Essentially, each compound was corrected in run-day blocks by registering the medians to equal one (1.00) and normalizing each data point proportionately (termed the "block correction"). For studies that did not require more than one day of analysis, no normalization is necessary, other than for purposes of data visualization. In certain instances, biochemical data may have been normalized to an additional factor (e.g., cell counts, total protein as determined by Bradford assay, osmolality, etc.) to account for differences in metabolite levels due to differences in the amount of material present in each sample.

**Gut microbiome sample processing**
Total genomic DNA was extracted from 120 mg fecal material or from pelleted material from 700 μl cell suspension using ZR BashingBead lysis tubes (0.1 and 0.5 mm, Zymo Research, Freiburg, Germany) in combination with the Chemagic DNA stool kit (Perkin Elmer, Rodgau, Germany) following the manufacturer's instructions[101–103]. After the addition of lysis buffer, mechanical lysis was performed using a Precellys 24 tissue homogeniser (Bertin Instruments, Frankfurt am Main, Germany). After extraction, the DNA was stored at −20 °C until further analysis. High-throughput 16S rRNA amplicon sequencing of the fecal microbiome was performed at Life & Brain GmbH (Bonn, Germany). In brief, the V3V4 region of the 16S rRNA gene was amplified in a first PCR step using the Bakt_341F (5′-CCTACGGGNGGCWGCAG-3′) and Bakt_805R (5′-GACTACHVGGGTATCTAATCC-3′) primers combination. The 25 μL PCR reaction contained 2.5 μL of template (5 ng/μL), 12.5 μL of

2× KAPA HiFi HotStart ReadyMix (Roche, Mannheim, Germany), and 5 μL of primer mix (1 μM). Thermal cycling conditions comprised an initial denaturation at 95 °C for 3 min, followed by 25 cycles of denaturation (30 s at 95 °C), annealing (30 s at 55 °C), and elongation (30 s at 72 °C), with a final elongation step at 72 °C for 5 min. In a subsequent PCR, dual indices and Illumina sequencing adapters were incorporated using the Nextera XT v2 Index Kit (Illumina, San Diego, CA, USA). The second PCR reactions consisted of 25 μL 2× KAPA HiFi HotStart ReadyMix, 5 μL Nextera XT index primers, and 10 μL PCR-grade water, in a total volume of 50 μL per sample. Cycling conditions were an initial denaturation at 95 °C for 3 min, followed by 8 amplification cycles of denaturation (30 s at 95 °C), annealing (30 s at 55 °C), and elongation (30 s at 72 °C), finalized by a 5 min elongation at 72 °C. After each PCR step, amplicon libraries were quality-checked using an Agilent TapeStation 4200 with D1000 ScreenTape (Santa Clara, CA, USA) and purified with AMPure XP beads (Beckman Coulter, Krefeld, Germany). Samples were normalized to 4 nM and pooled in equimolar amounts[104]. The ZymoBIOMICS Gut Microbiome Standard (D6331, Zymo Research, Freiburg, Germany) was used as the positive control, while ultrapure water (rH$_2$O) served as the negative control. The final pool was quantified using the Qubit dsDNA HS assay kit (Thermo Fisher Scientific, Waltham, MA, USA), and the fragment size was checked using a D1000 ScreenTape (Agilent, Santa Clara, CA, USA).

Sequencing was performed on an Illumina MiSeq system using the MiSeq reagent kit v3 with 2 × 300 cycles. Clustering was performed at 8 pM with a 20% spike-in of PhiX. Demultiplexing was performed using the MiSeq system. For the human intervention studies, the average number of paired-end (PE) reads was 76,504, with a minimum of 29,392 PE reads. In the batch culture fermentation experiment, the average number of PE reads was 137,854 (excluding blank and t0 samples), with a minimum of 31,045 PE reads. Blank samples yielded an average of 552 PE reads, with a minimum of 246 PE reads. t0 samples had an average of 21,587 PE reads, with a minimum of 4246 PE reads. The 16S rRNA sequencing data were processed using QIIME 2 version 2021.4[105]. Sequence quality control and denoising were performed using DADA2[106]. The QC step included the filtering of PhiX reads and chimeric sequences. After denoising, sequences were classified using the SILVA database (138.1 SSU Ref NR 99) to identify amplicon sequencing variants (ASVs) for sequences with >99% sequence similarity.

**Gut microbiome analysis**
Diversity analyses were performed in QIIME2 based on a rarefied table with a sampling depth of 28,767 sequences (short-term intervention) and 21,876 sequences (six-week intervention), respectively, and were used for subsequent statistical analysis. We quantified alpha diversity using Shannon entropy, Pielou's evenness, and Faith's phylogenetic diversity (Faith-PD). Alpha diversity was assessed per sample, and a LMM was applied to evaluate the differences between the two diet groups in each study. The LMM was adjusted for the confounders sex, age, and BMI (LMM$^{adj.}$), and estimated the diet effect relative to the baseline value and the control group (CG or CG$^{6w}$). An individual-specific random intercept was included to account for differences in baseline alpha diversity. The model was defined as: alpha diversity ~ time*group + age + sex + BMI + (1|Person-ID), with the covariate of interest "time*group".

We quantified beta diversity using Jaccard distance, Bray-Curtis dissimilarity, and UniFrac distance matrices. As beta diversity measures differences in community composition, we computed it with respect to each individual's baseline microbiome. For the analysis of beta diversity—which considers the change with respect to baseline—we used a linear regression model (LnR), as baseline differences are accounted for by the metric itself. Beta diversity was modeled as a function of the diet groups, using the control group of each study (CG or CG$^{6w}$) as the reference, and was adjusted for the confounders sex,

age, and BMI (LnR$^{adj.}$). The model was defined as: beta diversity ~ group + age + sex + BMI, with the covariate of interest 'group'.

To identify differences between the diet groups of the RCTs at feature level, linear models for differential abundance analysis (LinDA)[27], an established state-of-the-art method for analyzing microbiome data, was applied. The control group of each study (CG or CG$^{6w}$) and the baseline value were used as reference and the Person-ID was used as random effect. The model was defined as: taxon ~ time\*group + age + sex + BMI + (1|Person-ID), with the covariate of interest "time\*group". The analyses were performed at genus level with a relative abundance threshold of 0.01%. Data were normalized using a centered-log-ratio (CLR) transformation with an offset of 1 prior to all analyses. Twenty-eight participants from the short-term intervention ($n = 4$ dropouts [3 female, 1 male] due to an incomplete data set) and all 34 participants from the six-week intervention were included in the respective analysis.

PICRUSt2 (version 2.5.1)[30] was used to predict the functional potential of the fecal microbiota. Such functional prediction tools play a valuable role in facilitating preliminary hypothesis testing based on amplicon sequences; however, this method has inherent limitations in accurately reproducing functional potential. It should be acknowledged that the prediction model based on 16S or shotgun sequencing data does not confirm whether functional capacity is activated at the mRNA or protein level. Given the prohibitive cost of shotgun sequencing, functional prediction tools serve as a valid guide for subsequent studies of actual functional capacity. Predicted gene products in each bacterium were annotated using the KEGG database and classified into KEGG orthologous (KO) groups[29], resulting in the identification of 6616 KO (short-term intervention: $n = 28$) and 6424 KO (six-week intervention: $n = 34$) across all samples. The ko2kegg_abundance function of the ggpicrust2 package[107] was used for this task. In accordance with LinDA, a LMM was applied to the CLR-transformed count data with the control group of each study (CG or CG$^{6w}$) and the baseline value as reference and the Person-ID as random effect (LMM$^{adj.}$). The model was defined as: pathway ~ time\*group + age + sex + BMI + (1|Person ID), with the covariate of interest "time\*group". The analyses were performed with a relative abundance threshold of 0.01%.

## Sparse partial least squares-discriminant analysis

To identify the key features that differentiate the two groups in each intervention study in terms of diet-induced changes in clinical markers, gut microbiota composition and functional capacity, and global metabolomic profiles in plasma and feces, sPLS-DA was performed. sPLS-DA, implemented in the mixOmics v6.8.5 R package, enables the selection of the most discriminative features in the data to classify the samples by projecting the data into a lower dimensional space in a supervised manner[20,21]. This approach is optimal if the number of features is very high compared to the sample size, as in this present work. To account for the repeated-measures design of both intervention studies, the logarithmic fold change (log fold change) was used as the input to the models based on Lee et al.[108]. The performance of the sPLS-DA was assessed on the BER and AUC, which were calculated using a cross-validation approach. The results were visualized using two-component sample plots and loadings.

A detailed description of data processing and procedure is provided below.

**Data processing.** For the targeted plasma metabolomic profile, DHFA concentrations below the limit of quantitation, but above the peak area of the background DHFA peak in the blank samples, or concentrations above the limit of quantitation were extrapolated. Missing values in the global metabolomic profiles were imputed with the minimum observed value for each compound (%missing values short-term vs. six-week intervention study: 12% vs. 13% (plasma), 24% vs. 26%

(feces)). This is in line with the literature, reporting that missing values are quite common in large untargeted metabolomics datasets and might comprise up to 50% of the dataset[109]. Metabolic and metabolomics data were transformed with the natural logarithm prior to the analysis. Microbiome data were normalized by using CLR transformation with an offset of 1. The log fold change was calculated for all data sets and used as the input to the models.

**Procedure.** The number of features and PLS components was chosen via stratified fivefold cross-validation with 25 random repeats, according to the best-balanced classification error rate (BER). The models were calibrated and assessed by fivefold-cross-validation. For each fold, absolute shrinkage and selection operator (LASSO)[110] was used on the training data to minimize the number of features per component while maximizing class discrimination. BER and AUC were calculated for the final model on the test data. As result, the averages across the folds were presented.

The data set of the clinical markers, which comprised 26 variables, was transformed into two components with three and 19 clinical markers in the short-term intervention study, and into one component with all 26 clinical markers in the six-week intervention study. The microbiome data set, which comprised 143 or 146 genera (short-term vs. six-week), was transformed into two components with one and 41 genera in the short-term intervention study, and into one component with one microbe in the six-week intervention study. For the KEGG pathway data set, which comprised 145 pathways, two components with ten and five features were selected in the short-term intervention study. In the six-week intervention study, six components with 35, 13, two, two, five, and five pathways were selected from a total of 146 KEGG pathways. The model of the global plasma metabolomic profile comprised two components with 30 and 95 metabolites in the short-term intervention study and one component with 15 metabolites in the six-week intervention study, each selected from a total of 1128 metabolites. For the global fecal metabolomic profile, two components with 91 and 971 metabolites in the short-term intervention study and one component with 31 metabolites in the six-week intervention study, each selected from a total of 1054 metabolites, were selected as the best models. An overview of all selected features and their weight loadings are provided in the Supplementary Data 5. Further detailed information on the workflow can be found in the tutorials of mixOmics (http://mixomics.org/case-studies/).

## Integrative multi-omics analysis

To determine the relationship between diet-induced changes in the clinical markers, gut microbiota, and targeted and global metabolomic profiles, DIABLO, an integrative multi-omics method that aims to identify common information across different types of data while distinguishing between different phenotypes[32], in this study between the diet groups of each intervention trial, was applied for each study separately. Two different DIABLO evaluations ("models") were performed. For model 1, clinical markers, metabolomic profiles including targeted plasma, global plasma, and global fecal profiles, and the microbial composition were used as inputs (short-term intervention: model 1.1; six-week intervention: model 1.2). Model 2 considered the functional potential of the microbiota instead of the composition (short-term intervention: model 2.1; six-week intervention: model 2.2). The log fold changes were used as inputs for the models[108]. The performance of the final models was assessed on the BER and AUC derived from internal cross-validation of the model. The results were presented using weight loadings and Circos plots (correlation cut-off $r = \pm 0.7$).

In detail, DIABLO is a novel framework for the integration of multiple data sets in a supervised analysis that aims to identify key variables/common information across different data sets (e.g., clinical markers, metabolomic profiles, microbiome data) during integration

process while distinguishing between different phenotypes (e.g., study groups)[32]. Thus, the correlated information between multiple datasets is maximized. DIABLO extends sparse generalized canonical correlation analysis (sGCCA)[111] to the sPLS-DA framework for selecting co-expressed (correlated) variables from different datasets. To account for the repeated-measure design of the intervention studies, log fold change of the preprocessed data as described above was used as an input to the models. An initial partial least square analysis between the five data sets of model 1 including clinical marker, microbiome data (genera), targeted plasma metabolomic profile ((DH)FA), and global plasma and fecal metabolomic profiles showed correlations in the range of 0.66–0.96 (model 1.1: short-term intervention study) and 0.61–0.93 (model 1.2: six-week intervention study), therefore a square matrix filled with 0.75 and diagonal set to zeros was used as an input to the design parameter of the DIABLO model. The correlations for model 2, which included KEGG pathways instead of microbiome genera, ranged from 0.63–0.96 (model 2.1: short-term intervention study) and from 0.35–0.93 (model 2.2: six-week intervention study), so that the same input was used for the design parameters of the DIABLO model.

Tuning of the DIABLO model was done similar to the tuning of the sPLS-DA model as described above. An overview of all selected features and their weight loadings are provided in the Supplementary Data 5. The final model 1.1 consisted of two components. Component one comprised one plasma metabolite of the targeted analysis (FA), three clinical markers (LDL, TC, DBP), four bacterial genera (Erysipelotrichaceae UCG-003, *Marvinbryantia*, *Eggerthella*, *Intestinimonas*), and 15 metabolites each of the global plasma and fecal metabolomic profile. Component two comprised one plasma metabolite of the targeted analysis (DHFA), 15 clinical markers, one bacterial genus ([Erysipelotrichaceae] uncultured), and 15 metabolites each of the global plasma and fecal metabolomic profile. The final model 2.1 consisted of two components. Component one comprised one plasma metabolite of the targeted analysis (FA), one clinical marker (LDL), five KEGG pathways (naphthalene degradation, carbohydrate digestion and absorption, novobiocin biosynthesis, aminobenzoate degradation, nitrogen metabolism), and 15 metabolites each of the global plasma and fecal metabolomic profile. Component two comprised one plasma metabolite of the targeted analysis (DHFA), three clinical markers (bilirubin, NEFAs, uric acid), two KEGG pathways (aminobenzoate degradation, riboflavin metabolism), and 15 metabolites each from of the global plasma and fecal metabolomic profile. The final model 1.2 consisted of two components. Component one comprised one plasma metabolite of the targeted analysis (FA), one clinical marker (HDL), 21 bacterial genera, one plasma metabolite (prolylglycine), and eleven fecal metabolites of the global metabolomic profile. Component two comprised one plasma metabolite of the targeted analysis (DHFA), one clinical marker (hsCRP), eleven bacterial genera, and one plasma metabolite (21-hydroxypregnenolone disulfate) and eleven fecal metabolites of the global metabolomic profiles. The final model 2.2 consisted of two components. Component one comprised one plasma metabolite of the targeted analysis (FA), one clinical marker (insulin), 31 KEGG pathways, and one plasma metabolite (prolylglycine) and 31 fecal metabolites of the global metabolomic profiles. Component two comprised one plasma metabolite of the targeted analysis (DHFA), one clinical marker (hsCRP), 21 KEGG pathways, and eleven plasma metabolites and one fecal metabolite (N-palmitoyltaurine) of the global metabolomic profiles. Further detailed information on the workflow can be found in the tutorials of mixOmics (http://mixomics.org/mixdiablo/diablo-tcga-case-study/).

### Partial least square regression model

To indicate the predictive value of the microbial and phenolic changes in modulating cholesterol levels, partial least square (PLS) regression models were applied[112]. The models were implemented using the mixOmics package in R (version 4.2.2). The analyses were performed without sparse mode, i.e., all predictor variables identified in the previous DIABLO models (cutoff $r = 0.7$) were included in the models without variable selection. $R^2$ and $Q^2$ total (cross-validated $R^2$), calculated by leave-one-out cross-validation, were used to assess the fit and predictability of the model, respectively[112]. The log fold change of the identified features (nine phenolic compounds or two genera and one pathway) was used as predictor and the log fold change of total and LDL-cholesterol as response.

### Calibration analysis

Out-of-fold (OOF) predictions for each DIABLO and sPLS-DA model were generated using fivefold cross-validations with 25 repeats. For each participant, the Oats–Control score margin were taken and converted it to a probability with Platt scaling (logistic regression fit only to the OOF data). Probabilities were grouped into quantile bins; the bin mean vs. the observed oats rate with 95% Wilson intervals and overlaid a loess smoother were visualized. AUC was derived from cross-validation. Since the dashed 45° line marks perfect calibration, the majority of the calibration plots did not indicate a bias and showed that the predicted probabilities match the observed outcomes well, in particular, the DIABLO models and the sPLS-DA model of the global plasma metabolomic profile, from which the main outcome of our study were derived, performed very well (Supplementary Fig. 4).

### Statistics and sample size calculation

Statistical analyses were performed using SPSS (version 29.0; IBM Crop., Chicago, IL, USA), R (version 3.6.2; Boston, MA, USA), and Python (version 3.10). GraphPad Prism (version 10; GraphPad Software, San Diego, CA, USA), R, and Python were used for creating figures. For all analyses, a two-tailed significance level was set at $P < 0.05$. For metabolic data analysis, family-wise error rate correction was applied via the block-wise Bonferroni–Holm method[113] to correct for multiple testing ($P^{adj.} < 0.05$). This stringent multiple comparison method is particularly suitable for hypothesis-driven analyses as in the present case. For the microbiome and global metabolomic data with a high noise-to-signal ratio, FDR correction based on the Benjamini/Hochberg method[114] was applied ($q < 0.05$), which is frequently used for high-throughput data.

Prior to the analysis, the distribution of the data was visualized individually. Additionally, Shapiro-Wilk tests were performed to test for normality of the data and determine the appropriate approach for further analysis. Variables were analyzed either on their original scale or logarithmic transformed to approximate a normal distribution.

Postprandial results were described as the AUC for the 3-h OGTT, calculated using the trapezoidal method and only complete data sets.

To assess whether the data provided evidence of the superiority of the oat diets over the respective control in terms of our primary outcome and the metabolic and metabolomic outcomes, linear regression models were applied, including the baseline value and the confounding factors BMI, age, and sex (LnR$^{adj.}$) as covariates. The control group of each study (CG or CG$^{6w}$) was used as reference. The model was defined as: "post-intervention_value ~ baseline_value + group + age + sex + BMI". Beta estimates of intervention (β) with 95% CI were reported to assess the magnitude of the effect size. In addition, a LMM adjusted for BMI, age, and sex (LMM$^{adj.}$) was used to assess differences in clinical markers across all four time points of the six-week intervention study. The control group (CG$^{6w}$) and the baseline value were used as reference, the Person-ID was used as random effect to account for repeating measurements. The model was defined as: clinical marker ~ time*group + age + sex + BMI + (1|Person-ID), with the covariate of interest "time*group". Residuals obtained from the models were inspected for normality to control for the fit of the statistical model. If the residuals were not normally distributed, logarithmic transformation was applied.

To determine the changes over time within each study group of the RCTs (pre- vs. post-intervention), a paired Student's $t$-test (or Wilcoxon rank test) was performed depending on the data distribution. In addition, to account for the changes within the OG, including the follow-up period, a LMM adjusted for BMI, age, and sex (LMM$^{adj.}$) was build using the baseline value as reference and the Person-ID as random effect.

Pairwise correlations between diet-induced significantly altered clinical markers and changes in the microbiome and metabolomic profiles were assessed using Pearson's and Spearman's rank correlation coefficient. Only variables with at least five data points were included, using log fold change for analysis. Pairwise correlations were primarily intended to support the DIABLO results; therefore, no FDR correction was applied.

Sample size was calculated based on data from a previous intervention study that successfully assessed the effect of whole grain intake on blood DHFA concentration[16], expecting a twofold change in the plasma DHFA concentration between the two diet groups in each intervention study using a two-sided $t$-test at a 5% significance level and with 95% power. Thus, a sample size of 14 people per group was required. Assuming a dropout rate of 20%, 17 people per group have been included (34 people per study). In line with the SAGER guidelines, sex-disaggregated analyses should be pursued; however, due to the limited sample size, stratification by sex was not conducted.

### Reporting summary

Further information on research design is available in the Nature Portfolio Reporting Summary linked to this article.

## Data availability

The raw microbiome 16S rRNA data and the metabolomic data (global metabolomic profiles) generated in this study have been deposited in the open repository *Zenodo* (https://doi.org/10.5281/zenodo.16312602)[115]. These data and the de-identified personal data (including clinical and biochemical metadata) that support the findings of this study are not openly available due to reasons of sensitivity and are available from the corresponding author. These data are protected and are not openly available due to data privacy laws. Access to the data and additional, related documents like the study protocol can be obtained by contacting the corresponding author of the paper (Jun. Prof. Dr. Marie-Christine Simon, e-mail address: mcsimon@uni-bonn.de) and signing a data-sharing agreement in compliance with the EU General Data Protection Regulation (GDPR) and national data protection laws. Data sharing will be considered for researchers with a methodologically sound proposal aiming at analyses aligned with the scientific aims of this study. A response to requests for data access might be expected within two months, and the data will be available for one year after signing the data-sharing agreement. Source data are provided with this paper. Additional metabolomics related files are provided with this paper as Supplementary Data 6.

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

## Acknowledgements

The authors thank Anke Ernst and Christel Bierschbach (Institute of Nutrition and Food Science, Nutritional Physiology, University of Bonn, Bonn, Germany), and Anke Carstensen (Institute of Clinical Chemistry and Clinical Pharmacology, Central Laboratory, University Hospital Bonn, Bonn, Germany) for their excellent assistance during the study, including routine laboratory analysis. In addition, the authors thank Per Hoffmann, Markus Nöthen, and the Next Generation Sequencing (NGS) Core Facility for providing access to the laboratory technologies necessary for microbiome analysis and sequencing. This work was supported by Diet–Body–Brain (DietBB), Competence Cluster in Nutrition Research funded by the Federal Ministry of Education and Research (BMBF; grant no. 01EA1707; M.C.S), the German Diabetes Association (DDG; www.ddg.info; M.C.S.), the German Research Foundation (DFG) under Germany's Excellence Strategy (EXC 2047—390685813, J.H.; EXC 2151—390873048, M.C.S and J.H.), the project ID 450149205—TRR 333 (J.H.) and the project ID 432325352—SFB 1454 (M.C.S. and J.H.), the German Cereal Processing, Milling and Starch Industries' Association (VGMS e.V.; www.vgms.de; M.C.S. and L.K.) and RASO Naturprodukte (www.raso.de; L.K.). The funding sources did not contribute to the study design, data collection and analysis, decision to publish, or preparation of the manuscript for the oat study or present analysis.

## Author contributions

The authors' responsibilities were as follows: L.K. and M.C.S.: designed research; L.K., S.B., C.T. and M.C.S.: conducted research; L.K., W.S., L.S., M.H.Y., V.W., and B.S.W.: performed the laboratory analysis; L.K., A.M., M.P., W.S., L.S., M.H.Y., V.W., L.W., and T.F.: analyzed data, performed statistical analyses, and created figures; M.C., M.S., J.H., T.F., S.B., C.T., P.S., and M.C.S.: supervised the trial and the laboratory and data analysis; L.K. and M.C.S.: wrote the paper; M.C.S.: had primary responsibility for the final content. All authors read and approved the final manuscript and adhered to the author's ethics and inclusion criteria.

## Funding

## Competing interests

The authors declare no competing interests.
