## [Transparent Peer Review file · Nature Communications]

Cholesterol-lowering effects of oats induced by microbially produced phenolic metabolites in metabolic syndrome: a randomized controlled trial

Corresponding Author: Ms Marie-Christine Simon

Version 1:

Reviewer comments:

Reviewer #1

(Remarks to the Author)

Key results: Please summarise what you consider to be the outstanding features of the work.

Microbial degradation products might be the driving factors for the oat-induced decrease in serum cholesterol levels.

Changes in a number of metabolites of lipid and amino acid metabolism might contribute to the cholesterol-lowering effect in the gut microbiome composition and function due to oats.

Validity: Does the manuscript have flaws which should prohibit its publication? If so, please provide details.

The sections of the article do not align with CONSORT guidelines. Adjust sections to the following order Methods, Results, Discussion. Include a specific and more detailed Conclusion.

Originality and significance: If the conclusions are not original, please provide relevant references. On a more subjective note, do you feel that the results presented are of immediate interest to many people in your own discipline, and/or to people from several disciplines?

The conclusions are original but have indistinct links to previously conducted studies. It will be of interest to those working within Metabolic syndrome clinical trials specifically those who have investigated the properties of oats to improve lipid metabolism.

Data & methodology: Please comment on the validity of the approach, quality of the data and quality of presentation. Please note that we expect our reviewers to review all data, including any extended data and supplementary information. Is the reporting of data and methodology sufficiently detailed and transparent to enable reproducing the results?

The data analysis is very detailed in both supplementary tables and in the body of the text. Readers may find it hard to identify the more notable significance of the trials main data points due to the dense description of each data point. The lack of description surrounding secondary and tertiary outcomes could subject the trials to being considered of more exploratory nature rather than controlled which is noticeable by the extensive description of the listing of data points. Authors might condense findings from most significant to least significant as opposed to listing all findings and direct readers to supplementary tables for specific but non-notable findings.

Appropriate use of statistics and treatment of uncertainties: All error bars should be defined in the corresponding figure legends; please comment if that's not the case. Please include in your report a specific comment on the appropriateness of any statistical tests, and the accuracy of the description of any error bars and probability values.

Statistical tests are appropriate and have been explained where used but with little justification for why a method was valid for the particular data.

Conclusions: Do you find that the conclusions and data interpretation are robust, valid and reliable?

The conclusions are valid based on the data analysis presented in this paper.

Suggested improvements: Please list additional experiments or data that could help strengthening the work in a revision.

References: Does this manuscript reference previous literature appropriately? If not, what references should be included or excluded?

Appropriate references have been included

Clarity and context: Is the abstract clear, accessible? Are abstract, introduction and conclusions appropriate?

The abstract is clear however the statistics has been reduced to a percentage so could be more detailed.

Inflammatory material: Does the manuscript contain any language that is inappropriate or potentially libelous?

The manuscript does not contain any language that is inappropriate or potentially libelous

Reviewer #2

(Remarks to the Author)

This interesting study discusses the cholesterol-lowering effects of oats induced by microbially produced phenolic metabolites in individuals with metabolic syndrome through dietary interventions and multi-omics analysis.

The message is that short term high oat diet induces pronounced effects on microbial composition and metabolism, whilst a 6 wk moderate oat diet has very slight effects. This overall message has to be emphasized and discussed more. Diet composition at product/food items level has to be reported more extensively. When you add 100 g oat to one meal 3 times a day in the short term, how does that change overall diet composition and to what extent can differences be ascribed to oat?.

What are the clinical implications of the current findings, how long do effects of high oat last and what would be the advise, to consume consistently a high oat diet or intermittently?.

Statistical analyses like sPLS-DA and DIABLO were used for data interpretation. Many correlations were reported. Is it also possible to indicate the predictive value of microbial and phenolic changes in modulating cholesterol?

Is it possible to add a figure with major outcomes both short-term and long-term?

Reviewer #3

(Remarks to the Author)

Linda et al conducted two randomized clinical trial in subjects with MetS, and demonstrated that gut microbiota-derived phenolic metabolites mediate the cholesterol-lowering effect of oats. The study is interesting and of significance, however, several major concerns preclude the publication.

Major

1. Overall, this study is entirely based on bioinformatics analysis and all the data are correlative in nature. Given that 16S rRNA seq is not suitable for microbial function analysis, I don't think the data, especially for microbial metabolites, provided in the manuscript is convincing. Moreover, no in vitro or in vivo experiments have been carried out to validate or prove that phenolic compounds mediate the beneficial effect of oat on cholesterol metabolism.
2. As is described by the authors, SCFAs are also important microbial products of oats. Then, do the authors detect SCFAs in plasma and fecal samples before and after the intervention? If yes, how about its change? Given the established beneficial effect of SCFAs on cholesterol metabolism, how can you make sure that the modulating effect is exerted by phenolic compounds without detecting or adjusting for the effect of SCFAs?
3. The design and description of the two trials are wired. For the short-term intervention, I am not sure the change in phenolic compounds, gut microbiota, and clinical data refers to day 2 post intervention or the lasting effect by week 6 post intervention?
4. When comparing the change of certain metabolites or clinical parameter between two intervention groups, the baseline level of this metabolite or clinical parameter should also be included in the model.

Minor

1. Why the authors restrict the age of study participants to 45-70 yrs?
2. Why history of smoking or alcoholism is excluded? Ex-smokers are also excluded? The exclusion criteria seem to be too strict, which would substantially affect the applicability of this study.
3. For both untargeted and targeted metabolomics, more details should be provided. How plasma and fecal samples are extracted. What's the detailed setup of the LC-MS/MS system? The flow rate, the column used and how features or peaks are identified (including the database used) should be specified to ensure reproducibility.
4. Prediction of microbial genes using 16s rRNA seq data is not accurate enough. The authors should at least acknowledge this limitation clearly in the manuscript.
5. Why do the authors choose different methods for multiple correction in metabolomics and microbiota analysis?

Version 2:

Reviewer comments:

Reviewer #2

(Remarks to the Author)

The authors have addressed my comments sufficiently

Reviewer #3

(Remarks to the Author)

The authors have addressed most of my concerns and the manuscript has been substantially improved. However, several minor issues need to be addressed before being accepted.

1. Although the study is RCT in nature and follows good clinical research practice, I don't think you can claim that this is the highest level evidence. It should be noted that only multi-center RCT with a large sample size can be regarded as a high level. The authors should clearly acknowledge this limitation.

2. With a small sample size and strict inclusion criteria, the generalizability of this finding is inevitably influenced. Therefore, the authors should be more careful when drawing the conclusion and clearly acknowledge this limitation.

Reviewer #4

(Remarks to the Author)

The study by Klümpen et al goes deeper into examining the potential pathways related to the gut microbiome underlying a more established association of oats consumption with lower cholesterol levels among people with metabolic syndrome (MetS). They have conducted two well-designed parallel randomised controlled trials to look at both the short-term effects of a larger dose and the effects of a more consistent lower dose on several markers related to MetS, but also metabolic biomarkers including microbiotic end products. The thorough study design and statistical analyses, as well as minor deviations from the protocol despite the challenges during the COVID-19 pandemic are commendable. The authors have made the most of their data by integrating multiple sources of data to show how these may interact. The combination of two RCTs with in vitro experiments further strengthens the presented conclusions. The authors also addressed most of the reviewers' comments from the first round of review in detail.

There are four main comments. First, in order to put the conclusions into context, some important information is missing. Although compliance as measured with the biomarker FA (as well as DHFA) is presented for immediately after the intervention, there is no information on participant compliance during the follow-up. Ideally, if these biomarkers were measured during follow-up, the results should be presented, otherwise at least the results from the diet records should be presented. For example, participants might have liked the oat meals or thought that they are good for their health and might have kept a higher intake during follow-up even if they were instructed not to do so. For the interpretation of the results as "persistent effects", it needs to be clear that deviations from compliance did not happen.

Second, in some cases it looks like the randomisation may not have worked optimally as shown from the descriptive table e.g. the sex imbalance for the short-term intervention between the two groups. It is good that you have adjusted for sex in your analyses, but such an observation along with any other observed imbalances needs to be addressed in the discussion for possible implications on the interpretation of the results.

Third, as discussed in more detail below, sPLS-DA may overfit the data so any metrics reported e.g. AUC should ideally come from an internal validation. It is not clear so far whether this is what happens or the AUC is derived from the whole sample. Since this part of the analysis is related to prediction, relevant guidelines should be followed and ideally calibration should also be assessed (see below for more details).

Finally, the discussion does not include any strengths and most importantly limitations. It would be very important to highlight any limitations of these studies (e.g. not optimal randomisation etc) for the results to be interpreted more correctly, but also to inform future studies.

Please see some more detailed comments below.

Abstract: It is missing some more quantitative measures. The previous reviewer 1's comment has not been addressed. It is worth trying to reduce the number of words in other places in the abstract and include estimates with their 95% CIs rather than just the percentages.

Introduction: The rationale for why certain metabolites were chosen could be clearer – looks like mainly phenols were chosen. FA and AVAs are mentioned in line 30, then it switches to 'phenols' in line 44, then back to FA and DHFA in line 52 then two other phenols are introduced in the brief summary of results (line 56/57). Perhaps 'phenols such as...' to start would set this up to be clearer throughout the text.

Line 51: 'metabolism' this early on in the paper is a bit vague – perhaps 'components of metabolic syndrome' or 'lipid metabolism' would be clearer.

Results

Lines 93 and 99: It is not clear what you mean by "showed a visceral fat distribution". Do you mean high visceral fat? Abdominal or central obesity?

Line 103: typo – Replace 'tabel' with table

Lines 89-103: As mentioned above, would it be worth highlighting some differences that may show that randomisation did not happen optimally in some cases? For example, in the short-term intervention the CG has 60% men, while the OG has 35% men. In addition, SBP and TG are higher in the short-term intervention compared with the long-term one. These may have implications when comparing results between the CG and OG as well as the two interventions.

Lines 129-131: Is it correct that the oat biomarkers FA and DHFA were measured only at V1 and V2? Did you measure them at the rest of the visits? This would make it clearer whether the effects on lipids and BP at follow-up were indeed persistent

or were because the participants changed their diet. If not then it would be good to include information from the dietary records to complement this result and state that based on the records participants did not change their diet, so indeed these can be considered persistent effects.

Line 135: Supplementary Figure 1 does not show any major differences for glucose, so it is probably not worth highlighting this in your statements in lines 133-136.

Line 217 (and 228): Could you add "but not statistically significantly" after "modulated the microbial functions differently" for accuracy?

Methods

Lines 513-514: Do you think there are any implications from dropping the number of total visits from 9 as initially planned to 4 or would the information you get still be sufficient?

Can you clarify why one study is isocaloric but the other is hypocaloric? And why did you plan to look for persistent changes after the intervention in the 48h intervention but not the 6 week one?

Line 542: The International Diabetes Federation in the 2009 paper that you cite defines hypertension as Systolic ≥ 130 and/or diastolic ≥ 85 mm Hg (or antihypertensive treatment with a history of hypertension). You have cited a different paper for your blood pressure cut-offs and used their definition of elevated blood pressure rather than hypertension (≥ 120 and/or diastolic ≥ 80 mm Hg). It is very helpful that you explained why you added the insulin resistance criterion, can you likewise clarify why you changed the blood pressure cut off from that suggested by the IDF?

Line 557: In the rebuttal to reviewer 3, it states that ex-smokers were not excluded but in line 557, it states that one of the exclusion criteria is 'history of smoking'. This might need re-wording to make clear whether those with a past history of smoking were excluded or not.

Lines 731-744: From the description, it looks like the LMM and the LnR were used for the same purpose, but they contradict each other in terms of accounting for random effects. Could you clarify whether indeed the two methods were used for the same purpose and if that is the case which one would be superior and what is the advantage of using both?

sPLS-DA (775-788 and supplementary methods): You mention that "Missing values in the global metabolomics profiles were imputed with the minimum observed value for each compound.". How confident are you that all the values were missing for example because they were below the low limit of quantification, which would indeed justify imputation with the minimum observed value? Could there be other reason of missing for some of the values that would justify other types of imputation (depending on the % missing this could be multiple imputation or imputation to the mean/median). In addition, could you include the %missing? Finally, sPLS-DA is known for overfitting the data. You mention that you used 5-fold cross-validation to choose the number of features and components, which is a good start to partly account for the overfitting. Could you clarify whether the AUC was also derived from the cross-validation process or the whole dataset? To avoid overfitting, it may make sense to report the AUC from the cross-validation model that you selected based on BER. Finally, since this is a prediction model, to comply with the guidelines of reporting prediction models (see for example TRIPOD <https://pubmed.ncbi.nlm.nih.gov/25560730/>) apart from presenting a measure of discrimination like the AUC, it would be more complete to also evaluate calibration and the recommended approach is calibration plots (again derived through the cross-validation process to avoid overfitting).

Lines 802-804: Similarly to the previous comment, does the DIABLO workflow give an option to do any sort of internal validation to avoid overfitting of the AUCs especially since it is based on sPLS-DA as well?

Discussion

It would be helpful to have a limitations section of the discussion to address the potential impact of aspects of the study such as unequal sex distribution, waist circumference and baseline lipids between the arms of the studies etc.

Reviewer #5

(Remarks to the Author)

Version 3:

Reviewer comments:

Reviewer #3

(Remarks to the Author)

The authors have sufficiently addressed my concerns. One minor suggestion, the authors should avoid the use of "through alterations of gut microbiota". Given that no FMT experiments have been performed, it is strongly suggested to use "associated/related" instead.

Reviewer #4

(Remarks to the Author)

I would like to thank the authors for their responses. I have attached a document with a few additional comments, as we found it helpful to include tracked changes.

Reviewer #5

(Remarks to the Author)

Dear Referees,

Thank you very much for all your valuable feedback. We have addressed all comments and suggestions in a point-by-point response (in blue), and your feedback has significantly improved the outcome of the study and strengthen the manuscript. Thus, we hope that all concerns have been resolved and the paper is suitable for publication in the special issue *Microbiome & Nutrition in Nature Communications*.

Reviewer #1 (Remarks to the Author):

Key results: Please summarise what you consider to be the outstanding features of the work.

Microbial degradation products might be the driving factors for the oat-induced decrease in serum cholesterol levels. Changes in a number of metabolites of lipid and amino acid metabolism might contribute to the cholesterol-lowering effect in the gut microbiome composition and function due to oats.

We greatly appreciate your precise summary of our work. Thank you very much for your efforts.

Validity: Does the manuscript have flaws which should prohibit its publication? If so, please provide details.

The sections of the article do not align with CONSORT guidelines. Adjust sections to the following order Methods, Results, Discussion.

Thank you for pointing this out. The sections of the manuscript have been presented according to the publishing guidelines of *Nature Communications*, i.e. Results, Discussion, Methods. However, we are familiar with the CONSORT guidelines, which we adhere to except for the order of the article sections. We have submitted the CONSORT 2010 checklist with this manuscript and updated it accordingly as part of the revision.

Include a specific and more detailed Conclusion.

Thank you for this suggestion. We have rewritten the conclusion being more specific and detailed (lines 786-801):

“In conclusion, oat consumption, particularly a short-term, high-dose oat diet, provides metabolic health benefits for individuals with MetS by increasing circulating microbially

produced phenolic metabolites, thereby lowering serum cholesterol levels. This identified mechanism shows that phenolic metabolites besides SCFAs are driving factors for the cholesterol-lowering properties of oats and emphasizes that interactions between oats and the gut microbiome play an important role in this health-promoting effect. The mild effects of the six-week oat diet indicate that a more personalized strategy is needed to increase the health benefits of such moderate interventions, since inter-individual differences in response to oats based on person-specific characteristics will be considered. The results offer great potential because oat-based interventions, especially a short-term, high-dose oat diet, are a fast and effective approach to alleviate obesity-related lipid disorders, and they open new avenues for microbiota-targeted nutritional therapies, considering microbially produced phenolic metabolites as potential agents to improve metabolic disorders linked to obesity. However, whether the identified phenolic metabolites from oats contribute to lowering cholesterol levels in a specific pattern needs to be further investigated.”

Originality and significance: If the conclusions are not original, please provide relevant references. On a more subjective note, do you feel that the results presented are of immediate interest to many people in your own discipline, and/or to people from several disciplines?

The conclusions are original but have indistinct links to previously conducted studies. It will be of interest to those working within Metabolic syndrome clinical trials specifically those who have investigated the properties of oats to improve lipid metabolism.

Thank you for your positive assignment. We agree that the results are of great interest to many people working with metabolic syndrome and in nutritional and biomolecular science. Irrespective of this, we have improved our conclusions, highlighting the originality and significance of our own work (lines 786-801). Linking of our results with studies conducted previously is now solely part of the discussion.

Data & methodology: Please comment on the validity of the approach, quality of the data and quality of presentation. Please note that we expect our reviewers to review all data, including any extended data and supplementary information. Is the reporting of data and methodology sufficiently detailed and transparent to enable reproducing the results?

The data analysis is very detailed in both supplementary tables and in the body of the text. Readers may find it hard to identify the more notable significance of the trials main data points due to the dense description of each data point. The lack of description surrounding secondary and tertiary outcomes could subject the trials to being considered of more exploratory nature rather than controlled which is noticeable by the extensive description of the listing of data points. Authors might condense findings from most significant to least significant as opposed to listing all findings and direct readers to supplementary tables for specific but non-notable findings.

Thank you very much for this suggestion. Accordingly, we have adapted the structure of the result section so that the results are now presented depending on the physiological impact and significance, starting with our primary outcome (lines 132-145) and completing with our complex DIABLO models (lines 407-465), which is significant and biologically relevant. Thus, we highlighted the predefined primary outcome of the conducted randomized controlled trial and reduced the description of the secondary outcomes (lines 161-313). Our revision, following your suggestions, hopefully allows the reader to more easily identify the more notable significance of the trials main data points and clearly separate the analysis from the primary and secondary research questions.

Appropriate use of statistics and treatment of uncertainties: All error bars should be defined in the corresponding figure legends; please comment if that's not the case. Please include in your report a specific comment on the appropriateness of any statistical tests, and the accuracy of the description of any error bars and probability values.

Statistical tests are appropriate and have been explained where used but with little justification for why a method was valid for the particular data.

Thank you very much for pointing this out. We apologize that the description of why a method was valid for the particular data was not entirely clear.

The statistical methods employed in this study were carefully selected to match the complexity and nature of the data, ensuring the validity of the analysis. In accordance with the principle of parsimony in statistical modelling, where the simplest adequate explanation is preferred, linear regressions were applied for all data (except the microbiome data), as we were looking for a robust, linear relationship between the intervention and our primary and secondary parameters. This approach was chosen to

avoid unnecessary complexity, which would not improve the analysis's explanatory power, and to avoid artificially inflating the analysis with overly complex models. In case were more than two time points were included in the analysis (follow-up in the short-term study, additional analysis in the six-week study considering all four clinical visits), linear mixed models were used to better account for repeating measurements and a person-specific offset.

All of the variables were checked for their distribution before analysis to justify how to proceed. In case of linear regressions and LMM, residuals were checked for deviation to normality. Variables were log-transformed, if major deviations from the normal distribution were detected. Afterwards the distribution was checked again.

For the complex microbiome data, more advanced techniques, precisely linear models for differential abundance analysis (LinDA, linear mixed model), were applied. This established state-of-the-art method handles the compositional nature of the microbiome data, accounts for zero-inflation, and has high statistical power in detecting differentially abundant taxa¹. The count data used as the input to the model underwent a center log ratio (CLR) transformation during the analysis steps, which is a standardized processing step for microbiome data^{2,3}. In accordance, linear (mixed) models were used to analyze diversity and functional capacity of the gut microbiome. We have rewritten the corresponding section in the manuscript (lines 1052-1106, 1174-1203):

“Gut microbiome analysis

*[...] A LMM was applied to assess the differences [in alpha diversity] between the two diet groups in each study. This statistical method was used to address the natural complexity of microbiome data. [...] The Person-ID was used as random effect to account for repeating measurements and a person-specific offset. The model was defined as: $diversity\ matrix \sim time*group + age + sex + BMI + (1|Person-ID)$, with the covariate of interest ‘time*group’. [...] To determine the differences [in beta diversity] between the diet groups in each study, a linear regression analysis adjusted for confounding factors was applied, using the control group of each study (CG or CG^{6w}) as the reference (LnR^{adj}). As the distance metric for beta diversity is relative to the offset (baseline), the random effect was not applied. The model was defined as: $diversity\ matrix \sim group + age + sex + BMI$, with the covariate of interest ‘group’.*

To identify differences between the diet groups of the RCTs at feature level, linear models for differential abundance analysis (LinDA)¹, an established state-of-the-art method for analyzing microbiome data, was applied. [...] The model was defined as: $\text{taxon} \sim \text{time} * \text{group} + \text{age} + \text{sex} + \text{BMI} + (1|\text{Person-ID})$, with the covariate of interest 'time*group'. [...] Data were normalized using a centered-log-ratio (CLR) transformation with an offset of 1 prior to all analyses. [...] In accordance with LinDA, a LMM was applied to the CLR-transformed count data with the control group of each study (CG or CG^{6w}) and the baseline value as reference, and the Person-ID as random effect (LMM^{adj.}). The model was defined as: $\text{pathway} \sim \text{time} * \text{group} + \text{age} + \text{sex} + \text{BMI} + (1|\text{Person ID})$, with the covariate of interest 'time*group'..”

“Statistics and sample size calculation

[...] Prior to the analysis, the distribution of the data was visualized individually. Additionally, Shapiro-Wilk tests were performed to test for normality of the data and determine the appropriate approach for further analysis. Variables were analyzed either on their original scale or logarithmic transformed to approximate a normal distribution. [...]

To assess whether the data provided evidence of the superiority of the oat diets over the respective control in terms of our primary outcomes and the metabolic and metabolomic outcomes, linear regression models were applied [...]. The model was defined as: “ $\text{post-intervention_value} \sim \text{baseline_value} + \text{group} + \text{age} + \text{sex} + \text{BMI}$ ”. [...] In addition, a LMM was conducted adjusted for BMI, age, and sex (LMM^{adj.}) to assess differences in clinical markers across all four time points of the six-week intervention study. The control group of each study (CG or CG^{6w}) and the baseline value were used as reference, the Person-ID was used as random effect to account for repeating measurements. The model was defined as: $\text{clinical marker} \sim \text{time} * \text{group} + \text{age} + \text{sex} + \text{BMI} + (1|\text{Person-ID})$, with the covariate of interest 'time*group'. Residuals obtained from the models were inspected for normality to control for the fit of the statistical model. If the residuals were not normally distributed, logarithmic transformation was applied.”

Conclusions: Do you find that the conclusions and data interpretation are robust, valid and reliable?

The conclusions are valid based on the data analysis presented in this paper.

Thank you very much for your positive feedback.

Suggested improvements: Please list additional experiments or data that could help strengthening the work in a revision.

References: Does this manuscript reference previous literature appropriately? If not, what references should be included or excluded?

Appropriate references have been included.

Thank you very much for your overall positive assessment and that the literature we referenced is appropriate.

Clarity and context: Is the abstract clear, accessible? Are abstract, introduction and conclusions appropriate?

The abstract is clear however the statistics has been reduced to a percentage so could be more detailed.

Thank you very much for pointing this out. Due to the limited number of words in the abstract, we had to prioritized and highlighted the most important aspects, focusing on the results, as they provide the key insights from the study. In the main text, we have revised the statistics section accordingly and added more details about why a method was valid for the particular data (lines 1052-1107, 1161-1215).

Inflammatory material: Does the manuscript contain any language that is inappropriate or potentially libelous?

The manuscript does not contain any language that is inappropriate or potentially libelous.

Thank you very much for your overall positive feedback on the language of the manuscript.

Reviewer #2 (Remarks to the Author):

This interesting study discusses the cholesterol-lowering effects of oats induced by microbially produced phenolic metabolites in individuals with metabolic syndrome through dietary interventions and multi-omics analysis.

The message is that short term high oat diet induces pronounced effects on microbial composition and metabolism, whilst a 6 wk moderate oat diet has very slight effects. This overall message has to be emphasized and discussed more.

Thank you very much for your overall positive feedback and for highlighting the different outcomes of the two oat interventions. As suggested, we have discussed the overall message in more detail (lines 694-718):

“The attenuated effect of the six-week oat intervention on plasma phenolic metabolite levels may be attributed to the lower oat consumption (80 g/day) when integrating a single oat meal into the habitual diet, compared to the short-term high-dose oat diet (300 g/day) where three oat meals replaced the habitual diet entirely. The single meal leads to a relatively lower intake of oat bioactive compounds, including phenols, which may impact their biological effects such as their hypolipidemic activity³⁸. In addition, maintaining the habitual Western dietary pattern alongside a single oat meal introduced greater variability in foods and nutrient exposure, potentially covering intervention efficacy. We therefore assume that the impact of a single oat meal may not be strong enough to compensate for the inter-individual differences in habitual food intake during the study period. Heterogeneity of oat meal preparation²² as recommended in the OG6w in combination with inter-individual differences in the response to oats based on the gut microbiome and the host genetic phenotype may also contribute to the moderate effect in the six-week intervention²³. These observations suggest that personalized nutrition strategies may optimize oat interventions at moderate doses, where individual parameters seem to exert greater influence compared to a high-dose protocol. Despite the high individual variability, we could explain almost 20% of the variance in LDL-cholesterol in the short-term study solely by the changes in the plasma phenolic compounds. Furthermore, it is worth noting that the participants in the short-term intervention study additionally underwent calorie restriction, while the dietary intervention in the six-week study was isocaloric. The observed differences in the health effects of the two oat interventions may therefore be explained by different

underlying mechanisms such as calorie restriction⁴⁸, dose-dependent exposure to the bioactive compounds and potential synergistic effects.”

Diet composition at product/food items level has to be reported more extensively. When you add 100 g oat to one meal 3 times a day in the short term, how does that change overall diet composition and to what extent can differences be ascribed to oat?

Thank you for raising this question. We apologize that the description was not entirely clear and led to misunderstandings. Now we provided a concise and precise description of the intervention diets: During the short-term oat intervention, the participants consumed three oat meals per day instead of their habitual diet. Each meal contained 100 g oat flakes and have been consumed as porridge prepared with 300-500 mL water. In the control group, the participants consumed three macronutrient-adapted control meals without oats (carbohydrates: 67 E%, protein: 15 E%, fat: 17 E%) per day instead of their habitual diet. The habitual diet of the participants was assessed using an FFQ and participants were screened according to a Western dietary pattern with low oat consumption (carbohydrates: 38 E%, protein: 15 E%, fat: 43 E%). Thus, the short-term oat and control diet induced a change in the overall diet composition (switch from low-carb/high-fat to high-carb/low-fat). The observed differences are therefore, as we have showed here in this study, ascribable to oat consumption. For clarification, we have rewritten the corresponding section in the manuscript (lines 880-892):

“During the short-term intervention period, the participants followed a two-day, high-dose oat diet or a macronutrient-adapted control diet, both hypocaloric and high in fiber (1100-1200 kcal/d; carbohydrates (67 energy% (E%), of which fiber > 15%, β -glucan: 13.5 g vs. 0 g), protein (15 E%), and fat (17 E%)), depending on their allocation (OG vs. CG). Participants in the OG consumed three oat meals per day instead of their habitual diet. Each meal consisted of 100 g rolled oat flakes (Demeterhof Schwab GmbH & Co. KG, Windsbach, Germany) and has been consumed as porridge, prepared with water. Fruits restricted to apples, pears, and berries and vegetables restricted to spinach and leeks were used as additives and added to the meal plans of the CG in the same amount. No salt, sugar, or sweeteners were added. Participants in the CG consumed two meals per day comprising bread and raw vegetables (breakfast and dinner) and one warm meal per day (lunch) instead of their habitual diet.”

What are the clinical implications of the current findings, how long do effects of high oat last and what would be the advise, to consume consistently a high oat diet or intermittently?

Thank you very much for bringing this up. Based on our study results, a short-term high-dose oat intervention may lead to an improvement in cholesterol levels within a relevant clinical range (total cholesterol: -8%, LDL-cholesterol: -10%). Comparing these results of a short-term dietary intervention with the mean effect size of e.g. statins, which depending on the specific product and dose (5-80 mg) cause an average relative reduction in LDL-cholesterol of 15% to 58%⁶⁴, the clinical implications should be considered relevant. Additionally, the results of the 6-week follow-up phase of the short-term intervention study show that cholesterol levels tend to remain below the baseline value. Therefore, the advice would be to consume a high-dose oat diet intermittently. However, from a physiological nutritional perspective any form of long-term imbalanced diet is not recommended.

We have added a corresponding section in the revised manuscript (lines 767-773):

“Compared to the mean effect size of cholesterol-lowering drugs, e.g., statins, which cause an average relative reduction in LDL-C of 15% to 58%, depending on the specific product, dose (5-80 mg) and the continuous intake⁶⁴, the clinical implications of the oat diet can be considered as relevant. Especially since cholesterol levels tended to remain below the baseline value during the 6-week follow-up phase of the short-term intervention study, it would be advisable to consume a high-dose oat diet intermittently.”

Statistical analyses like sPLS-DA and DIABLO were used for data interpretation. Many correlations were reported. Is it also possible to indicate the predictive value of microbial and phenolic changes in modulating cholesterol?

Thank you very much for raising this question. We have performed additional analyses for indicating the predictive value of the microbial and phenolic changes in modulating cholesterol and included corresponding sections in the manuscript. The results indicated that the oat-induced changes in gut microbiota and plasma phenolic compounds play a valuable role in modulating cholesterol levels and may be additional mechanisms to those already known.

We have added corresponding sections in the manuscript:

Methods (lines 1149-1159):

“Partial least square regression model

To indicate the predictive value of the microbial and phenolic changes in modulating cholesterol levels, partial least square (PLS) regression models were applied. The models were implemented using the mixOmics package in R (version 4.2.2). The analyses were performed without sparse mode, i.e. all predictor variables identified in the previous DIABLO models (cutoff $r = 0.7$) were included in the models without variable selection. R^2 and Q^2 total (cross-validated R^2), calculated by leave-one-out cross-validation, were used to assess the fit and predictability of the model, respectively. The log fold changes of the identified features (nine phenolic compounds or two genera and one pathway) were used as predictors and that of total and LDL-cholesterol as response.”

Results (lines 418-422, 431-435):

“Cholesterol reduction by oats linked to increase in phenols

[...] In addition, PLS regression models revealed that the change in plasma phenolic compounds solely predicted 13.6% of the variation in cholesterol levels ($Q^2 = 0.136$) and explained 13.5% of the variance in TC ($R^2 = 0.135$) and 19.3% of the variance in LDL-C ($R^2 = 0.193$). Thus, our results suggest that phenolic compounds play an important role in the cholesterol-lowering effect of oats.

[...] In addition, these changes in the gut microbiome solely predicted 13.5% of the variation in cholesterol levels ($Q^2 = 0.135$) and explained 11.1% of the variance in TC ($R^2 = 0.111$) and 19.6% of the variance in LDL-C ($R^2 = 0.196$). This supports our assumption that alterations in microbial composition and functional capacity may contribute to lowering cholesterol levels.”

Is it possible to add a figure with major outcomes both short-term and long-term?

Thank you for this suggestion. We have included a corresponding figure with the major outcomes of both short-term and 6-wk intervention study (Fig. 9).

Reviewer #3 (Remarks to the Author):

Linda et al conducted two randomized clinical trial in subjects with MetS, and demonstrated that gut microbiota-derived phenolic metabolites mediate the cholesterol-lowering effect of oats. The study is interesting and of significance, however, several major concerns preclude the publication.

Thank you very much for your overall positive evaluation.

Major

1. Overall, this study is entirely based on bioinformatics analysis and all the data are correlative in nature.

Thank you. Indeed, we present data correlative in nature (e.g., DIABLO, Spearman correlation) based on two randomized, controlled clinical intervention studies (RCTs), which are the gold standard for the investigation of causal relationships in evidence-based-medicine (cause-effect principle) and thus for the investigation of nutritional effects *in vivo*⁵. Since we followed the international ethical and scientific quality standards of clinical research practices (Good Clinical Practice (GCP)) and the consolidated standards of reporting trials (CONSORT guidelines), we achieved the highest level of evidence^{6,7}. The correlation analysis performed provides additional insight into the identified intervention effects and enables the uncovering of associations between oat-induced alterations in metabolism, gut microbiota and global metabolomic profiles. In this way, possible underlying mechanisms have been identified and new targeted hypotheses have been generated and tested *in vitro*.

Given that 16S rRNA seq is not suitable for microbial function analysis, I don't this the data, especially for microbial metabolites, provided in the manuscript is convincing.

Thank you very much for pointing this out. We appreciate your insightful feedback regarding the limitations of using 16S rRNA sequencing for microbial functional analysis. We recognize that while 16S rRNA sequencing has its constraints, it can still yield valuable insights into functional potential, particularly when combined with tools like PICRUSt2 for predicting microbial pathways⁸⁻¹⁰.

We are aware that the accuracy of PICRUSt2 may fluctuate depending on the complexity of the microbial community and the availability of reference data. However, it has been demonstrated that PICRUSt2 provide reasonably accurate predictions of functional potential, especially for well-characterized microbial communities. As noted

by Douglas et al. (2020), PICRUSt2 achieves an accuracy ranging from 79% to 88% when compared to shotgun whole genome sequencing approaches⁸. Moreover, there are several publications that have successfully utilized 16S rRNA data in conjunction with PICRUSt2 to explore microbial metabolites. For instance, Wang et al. (2022) demonstrated this approach in their study of alfalfa silage⁹, and Li et al. (2024) recently explored functional metabolism in drinking water using similar methods¹⁰.

However, for clarification and to address this aspect, as it is also mentioned in minor comment point 4, we have revised our manuscript (lines 1089-1095) and emphasized the robustness of our findings, supported by both the integration of multiple methodologies and relevant literature. This enhanced discussion improves the manuscript and addresses the reviewer's concerns regarding the use of 16S rRNA sequencing and its interpretation in the context of microbial metabolites:

“Such functional prediction tools play a valuable role in facilitating preliminary hypothesis testing based on amplicon sequences; however, this method has inherent limitations in accurately reproducing functional potential. It should be acknowledged that the prediction model based on 16S or shotgun sequencing data does not confirm whether functional capacity is activated at the mRNA or protein level. Given the prohibitive cost of shotgun sequencing, functional prediction tools serve as a valid guide for subsequent studies of actual functional capacity.”

Moreover, no *in vitro* or *in vivo* experiments have been carried out to validate or prove that phenolic compounds mediate the beneficial effect of oat on cholesterol metabolism.

Thanks for pointing this out. It has been shown in *in vitro* and *in vivo* experiments that ferulic acid, as a single compound, improved cholesterol metabolism by various mechanisms, including reducing the activity of HMG-CoA reductase, the rate-limiting enzyme in cholesterol synthesis¹¹, and restoring mitochondrial dynamics and autophagy in hepatocytes via the AMPK signaling pathway¹². The protective effects of ferulic acid on metabolic syndrome, including hyperlipidemia, was recently summarized by Ye et al.¹³. In addition, a mixture of ferulic acid esters (oryzanol) has been demonstrated to have anti-hyperlipidemic effects in hamster¹⁴. However, a potential cholesterol-lowering effect of dihydroferulic acid (DHFA), which has been identified as the primary microbial metabolite of ferulic acid¹⁵, as well as of other

(microbially produced) phenolic metabolites derived from oats has not yet been investigated, particularly not in humans. This emphasizes the novelty of our results, showing an inverse correlation between the changes in circulating cholesterol levels and plasma concentration of microbial phenolic compounds, which increased in the oat group compared to baseline and the control group (causal relationship).

Nevertheless, in order to address your suggestion and to close this knowledge gap regarding the microbial produced phenolic metabolites, we have performed two additional *in vitro* cell culture experiments:

1. Our results thus confirm that microbially produced phenolic compounds such as DHFA may mediate the oats' cholesterol-lowering effects.

We have added corresponding sections with detailed descriptions in the manuscript and the Supplementary Information:

Methods (lines 979-991):

“In vitro cell culture experiments

To confirm that the identified phenolic compounds mediate the beneficial effect of oats on cholesterol metabolism, we performed in vitro cell culture experiments. Briefly, blood was collected from participants with MetS (n = 4; 65 ± 9 y, 29.3 ± 2.2 kg/m²) and MHCs (n = 4; 33 ± 11 y, 25.1 ± 4.9 kg/m²) to isolate peripheral blood mononuclear cells (PBMCs). After pre-processing, PBMCs were exposed to medium with or without 10 μM DHFA (Sigma-Aldrich, #17803-5G) that additionally contained 20 μg/mL alkyne cholesterol. In addition, HuH7 cells (JCRB0403) were treated with the same medium with or without 10 μM DHFA after 48 h of cultivation in growth medium. The medium was additionally supplemented either with 20 μg/mL alkyne cholesterol or 5 mM 1,2-¹³C₂ sodium acetate (CIL, #CLM-440-1) and 100 μM alkyne fatty acid 182. After 24 h of incubation, the samples were subjected to lipid extraction and dissolved lipids were analyzed by mass spectrometry (Supplementary Methods).”

Results (lines 507-527):

“DHFA altered cholesterol metabolism in vitro

In PBMCs fed with alkyne cholesterol, DHFA led to a significant decrease in the molar fraction of alkyne cholesterol (-15.71 ± 6.33 mol% (mean ± SEM), P = 0.042), while total lipidome remained stable (Fig. 8a). In addition, in the metabolically healthy

controls (MHCs), the molar fraction of alkyne cholesterol esters tended to be lower in the PBMCs treated with DHFA (-5.54 ± 1.94 mol%, $P = 0.065$; Fig. 8a). These results support that microbially produced phenolic compounds such as DHFA mediate the oats' cholesterol-lowering effects by reducing alkyne cholesterol and cholesterol esters incorporation into PBMCs relative to the total lipidome.

In HuH7 cells fed with ^{13}C labeled acetate and alkyne fatty acid 182, DHFA seemed to decrease the molar fraction of ^{13}C labeled-alkyne cholesterol esters and alkyne cholesterol esters of the total lipidome, while the total lipidome itself increased by trend (Fig. 8b). Consistent with this, a decrease in the molar fraction of the unlabeled cholesterol esters tended to take place (Supplementary Figure 5a). No differences in total lipidome, the molar fraction of alkyne cholesterol esters and alkyne cholesterol were observed between HuH7 cells either treated with or without DHFA and additionally fed with alkyne cholesterol (Supplementary Figure 5b). Our results indicate that DHFA might impact different effects i) the totality of lipids, ii) the *de novo* synthesis of cholesterol esters and iii) the esterification of cholesterol in HuH7 cells, supporting that phenolic compounds such as DHFA might contribute to the cholesterol-lowering properties of oats.”

2. Moreover, we conducted an anaerobic *in vitro* fecal batch culture experiment demonstrating that the gut microbes produced a variety of phenolic metabolites such as 2-aminophenol and DHFA through the interaction with oats and confirming the observed increase of phenolic components in our RCTs.

We have added corresponding sections with detailed descriptions in the manuscript and the Supplementary Information:

Methods (lines 993-1003):

“Fecal batch culture fermentation of *in vitro* digested oats

To evaluate if the metabolism of the microbiome results in the production of the identified oat-derived metabolites, in particular the phenolic compounds, we conducted an anaerobic *in vitro* fecal batch culture experiment. In brief, aliquots of homogenized baseline stool samples from four representative participants of the short-term intervention study were used to inoculate sterile anoxic basal medium either with or without *in vitro* digested oats in a starving group (anoxic basal buffer; SOG vs. SCG) or a physiological group (Brain Heart Infusion medium; POG vs. PCG). Samples for

global metabolomic profiling including phenolic compounds, short-chain fatty acids (SCFAs) and microbiome analysis were taken directly after inoculation, after 6 h, 24 h, and 36 h of incubation time (Supplementary Methods).”

Results (lines 530-556):

“Microbial oat metabolization produced phenolics in vitro

To prove whether the identified phenolic compounds stem from the microbial metabolization of oats, anaerobic fecal batch culture fermentations of in vitro digested oat flakes were conducted. This demonstrated that the gut microbes are capable of producing a variety of metabolites including phenolic compounds by directly interacting with the provided oats. In particular, a significant increase in 2-aminophenol was shown in both oat groups (physiological oat group (POG) vs. physiological control group (PCG): $P < 0.001$ (6 h, 36 h), $P < 0.01$ (24 h), starving oat group (SOG) vs. starving control group (SCG): $P < 0.01$ (24 h, 36 h); Fig. 8c), which is in line with the observed increase in the plasma level of 2-aminophenol sulfate in our RCTs. In addition, a rapid production of microbial products such as DHFA and vanillic acid were observed in both oat groups by a simultaneous reduction of phenolic compounds such as FA and vanillin mainly within 6 h (POG vs. PCG: $P < 0.01$ (DHFA, FA, vanillin), $P < 0.05$ (vanillic acid); SOG vs. SCG: $P < 0.05$ (DHFA, FA, vanillin); Supplementary Figure 5c), emphasizing that the microbial activity results in the production of phenolic oat-derived metabolites. Moreover, as expected, shifts in the concentrations of various SCFAs, in particular an increase in acetic acid (POG vs. PCG: all $P < 0.05$) and succinate (SOG vs. SCG: $P = 0.033$ (36 h)) and a decrease in isobutyric acid (POG vs. PCG: $P = 0.002$ (24 h), SOG vs. SCG: $P = 0.001$ (6 h), $P = 0.047$ (36 h); Fig. 8c), were found. Furthermore, significantly higher abundances of known SCFA-producing bacteria such as *Faecalibacterium*, *Fusicatenibacter*, *Bifidobacterium*, *Blautia*, and *Roseburia* were observed in POG compared with PCG, especially after 36 h (all $P < 0.05$; Supplementary Figure 5d). In addition, *Erysipelotrichaceae* UCG-003 seemed to increase in POG compared to PCG after 36h of incubation ($P = 0.172$; Fig. 8d), supporting that the identified increase in the abundance of *Erysipelotrichaceae* UCG-003 in our RCT is induced by the short-term, high-dose oat intake.”

2. As is described by the authors, SCFAs are also important microbial products of oats. Then, do the authors detect SCFAs in plasma and fecal samples before and after the intervention > If yes, how about its change? Given the established beneficial effect of SCFAs on cholesterol metabolism, how can you make sure that the modulating effect is exerted by phenolic compounds without detecting or adjusting for the effect of SCFAs?

Thank you very much for bringing this up. We detected SCFAs before and after the intervention and observed differences between the two diet groups of the short-term intervention study as expected, in particular higher plasma butyric acid levels in the oat group compared with the control group, which is in line with the results of our *in vitro* batch culture experiment and the current literature^{16–18}. Since the beneficial effect of SCFAs on cholesterol metabolism has already been shown^{19–22} and we aimed to identify new metabolites beside SCFAs, we decided to focus on further microbial produced metabolites derived from oats. Our analysis revealed that the phenolic compounds are also of great importance of the cholesterol-lowering effect of oats, in both *in vivo* (RCT) and *in vitro*.

Nevertheless, to make sure that the modulating effect is exerted by phenolic compounds as suggested, we performed additional DIABLO models in which we included the plasma SCFAs as a separate data set. We found basically the same associations as in the previous models, which are currently reported in our manuscript. In particular, several inverse correlations between the changes in cholesterol levels and phenolic compounds were observed, confirming the important role of the phenolic compounds in modulating circulating cholesterol levels.

3. The design and description of the two trials are wired. For the short-term intervention, I am not sure the change in phenolic compounds, gut microbiota, and clinical data refers to day 2 post intervention or the lasting effect by week 6 post intervention?

Thank you for your comment. We apologize that the description of our two trials was not entirely clear and led to misunderstandings. The observed changes in phenolic compounds, gut microbiota, and clinical data refer to the visit taking place immediately after the two days of intervention (V2). Additionally, the lasting effects by week 6 post intervention (V5) were reported for significantly changed clinical markers (Fig. 3d) and

microbial genera (Fig. 6c). These lasting effects were referred to as the “follow-up period”.

In addition to the short-term intervention (2-days of high-dose oat intervention) with a 6-week follow-up, we performed in our second trail a 6-week moderate oat intervention and assessed the effects immediately after the six weeks.

For clarification we have rewritten the study design (lines 821-848) and have been more specific about which studies and visits we are referring to in the results section (lines 64-313):

“Study design

“[...] In the short-term intervention study, eligible participants were invited for two clinical visits, the first before (V1) and the second directly after the two-day intervention period (V2), followed by three clinic visits at two-week intervals during a six-week follow-up period (V3 – V5) to determine potential long-term effects of the oat diet (Fig. 1a). During the six-weeks, intervention study, eligible participants were invited for four clinical visits at two-weeks intervals with the first before (V1) and the last after the six-week intervention period (V4) (Fig. 1b).

After the first visit (V1), participants were randomly assigned to the experimental group (OG or OG^{6w}) or the control group (CG or CG^{6w}) of each study using computer-generated randomization tables in a block format with variable block length generated by a researcher not clinically involved in the study. [...]

At V1 and the clinical visit directly after each intervention period (V2 or V4), anthropometric data, fecal samples as well as fasting and postprandial blood samples taken during an oral glucose tolerance test (OGTT) were collected. [...] At the clinical visits during the six-week intervention study (V2+V3), anthropometric and BP measurements were performed and fasting blood and fecal samples were collected. These also took place at the clinical visits during the follow-up period of the short-term intervention study (V3 – V5). [...]”

4. When comparing the change of certain metabolites or clinical parameter between two intervention groups, the baseline level of this metabolite or clinical parameter should also be included in the model.

Thank you very much for pointing this out. We agree that it is important to include baseline values in the analysis of longitudinal data. We apologize if our data analysis approach was not entirely clear. All of the linear regression models included the baseline level as a covariate. In case of linear mixed models, the baseline value was used as a reference. We have revised the corresponding sections to emphasize this aspect (lines 1059-1106, 1183-1209).

For the discriminant analyses, we focused on the fold-change of the variables. These methods do not specifically account for baseline values, as the functions used do not allow for additional covariates. However, recognizing the importance of including baseline values to avoid potential bias, we conducted linear regression models after the discriminant analysis (adjusting for the baseline levels) to ensure consistency. The results were consistent with the discriminant analyses, which is why we did not place additional emphasis on this point. Detailed results can be found in the lines 161-313.

“Gut microbiome analysis

*[...] The LMM was adjusted for confounding factors and used the corresponding control group (CG or CG^{6w}) and the baseline value as reference (LMM^{adj.}). [...] The model was defined as: [alpha] diversity matrix ~ time*group + age + sex + BMI + (1|Person-ID), with the covariate of interest time*group. [...] To determine the differences between the diet groups in each study, a linear regression analysis adjusted for confounding factors was applied, using the control group of each study (CG or CG^{6w}) as the reference (LnR^{adj.}). [...] The model was defined as: [beta] diversity matrix ~ group + age + sex + BMI, with the covariate of interest group.*

*To identify differences between the diet groups of the RCTs at feature level, linear models for differential abundance analysis (LinDA)¹, an established state-of-the-art method for analyzing microbiome data, was applied. The control group of each study (CG or CG^{6w}) and the baseline value were used as reference and the Person-ID was used as random effect. The model was defined as: taxon ~ time*group + age + sex + BMI + (1|Person-ID), with the covariate of interest time*group.*

[...] In accordance with LinDA, a LMM was applied to the CLR-transformed count data with the control group of each study (CG or CG^{6w}) and the baseline value as reference

and the Person-ID as random effect (LMM^{adj.}). The model was defined as: $\text{pathway} \sim \text{time} * \text{group} + \text{age} + \text{sex} + \text{BMI} + (1 | \text{Person ID})$, with the covariate of interest 'time*group'."

"Statistics and sample size calculation

[...] To assess whether the data provided evidence of the superiority of the oat diets over the respective control in terms of our primary outcomes and the metabolic and metabolomic outcomes, linear regression models were applied including the baseline value and the confounding factors BMI, age, and sex (LnR^{adj.}) as covariates. The control group of each study (CG or CG^{6w}) was used as reference. The model was defined as: "post-intervention_value ~ baseline_value + group + age + sex + BMI". [...] In addition, a LMM was conducted adjusted for BMI, age, and sex (LMM^{adj.}) to assess differences in clinical markers across all four time points of the six-week intervention study. The control group of each study (CG or CG^{6w}) and the baseline value were used as reference, the Person-ID was used as random effect to account for repeating measurements. The model was defined as: $\text{clinical marker} \sim \text{time} * \text{group} + \text{age} + \text{sex} + \text{BMI} + (1 | \text{Person-ID})$, with the covariate of interest time*group. [...] In addition, to account for the changes within the OG including the follow-up period, a LMM adjusted for BMI, age, and sex (LMM^{adj.}) was performed using the baseline value as reference and the Person-ID as random effect."

Minor

1. Why the authors restrict the age of study participants to 45-70 yrs?

Thank you for raising this question. We restricted the age of the study participants to 45-70 years for several reasons. Firstly, the prevalence of metabolic disorders such as metabolic syndrome increases significantly with age^{23,24}, making it important to study participants in the age range in which these diseases occur most frequently. Secondly, it is important to maintain an appropriate balance between health benefit and risk. People between the ages of 45 and 70 are more likely to develop metabolic syndrome and additionally have a relatively stable health, thus maximizing the benefits and minimizing the health risks. In contrast, older people (70+) are more vulnerable and may suffer from multiple diseases or frailty, which could complicate the interpretation of the results due to various confounding factors such as different chronic diseases and medications. Thirdly, the 45- to 70-year-old age group is considered more

representative of the general adult population, which enables better generalizability of the results. The results are often more applicable to clinical practice for middle-aged and older adults, allowing to examine the safety and efficacy of dietary interventions in a group that is likely to use them. By focusing on the 45-70 age group, we maximize the benefit for a population at high risk of the disease of interest, namely metabolic syndrome, while minimizing safety risks and confounding variables that could distort the study results (benefit-risk assessment according to principle 4 of the Good Clinical Research Practice (GCP)⁶).

2. Why history of smoking or alcoholism is excluded? Ex-smokers are also excluded? The exclusion criteria seem to be too strict, which would substantially affect the applicability of this study.

Thank you for raising these questions. Smoking and alcoholism, both of which have a major impact on metabolism^{25,26}, nutritional status^{27,28}, and microbiome^{29,30}, were excluded in order to reduce confounding factors. In nutritional science, we have to deal with many confounding factors that may potentially mask rather moderate dietary effects, as we do not expect pharmacological effect sizes. However, ex-smokers were not excluded. With this, we aimed to ensure that the results of our randomized controlled studies are generalized to the broader population. We therefore adhered to the good clinical practice (GCP) by defining sample size as well as inclusion and exclusion criteria and by performing a benefit-risk assessment to ensure the international ethical and scientific quality for designing, conducting, recording and reporting human studies⁶.

3. For both untargeted and targeted metabolomics, more details should be provided. How plasma and fecal samples are extracted. What's the detailed setup of the LC-MS/MS system? The flow rate, the column used and how features or peaks are identified (including the database used) should be specified to ensure reproducibility. Thank you for bringing this up. Both the untargeted and targeted metabolomics were provided by Metabolon Inc. (Morrisville, NC). For the untargeted analysis, Metabolon used their well-established global metabolomics platform (Global Discovery Panel), which has been referenced in over 800 publications to date³¹. A detailed description of the method was provided by Ford et al.³² to which we referred in our manuscript. In addition, an extensive description of the method that ensure reproducibility can be

found in Benedetti et al.³³ Therefore, we have included a corresponding reference to this publication (lines 1014-1016):

“For both fasting plasma and fecal samples, non-targeted global metabolomic profiles before and after each intervention period were generated by Metabolon Inc. (Research Triangle) using UPLC-MS/MS, as previously described^{32,33}.”

To ensure reproducibility of the targeted metabolomics, we have included a detailed description of the LC-MS/MS system setup based on the information provided by Metabolon (Supplementary Methods page 3):

“Human plasma samples were spiked with stable labeled internal standards and incubated with glucuronidase and sulfatase enzymes to convert conjugated metabolites to their free form. After incubation, the samples were subjected to protein precipitation with acetonitrile. Next, the samples were centrifuged, and the supernatant dried down under nitrogen and reconstituted with water containing formic acid. The samples were mixed and injected onto an Agilent 1290/AB Sciex QTRAP 7500 LC MS/MS system equipped with a BEH C18 reversed phase UHPLC column. The flow rate was 625 μ L/min. The mass spectrometer was operated in negative mode using electrospray ionization (ESI). The peak areas of the respective analyte product ions were measured against the peak area of the corresponding internal standard product ions (e.g., dihydroferulic acid, avenanthramides). Quantitation was performed using a linear regression with a 1/x weighting generated from fortified calibration standards prepared immediately prior to each run. LC-MS/MS raw data were collected and processed using AB SCIEX software Analyst 1.6.3 and processed using SCIEX OS-MQ software v1.7. Data reduction was performed using Microsoft Excel for Office 365 v.16.”

4. Prediction of microbial genes using 16s rRNA seq data is not accurate enough. The authors should at least acknowledge this limitation clearly in the manuscript.

Thank you for your constructive feedback. We appreciate your comment regarding the accuracy of predicting microbial genes using 16S rRNA sequencing data. We acknowledge that this method has inherent limitations in accurately reflecting functional potential. In our revised manuscript, we have clarified these points and

emphasized both the robustness and limitations of using functional predictions based on a 16S dataset (lines 1089-1095, see major point 1.2). However, it is important to note that the same limitation applies to shotgun sequencing; while it may provide more complete functional information about the bacterial community, it merely predicts the functional capacity of the microbiome. This prediction does not confirm whether the functional capacity is activated at the mRNA or protein level. Given the prohibitive costs associated with shotgun sequencing, we believe that functional prediction tools like PICRUSt and Tax4Fun serve a valuable role in facilitating preliminary hypothesis testing based on amplicon sequences. These tools allow us to explore the functional capabilities of microbial communities, guiding subsequent investigations on the actual functional capacity. We trust that our revisions adequately address this important limitation, and we appreciate your suggestion to clarify this aspect in our manuscript.

5. Why do the authors choose different methods for multiple correction in metabolomics and microbiota analysis?

Thank you for raising this question. We used the Benjamini-Hochberg as a method of controlling the false-discovery rate (FDR) for microbiome and metabolomic data, introduced by Benjamini and Hochberg in 1995³⁴. The FDR is especially suitable for the screening of many, potentially interesting biomarkers and is frequently used for high-throughput data in biology (e.g., in ^{35–38}). In testing the effects of oat-based diets on metabolic markers (clinical markers), we opted for a more stringent multiple comparison method for one primary reason: associations between oat-based diets and markers such as cholesterol^{39,40} and postprandial glucose^{41,42} levels have already been well-documented in literature. Therefore, our analysis was hypothesis-driven, focusing on verifying previously established relationships rather than exploratory discovery. We have included these explanations in the manuscript (lines 1166-1172):

“For metabolic data analysis, family-wise error rate correction was applied via the block-wise Bonferroni-Holm method⁸⁰ to correct for multiple testing ($P^{adj.} < 0.05$). This stringent multiple comparison method is particularly suitable for hypothesis-driven analyses as in the present case. For the microbiome and global metabolomic data with a high noise-to-signal ratio, FDR correction based on the Benjamini/Hochberg method⁸¹ was applied ($q < 0.05$), which is frequently used for high-throughput data.”

Dear Referees,

Thank you very much for all your valuable feedback. We have addressed all comments and suggestions in a point-by-point response (in blue), and your feedback has improved our manuscript even more. Thus, we hope that all concerns have been resolved and the manuscript is now suitable for publication in the special issue *Microbiome & Nutrition in Nature Communications*.

Reviewer #2 (Remarks to the Author):

The authors have addressed my comments sufficiently.

Thank you very much for your positive evaluation and for confirming that our revisions have sufficiently addressed your previous comments.

Reviewer #3 (Remarks to the Author):

The authors have addressed most of my concerns and the manuscript has been substantially improved. However, several minor issues need to be addressed before being accepted.

Thank you very much for your overall positive assessment.

1. Although the study is RCT in nature and follows good clinical research practice, I don't think you can claim that this is the highest-level evidence. It should be noted that only multi-center RCT with a large sample size can be regarded as a high level. The authors should clearly acknowledge this limitation.

Thank you for your comment. We agree that a multi-center RCT with a large sample size can be regarded as the highest level of evidence. As suggested, we have acknowledged this limitation in the manuscript (lines 496-499):

“To confirm our results based on an RCT, further studies are needed. For example, a multicenter RCT with a sample size calculation may provide the next level of evidence in clinical research since it would allow better generalizability across different populations and reduce a potential center-specific bias.”

2. With a small sample size and strict inclusion criteria, the generalizability of this finding is inevitably influenced. Therefore, the authors should be more careful when drawing the conclusion and clearly acknowledge this limitation.

Thank you for pointing this out. We now have been more careful when drawing the conclusion and acknowledged this limitation:

Strengths and limitations (line 496-499): “To confirm our results based on an RCT, further studies are needed. For example, a multicenter RCT with a sample size calculation may provide the next level of evidence in clinical research since it would allow better generalizability across different populations and reduce a potential center-specific bias.”

Conclusion (lines 515-517): “However, whether the identified phenolic metabolites from oats contribute to lowering cholesterol levels in a specific pattern in general and across different populations needs to be further investigated.”

Reviewer #4 (Remarks to the Author):

The study by Klümpen et al goes deeper into examining the potential pathways related to the gut microbiome underlying a more established association of oats consumption with lower cholesterol levels among people with metabolic syndrome (MetS). They have conducted two well-designed parallel randomised controlled trials to look at both the short-term effects of a larger dose and the effects of a more consistent lower dose on several markers related to MetS, but also metabolic biomarkers including microbiotic end products. The thorough study design and statistical analyses, as well as minor deviations from the protocol despite the challenges during the COVID-19 pandemic are commendable. The authors have made the most of their data by integrating multiple sources of data to show how these may interact. The combination of two RCTs with in vitro experiments further strengthens the presented conclusions. The authors also addressed most of the reviewers' comments from the first round of review in detail.

Thank you very much for your overall positive feedback and the precise summary of our work.

There are four main comments.

- First, in order to put the conclusions into context, some important information is missing. Although compliance as measured with the biomarker FA (as well as DHFA) is presented for immediately after the intervention, there is no information on participant compliance during the follow-up. Ideally, if these biomarkers were measured during follow-up, the results should be presented, otherwise at least the results from the diet records should be presented. For example, participants might have liked the oat meals or thought that they are good for their health and might have kept a higher intake during follow-up even if they were instructed not to do so. For the interpretation of the results as “persistent effects”, it needs to be clear that deviations from compliance did not happen.

Thank you very much for pointing this out. To be clear that deviations from compliance did not happen during the follow-up period, we recorded adherence to the study instructions in two different ways. First, we specifically asked the participants at each follow-up visit (V3-V5) whether they had consumed an oat meal or even a specific oat product within the last two weeks since their last clinical visit at the study center and if so, documented the frequency. Second, at baseline (V1) and endline (V5), the participants completed an additional 3-day dietary record.

Both, the precise query and the dietary records clearly show that the participants adhered to the study instructions during the follow-up period by refraining from eating oats and returning to their habitual Western diet.

We have included this information in the manuscript to supplement and corroborate our findings on possible persistent effects:

Methods (lines 640-645): “During the six-week follow-up period in the oat group of the short-term intervention (OG), compliance with the study instruction (i.e., refraining from oat consumption and returning to habitual Western diet) was assessed using two complementary methods: (1) specific query by the study staff about abstinence of oat meals as consumed during intervention and any other oat products, and (2) 3-day dietary records completed by the participants at baseline (V1) and at the end of the follow-up period (V5).”

Results (lines 136-139): “This assumption is further supported by the high compliance observed during the follow-up period, as all participants abstained from eating oats and returned to their habitual Western diet with no significant differences compared to their pre-study dietary pattern (Supplementary Table 2).”

Supplementary Table 2: Energy and nutrient intake of the participants in the oat group (OG) at baseline and at the end of the follow-up period assessed by 3-day dietary record¹

	Before study participation (V1)	End of the follow-up period (V5)	P -value
Total energy, kcal/d	2,207.23 (886.25)	2,163.58 (505.38)	0.975
Total protein, g/d	92.78 (28.17)	88.37 (43.07)	0.949
Total fat, g/d	89.03 ± 6.37	91.74 ± 8.41	0.759
Total CH, g/d	202.39 (94.57)	190.71 (85.59)	0.711
Fiber, g/d	22.38 ± 1.21	22.61 ± 1.67	0.877
Energy%			
Protein	16.50 (4.36)	16.40 (4.19)	0.895
Fat	36.30 ± 1.62	36.67 ± 1.64	0.821
CH	37.98 ± 1.79	38.70 ± 1.69	0.736

¹Data are presented as mean ± SEM or median (IQR) depending on data distribution. Absolute change (Δ) calculated as pre-intervention value subtracted from post-intervention value. T-values and P-values refer to paired Student's t-test. In case log-transformed data were used, the P-value is labelled with a superscript a. Abbreviations: CH, carbohydrates.

- Second, in some cases it looks like the randomisation may not have worked optimally as shown from the descriptive table e.g. the sex imbalance for the short-term intervention between the two groups. It is good that you have adjusted for sex in your analyses, but such an observation along with any other observed imbalances needs to be addressed in the discussion for possible implications on the interpretation of the results.

Thank you for bringing this up. Participants were randomly assigned to groups using computer-generated randomization tables in a block format with variable block length to provide statistical control over unknown confounding factors. In contrast to a stratified approach, the random allocation does not produce an equal distribution of each variable, but rather ensures that they are allocated to the groups at random. According to our internal validation, the variables in Table 1 were assigned to each group at random. In particular for the variable sex, a z-test yielded $p = 0.1622$, and a Fisher's exact test (odds ratio = 2.75, 95% CI [0.655, 11.538], $p = 0.2872$) similarly showed no statistically significant difference, suggesting the observed variation is likely due to chance. Nonetheless, we adjusted for sex, age, and BMI as prominent confounding factors in our regression analyses, and we now discuss the potential implications of baseline imbalances in the revised discussion section (lines 492-499).

“However, our findings remain constrained by the limited sample size typical in clinical studies, making it susceptible to baseline imbalances, despite the randomized group allocation. As a result, modest effects of the nutritional interventions may have gone undetected due to large inter-individual variability and potential confounding factors. To confirm our results based on an RCT, further studies are needed. For example, a multicenter RCT with a sample size calculation may provide the next level of evidence in clinical research since it would allow better generalizability across different populations and reduce a potential center-specific bias.”

- Third, as discussed in more detail below, sPLS-DA may overfit the data so any metrics reported e.g. AUC should ideally come from an internal validation. It is not clear so far whether this is what happens or the AUC is derived from the whole sample. Since this part of the analysis is related to prediction, relevant guidelines should be followed and ideally calibration should also be assessed (see below for more details).

Thank you for pointing this out. As recommended and in line with relevant guidelines, the AUC was derived from the internal cross-validation model. Further details are provided below.

- Finally, the discussion does not include any strengths and most importantly limitations. It would be very important to highlight any limitations of these studies (e.g. not optimal randomisation etc) for the results to be interpreted more correctly, but also to inform future studies.

Many thanks for your suggestion. We have now included the strengths and limitations of our study in the discussion (lines 484-499):

“Our study has several strengths. We conducted two well-designed parallel randomized controlled trials and confirmed our results in three different *in vitro* experiments. In addition, our study groups—including males and females—were characterized in depth and adherence to the respective diets was rigorously monitored using three complementary methods. By integrating multiple data sets—including various clinical markers related to MetS, metabolomic profiles in plasma and feces, as well as the gut microbiome—we provide a comprehensive understanding of the intervention effects and underlying mechanisms.

However, our findings remain constrained by the limited sample size typical in clinical studies, making it susceptible to baseline imbalances, despite the randomized group allocation. As a result, modest effects of the nutritional interventions may have gone undetected due to large inter-individual variability and potential confounding factors. To confirm our results based on an RCT,

further studies are needed. For example, a multicenter RCT with a sample size calculation may provide the next level of evidence in clinical research since it would allow better generalizability across different populations and reduce a potential center-specific bias.”

Please see some more detailed comments below.

- Abstract: It is missing some more quantitative measures. The previous reviewer 1’s comment has not been addressed. It is worth trying to reduce the number of words in other places in the abstract and include estimates with their 95% CIs rather than just the percentages.

Thank you for this recommendation. We have rewritten the abstract accordingly and provided the estimates with their 95% CIs as suggested (lines 1-17):

“Oats have various positive effects on human health, but the underlying mechanisms are not fully understood. To identify oat-microbiome-host interactions that contribute to metabolic improvements, we conducted two randomized controlled dietary interventions in individuals with metabolic syndrome, comparing a short-term, high-dose and a six-week, moderate oat intake with respective controls (DRKS00022169). While the moderate oat-diet leads to slight changes in metabolism, gut microbiota and metabolomic profiles, the high-dose oat-diet decreases total cholesterol by 8% (-15.61 [-24.17, -7.80] mg/dL, $P^{\text{adj.}} = 0.02$, Bonferroni-Holm correction) and low-density lipoprotein cholesterol by 10% (-16.26 [-23.60, -10.32] mg/dL, $P^{\text{adj.}} = 0.011$), accompanied by distinct changes in gut microbiota and metabolomic profiles, most notably an increase in microbially produced phenolic compounds. Here we show that phenolic metabolites are driving factors for the cholesterol-lowering effect of oats, which might be of relevance since a short-term, high-dose oat-diet is a suitable approach to alleviate obesity-related lipid disorders.”

- Introduction: The rationale for why certain metabolites were chosen could be clearer – looks like mainly phenols were chosen. FA and AVAs are mentioned in line 30, then it switches to ‘phenols’ in line 44, then back to FA and DHFA in line 52 then two other phenols are introduced in the brief summary of results (line 56/57). Perhaps ‘phenols such as...’ to start would set this up to be clearer throughout the text.

Thank you for this suggestion. We have rewritten the section as suggested (lines 30-33 and line 50-57):

“Oats offer an interesting and promising approach for treating MetS due to their unique composition characterized by a high fiber content, especially β -glucan, various bioactive substances including phenols such as ferulic acid (FA) and avenanthramides (AVAs), and essential minerals and vitamins⁸. [...] The aim of this randomized controlled trial (RCT) was to investigate the effects of a short-term, high-dose oat diet (a modified version of the original oat cure by Carl von Noorden⁹) and a six-week, moderate oat diet compared with a macronutrient-adapted control diet and a Western diet, respectively, on lipid metabolism, the gut microbiota, and global metabolomic profiles, in particular the phenolic compounds FA and dihydroferulic acid (DHFA), in 68 individuals with MetS (n = 17 participants/group) and to elucidate the underlying mechanism using an integrative multi-omics analysis.

In this work, our data demonstrate the superiority of the oat diets in increasing plasma levels of phenolic compounds, such as FA, DHFA, 2-aminophenol sulfate and 2-acetamidophenol sulfate compared with the respective controls.”

- Line 51: ‘metabolism’ this early on in the paper is a bit vague – perhaps ‘components of metabolic syndrome’ or ‘lipid metabolism’ would be clearer.

Thank you for pointing this out. We have rewritten this sentence to define our aim more clearly (lines 50-57):

“The aim of this randomized controlled trial (RCT) was to investigate the effects of a short-term, high-dose oat diet (a modified version of the original oat cure by Carl von Noorden⁹) and a six-week, moderate oat diet compared with a

macronutrient-adapted control diet and a Western diet, respectively, on lipid metabolism, the gut microbiota, and global metabolomic profiles, in particular the phenolic compounds FA and dihydroferulic acid (DHFA), in 68 individuals with MetS (n = 17 participants/group) and to elucidate the underlying mechanism using an integrative multi-omics analysis.”

- Results

- Lines 93 and 99: It is not clear what you mean by “showed a visceral fat distribution”. Do you mean high visceral fat? Abdominal or central obesity?

Thank you for bringing this up. We apologize that our description was not entirely clear. We have reworded the sentence as suggested and in line with the IDF definition, as described in the methods (lines 97+104):

“All participants had central obesity (100%) and at least two further metabolic syndrome traits including increased BP (100%), impaired glucose metabolism (75%) and dyslipidemia (63%). [...] According to the inclusion criteria, all participants had central obesity (100%) and at least two further metabolic syndrome traits. Thus, at baseline, 97% of the subjects had elevated BP, 76% impaired glucose metabolism, and 59% dyslipidemia.”

- Line 103: typo – Replace ‘tabel’ with table

Thank you for pointing this out. We have corrected the spelling mistake accordingly (line 108).

- Lines 89-103: As mentioned above, would it be worth highlighting some differences that may show that randomisation did not happen optimally in some cases? For example, in the short-term intervention the CG has 60% men, while the OG has 35% men. In addition, SBP and TG are higher in the short-term intervention compared with the long-term one.

These may have implications when comparing results between the CG and OG as well as the two interventions.

Thank you for raising this question. We mentioned the differences in the sex distribution at baseline of the short-term diet as suggested (lines 96f). As mentioned above, participants were randomly assigned to groups to provide statistical control over unknown confounding factors; however, random allocation does not produce an equal distribution of each variable. We agree that an imbalance in sex distribution between study groups may have implications when comparing the results of these study groups. Therefore, we adjusted our regression analyses for sex, age, and BMI as important confounding factors.

Furthermore, nutritional interventions might potentially have a stronger impact on higher or even pathological baseline level of clinical markers—such as SBP and TG—compared to physiological levels and this may have implications when comparing and interpreting these results. Additionally, we have discussed the potential implications of baseline imbalances in the revised discussion section (lines 492-499).

Results (lines 96f): “In the CG, 60% of participants were men, while in the OG 35% were men.”

Discussion (lines 492-499): “However, our findings remain constrained by the limited sample size typical in clinical studies, making it susceptible to baseline imbalances, despite the randomized group allocation. As a result, modest effects of the nutritional interventions may have gone undetected due to large inter-individual variability and potential confounding factors. To confirm our results based on an RCT, further studies are needed. For example, a multicenter RCT with a sample size calculation may provide the next level of evidence in clinical research since it would allow better generalizability across different populations and reduce a potential center-specific bias.”

- Lines 129-131: Is it correct that the oat biomarkers FA and DHFA were measured only at V1 and V2? Did you measure them at the rest of the visits? This would make it clearer whether the effects on lipids and BP at follow-up were indeed persistent or were because the participants changed their diet. If not then it would be good to include information from the dietary records to complement this result and state that based on the records participants did not change their diet, so indeed these can be considered persistent effects.

Thank you for raising this point again. As mentioned above, we recorded adherence to the study instructions during the follow-up period in two different ways: (1) we specifically asked the participants at each follow-up visit (V3-V5) whether they had consumed an oat meal or even a specific oat product within the last two weeks, and (2) participants completed an additional 3-day dietary record at baseline (V1) and endline (V5).

Both, the precise query and the dietary records clearly show that the participants adhered to the study instructions during the follow-up period by refraining from eating oats and returning to their habitual Western diet.

- Line 135: Supplementary Figure 1 does not show any major differences for glucose, so it is probably not worth highlighting this in your statements in lines 133-136.

Thank you for pointing this out. We agree that Supplementary Figure 1 does not show any major differences for glucose. Therefore, we have omitted the figure (as well as Supplementary Figure 2 for consistency) (lines 144 and 149).

- Line 217 (and 228): Could you add “but not statistically significantly” after “modulated the microbial functions differently” for accuracy?

Many thanks for this question. To increase accuracy, we have rewritten these sentences as suggested (lines 226-233 and 238-244):

“According to sPLS-DA, OG and CG modulated the microbial functions differently but not statistically significantly (AUC = 0.69 (P = 0.08), BER = 0.395; Fig. 6g); even though, several pathways seemed to be of relevance for differentiation, including carbohydrate digestion and absorption, aminobenzoate degradation, selenocompound metabolism, naphthalene degradation, and phosphotransferase system (PTS) (Fig. 6h).”

“Furthermore, a different modulation of the microbial functional capacity between the diet groups of the six-week intervention study was revealed, although not statistically significant (AUC = 0.68 (P = 0.07), BER = 0.353; Fig. 6i), with phenylalanine metabolism, proximal tubule bicarbonate reclamation, beta-lactam resistance, fructose and mannose metabolism, and PTS seemed to be the most relevant pathways for differentiation OG6w and CG6w (Fig. 6j).”

- **Methods:**

- Lines 513-514: Do you think there are any implications from dropping the number of total visits from 9 as initially planned to 4 or would the information you get still be sufficient?

Thank you for raising this important question. We agree that reducing the number of visits will result in a lower temporal resolution of data collection. Thus, we carefully ensured that the main research questions with all critical assessments and sample collections were maintained at strategically relevant time points within the revised schedule. Thus, the main endpoints of the study remain well covered. Additionally, we implemented rigorous monitoring procedures to ensure participant compliance and data completeness, although the practical requirements posed by the COVID-19 pandemic were challenging.

- Can you clarify why one study is isocaloric but the other is hypocaloric? And why did you plan to look for persistent changes after the intervention in the 48h intervention but not the 6 week one?

Thank you for raising this question regarding our study design. Based on scientific objectives and due to practical considerations associated with each intervention (e.g., easy to implement), we decided to implement an isocaloric diet in one study and a hypocaloric diet in the other. For the six-week intervention, an isocaloric protocol was chosen to isolate the effects of oat intake per se on metabolic and microbiome outcomes, independent of caloric restriction or changes in energy balance. This approach allowed us to attribute observed physiological and microbiota changes specifically to the dietary intervention rather than to weight loss or caloric deficit. Conversely, the 48h intervention was designed as a short-term diet (metabolic challenge) based on a modified version of the original “oat cure” developed by Carl von Noorden and, for practical reasons related to participant compliance and the acute nature of the intervention, was conducted under hypocaloric conditions. This allowed us to detect rapid metabolic and microbiota responses to a defined oat diet while controlling for dietary intake in a feasible time frame.

Due to scientific rationale, we investigated persistent changes after the 48h intervention study. Thus, it was important to assess whether the observed effects were transient or persisted after cessation of the oat challenge. Therefore, the follow-up period was conducted to evaluate the durability of the responses, which allowed us to draw conclusions about the optimal intervals at which a short-term, high-dose oat diet should be implemented on a regular basis. The six-week intervention was specifically designed to assess sustained changes induced by a prolonged and daily recurring dietary modification. Given the extended duration of the six-week intervention compared to the 48h intervention and the expected longer-lasting physiological and microbiota adaptations, our primary focus was on characterizing the outcomes at the end of the dietary period. We hope this clarifies the rationale underlying our study design choices.

- Line 542: The International Diabetes Federation in the 2009 paper that you cite defines hypertension as Systolic ≥ 130 and/or diastolic ≥ 85 mm Hg (or antihypertensive treatment with a history of hypertension). You have cited a different paper for your blood pressure cut-offs and used their definition of elevated blood pressure rather than hypertension (≥ 120 and/or diastolic ≥ 80 mm Hg). It is very helpful that you explained why you added the insulin resistance criterion, can you likewise clarify why you changed the blood pressure cut off from that suggested by the IDF?

Thank you for raising this question. We appreciate the opportunity to explain our rationale for using the different blood pressure (BP) cut-off. Several meta-analysis and systematic reviews have shown that blood pressure values in the range classified as “prehypertension” (120-139 mmHg systolic and/or 80-89 mmHg diastolic) are associated with a significantly increased risk of cardiovascular disease (Huang *et al. JAHA*. 2015. doi: 10.1161/JAHA.114.001519; Huang *et al. BMC Medicine*. 2013. doi: 10.1186/1741-7015-11-177; Guo *et al. Curr Hypertens Rep*. 2013. doi: 10.1007/s11906-013-0403-y; Gupta *et al. Hypertension Research*. 2010. doi: 10.1038/hr.2010.91). Thus, individuals in this “high-normal” BP range are at increased risk but would not be identified using the standard IDF definition for hypertension. Choosing a lower threshold additionally allows for earlier identification of individuals at increased risk, enabling preventive dietary interventions to be initiated at a stage when they may be more effective.

For clarification, we have included an explanation for changing the blood pressure threshold from the value suggested by the IDF in the revised manuscript (lines 574-578):

“The specific threshold values for BP were chosen because prehypertension (≥ 120 mmHg systolic and/or ≥ 80 mmHg diastolic) has been associated with a significantly increased risk of cardiovascular disease⁶⁵⁻⁶⁸, allowing preventive dietary interventions to be initiated at a stage where they may be more effective.”

- Line 557: In the rebuttal to reviewer 3, it states that ex-smokers were not excluded but in line 557, it states that one of the exclusion criteria is 'history of smoking'. This might need re-wording to make clear whether those with a past history of smoking were excluded or not.

Thank you for pointing this out. We have reworded this exclusion criteria accordingly to make clear that only participants who are non-smokers were included (line 588). However, ex-smokers included in this study reported an average duration of 26 years since quitting smoking, while participants with a recent history of smoking were excluded.

“Exclusion criteria were as follows: (1) [...]; (6) smoking or alcoholism”

- Lines 731-744: From the description, it looks like the LMM and the LnR were used for the same purpose, but they contradict each other in terms of accounting for random effects. Could you clarify whether indeed the two methods were used for the same purpose and if that is the case which one would be superior and what is the advantage of using both?

Many thanks for raising this question. In the respective section, we consider alpha and beta diversity. While alpha diversity is a sample specific measure for microbial diversity, beta diversity provides a measure for the difference in diversity. We compute alpha diversity for all samples separately, while beta diversity is always computed with respect to the baseline microbial profile for an individual. Accordingly, the measures are fundamentally different and required different statistical analysis approaches.

We analyzed alpha diversity using a Linear mixed effect model (LMM) with an individual specific random effect as the baseline diversity between individual may differ. For beta diversity – which considers the change with respect to baseline – we used a linear regression as differences in baseline are accounted for by the metric itself. We have rewritten the corresponding section in the revised manuscript (lines 767-791):

“We quantified alpha diversity using Shannon entropy, Pielou’s evenness, and Faith’s phylogenetic diversity (Faith-PD). Alpha diversity was assessed per sample, and a linear mixed model (LMM) was applied to evaluate the differences between the two diet groups in each study. The LMM was adjusted for the confounders sex, age and BMI (LMM^{adj.}), and estimated the diet effect relative to the baseline value and the control group (CG or CG^{6w}). An individual-specific random intercept was included to account for differences in baseline alpha diversity. The model was defined as: alpha diversity ~ time*group + age + sex + BMI + (1|Person-ID), with the covariate of interest ‘time*group’.

We quantified beta diversity using Jaccard distance, Bray-Curtis dissimilarity, and UniFrac distance matrices. As beta diversity measures differences in community composition, we computed it with respect to each individual’s baseline microbiome. For the analysis of beta diversity –which considers the change with respect to baseline – we used a linear regression model (LnR), as baseline differences are accounted for by the metric itself. Beta diversity was modeled as a function of the diet groups, using the control group of each study (CG or CG^{6w}) as the reference (LnR^{adj.}), and was adjusted for the confounders sex, age and BMI. The model was defined as: beta diversity matrix ~ group + age + sex + BMI, with the covariate of interest ‘group’.”

- sPLS-DA (775-788 and supplementary methods): You mention that “Missing values in the global metabolomics profiles were imputed with the minimum observed value for each compound.”. How confident are you that all the values were missing for example because they were below the low limit of quantification, which would indeed justify imputation with the minimum observed value? Could there be other reason of missing for some of the values that would justify other types of imputation (depending on the % missing this could be multiple imputation or imputation to the mean/median). In addition, could you include the %missing?

Thank you for these questions. Indeed, we are quite confident that the analysis of the global metabolomic profiles generated by Metabolon are the gold-standard. According the white paper of Metabolon, the majority (roughly 90%) of the missing values are below the limit of detection, which would justify imputation with the minimum observed value. Additionally, Metabolon has run multiple imputation studies and has extensively tested other methods and has found no advantages to other methods over this. While imputation by mean or median is pragmatic and interpretable, it may lead to issues in the downstream sPLS-DA analysis. Mean/Median imputation may artificially elevate low-abundance features and reduce their variance, which would potentially mask biological differences and introduce artificial correlations between features. In contrast, imputation by minimum preserves the relative structure (rank order) and avoids spurious results in the downstream sPLS-DA analysis. Therefore, based on the assumption that the missing values in the global metabolomics profiles mainly arise from the detection limit, a conservative imputation approach using the minimum observed value for each compound was employed.

However, other reasons for missing values could be due to integration issues during curation, for example software had a hard time defining the peak or a smaller peak being absorbed into a larger one rather than being resolved.

In our global overall metabolomic profiles, we had 12% and 24% missing values in the plasma and fecal metabolomic profiles of the short-term intervention study as well as 13% and 26 % in the plasma and fecal metabolomic profiles of the six-week intervention study. This is in line with the literature, reporting that missing values are quite common in large untargeted metabolomics datasets and might comprise up to 50% of the dataset (Krutkin *et al. J Am Soc Mass Spectrom.* 2025. doi: 10.1021/jasms.4c00434). We included the %missing values in the supplementary methods (page 8).

- Finally, sPLS-DA is known for overfitting the data. You mention that you used 5-fold cross-validation to choose the number of features and components, which is a good start to partly account for the overfitting. Could you clarify whether the AUC was also derived from the cross-validation process or the whole dataset? To avoid overfitting, it may make sense to report the AUC from the cross-validation model that you selected based on BER.

Thank you for pointing out that it was not clear how the analysis was conducted. Indeed, the AUC and BER for the sPLA-DA models were derived from the cross-validation as recommended. Thus, the performance metrics for all sPLS and DIABLO models were derived from the cross-validation process. We have rewritten the description in the manuscript being more specific and detailed:

Methods (lines 832-834): “The performance of the sPLS-DA was assessed on the BER and AUC, which were calculated using a cross-validation approach.”

Methods (lines 850f): “The performance of the final (DIABLO) models was assessed on the BER and AUC derived from internal cross-validation of the model.”

Supplementary Methods (page 8): “The models were calibrated and assessed using 5-fold cross-validation. For each fold, absolute shrinkage and selection operator (LASSO) was used on the training data to minimize the number of features per component while maximizing class discrimination. BER and AUC were calculated for the final model on the test data. As result, we present the averages across the folds.”

Finally, since this is a prediction model, to comply with the guidelines of reporting prediction models (see for example TRIPOD <https://pubmed.ncbi.nlm.nih.gov/25560730/>) apart from presenting a measure of discrimination like the AUC, it would be more complete to also evaluate calibration and the recommended approach is calibration plots (again derived through the cross-validation process to avoid overfitting).

Thank you for this suggestion. As recommended based on the guidelines, we generated out-of-fold (OOF) predictions for each DIABLO and sPLS-DA model using 5-fold cross-validations with 25 repeats. For each participant, we took the Oats–Control score margin and converted it to a probability with Platt scaling (logistic regression fit only to the OOF data). Probabilities were grouped into quantile bins; we plotted the bin mean vs. the observed Oats rate with 95% Wilson intervals and overlaid a loess smoother. The dashed 45° line marks perfect calibration. AUC is derived from cross-validation.

The majority of the calibration plots did not indicate a bias and showed that the predicted probabilities match the observed outcomes well. The DIABLO models (see Figure A, C) and the global plasma metabolome sPLS-DA model (Figure G), from which the main outcome of our study were derived, performed particularly very well. Being aware of the underperformance of the sPLS-DA models with respect to the KEGG pathways in both studies (Figure M, N) and the clinical markers in the six-week study (Figure F), we critically examined these results and viewed them with caution to avoid overstatement.

Caption: Calibration plots for all DIABLO and sPLS-DA models classifying Oats versus Control, prepared following the TRIPOD guidance (according to Moons et al. Ann Intern Med. (2015), Figure 8). The x-axis shows each model's predicted probability of Oats (from out-of-fold predictions, Platt-scaled to [0,1]); the y-axis shows the observed proportion of Oats within probability bins. Points are bin means with 95% Wilson CIs; numbers above points indicate the number of participants in each bin. The dashed line denotes perfect calibration ($y = x$); the solid line is a LOESS smoother. AUC is reported above each panel.

- Lines 802-804: Similarly to the previous comment, does the DIABLO workflow give an option to do any sort of internal validation to avoid overfitting of the AUCs especially since it is based on sPLS-DA as well?

Thank you for raising this question. Similar to the sPLS-DA, we have derived the AUC and BER for the DIABLO model from internal cross-validation (to check the performance of the model). As mentioned above, we have rewritten the corresponding section in the manuscript:

Methods (lines 850f): “The performance of the final models was assessed on the BER and AUC derived from internal cross-validation of the model.”

- Discussion:
 - It would be helpful to have a limitations section of the discussion to address the potential impact of aspects of the study such as unequal sex distribution, waist circumference and baseline lipids between the arms of the studies etc.

Many thanks for this suggestion. We have included a limitations section in the discussion (lines 492-499):

“However, our findings remain constrained by the limited sample size typical in clinical studies, making it susceptible to baseline imbalances, despite the randomized group allocation. As a result, modest effects of the nutritional interventions may have gone undetected due to large inter-individual variability and potential confounding factors. To confirm our results based on an RCT, further studies are needed. For example, a multicenter RCT with a sample size calculation may provide the next level of evidence in clinical research since it would allow better generalizability across different populations and reduce a potential center-specific bias.”

Reviewer #5 (Remarks to the Author):

Thank you very much for co-reviewing our manuscript. We appreciate your efforts.

Dear Referees,

Thank you very much for all your valuable feedback. We have addressed all remaining comments and suggestions in a point-by-point response (in blue). Thus, we hope that the manuscript is suitable for publication in the special issue *Microbiome & Nutrition in Nature Communications*.

Reviewer #3 (Remarks to the Author):

The authors have sufficiently addressed my concerns. One minor suggestion, the authors should avoid the use of "through alterations of gut microbiota". Given that no FMT experiments have been performed, it is strongly suggested to use "associated/related" instead.

Thank you very much for confirming that our revisions have sufficiently addressed your previous comments, and for this valuable suggestion. We have adjusted the wording throughout the manuscript as recommended.

Reviewer #4 (Remarks to the Author):

I would like to thank the authors for their responses. I have attached a document with a few additional comments, as we found it helpful to include tracked changes.

Thank you very much for the careful re-evaluation of our manuscript and for the additional valuable comments. We have thoroughly addressed all new suggestions and implemented the recommended text changes accordingly. Details are provided in our responses below and in the revised manuscript.

Additional comments transferred from the document

We would like to thank the authors for their responses and revisions. Most comments have been addressed, but there are a few additional comments below. We have included suggestions for text changes in the manuscript as tracked changes.

Thank you very much for your overall positive feedback and the additional comments, including the helpful suggestions for text changes.

Introduction

- The rationale for why certain metabolites, particularly phenols, were chosen is still not clear, especially for readers who are not very familiar with the biological functions/importance of phenols. Despite being in the title, comments about phenols are lost within lists of other components (line 30, line 44). It would be helpful to have one sentence introducing or highlighting phenols and their biological role earlier in the introduction and one sentence on why these specific ones were chosen. Are they all the same type or have they been suggested in the literature before?

Suggestions:

- Perhaps highlight phenols and what they do earlier (lines 28-31):
Oats offer an interesting and promising approach for treating MetS due to their unique composition characterized by a high fiber content, especially β -glucan, various bioactive substances including phenols (which... [insert something here to highlight what the biological role/importance of phenols is] and essential minerals and vitamins.
- Then bring forward the ones you look at here somewhere (lines 39-48):
However, the underlying mechanisms of oat-induced improvements in metabolic health are not fully understood. Recent studies suggest the ability of oats to modulate the gut microbiota as a mechanism for its health-promoting effects¹³. In particular, microbially produced metabolites were ascribed a decisive role in nutrition-induced metabolic improvement¹⁴. However, it is still unknown whether metabolites produced by the microbial degradation of oats, such as phenolic compounds, play an important health-promoting role alongside short-chain fatty acids (SCFAs)¹⁵ produced by the bacterial fermentation of β -glucan. We hypothesized that the beneficial effects of oats on metabolism might be influenced by host-microbiome interactions, leading to changes in phenolic compounds such as FA, DHFA, 2-aminophenol sulfate and 2-acetamidophenol sulfate which have been indicated in...[insert here something on why these ones have been picked].

Thank you for pointing this out. We agree that the rationale for selecting phenolic metabolites should be explained more clearly by including a sentence introducing phenols and their biological role earlier in the introduction and by adding an additional sentence on the specifically selected metabolites. We have included the additional information as suggested.

Lines 28-32: “Oats offer an interesting and promising approach for treating MetS due to their unique composition characterized by a high fiber content, especially β -glucan, essential minerals and vitamins, and various bioactive substances including phenols, which exert antioxidant and anti-inflammatory effects that may improve metabolic function⁸.”

Lines 44-52: “However, the underlying mechanisms of oat-induced improvements in metabolic health are not fully understood. Recent studies suggest the ability of oats to modulate the gut microbiota as a mechanism for its health-promoting effects¹³. In particular, microbially produced metabolites were ascribed a decisive role in nutrition-induced metabolic improvement¹⁴. However, it is still unknown whether metabolites produced by the microbial degradation of oats, such as phenolic compounds, play an important health-promoting role alongside short-chain fatty acids (SCFAs)¹⁵ produced by the bacterial fermentation of β -glucan. We hypothesized that the beneficial effects of oats on metabolism might be influenced by host-microbiome interactions, leading to changes in phenolic compounds such as ferulic acid (FA), dihydroferulic acid (DHFA), 2-aminophenol sulfate and 2-acetamidophenol sulfate which have been indicated in recent studies as relevant (microbial) metabolites from whole grains such as oats with different biological functions^{16,17}.”

Methods

- Thank you for clarifying compliance with the study instruction during follow-up. A brief comment in the limitations section that this is self-reported and not an objective biomarker measure would be important, especially since you used biomarkers to assess compliance during the interventions.

Thank you for bringing this up. As suggested, we included a brief comment in the limitations section (lines 482-486):

“In addition, adherence to the study instruction during the follow-up period was self-reported, which is common practice in nutritional studies, though potential overestimation should be considered compared to the objective biomarker measures which were conducted before and after the short-term and six-week intervention.”

- Thank you for clarifying the rationale underlying the study design choices. The following point from your response would be a good addition to the methods section: *Therefore, the follow-up period was conducted to evaluate the durability of the responses, which allowed us to draw conclusions about the optimal intervals at which a short-term, high-dose oat diet should be implemented on a regular basis.*

Thank you for pointing this out. We have included this aspect in the methods section as suggested (lines 530-532):

“In the short-term intervention study, eligible participants were invited for two clinical visits, the first before (V1) and the second directly after the two-day intervention period (V2), followed by three clinic visits at two-week intervals during a six-week follow-up period (V3 – V5) to determine potential long-term effects of the oat diet (Fig. 1a). Therefore, the follow-up period was conducted to evaluate the durability of the responses, allowing conclusions to be drawn about the optimal intervals at which a short-term, high-dose oat diet might be implemented on a regular basis.”

- To be clear where your definitions deviate from the IDF we suggest: Lines 565-576
...coupled with two of the following five criteria: (1) elevated BP (≥ 120 mmHg systolic and/or ≥ 80 mmHg diastolic), (adapted to include pre-hypertension)⁶³; [...] (adapted to the WHO definition⁶⁴). The specific threshold values for BP were chosen because prehypertension (≥ 120 mmHg systolic and/or ≥ 80 mmHg

diastolic) has been associated with a significantly increased risk of cardiovascular disease^{65–68} and therefore represents a stage in which preventive dietary interventions may be more effective. The fifth criterion was added after the initial study registration in order to attach more importance to impaired glucose metabolism as a characteristic of MetS and to counteract recruitment difficulties due to the COVID-19 pandemic.

Thank you for this suggestion. To clarify where our definitions deviate from the IDF, we have revised the description of the inclusion criteria as recommended (lines 563-575).

“... coupled with two of the following five criteria: (1) elevated BP (≥ 120 mmHg systolic and/or ≥ 80 mmHg diastolic) (adapted to include pre-hypertension⁶³); (2) elevated fasting serum triglycerides (≥ 150 mg/dl); (3) decreased fasting serum HDL-C (< 40 mg/dl for men and < 50 mg/dl for women); (4) elevated fasting plasma glucose (≥ 100 mg/dl); (5) indication of insulin resistance (HOMA-IR index > 2.5) (adapted to the WHO definition⁶⁴). The specific threshold values for BP were chosen because prehypertension (≥ 120 mmHg systolic and/or ≥ 80 mmHg diastolic) has been associated with a significantly increased risk of cardiovascular disease^{65–68} and therefore represents a stage in which preventive dietary interventions may be more effective. The fifth criterion was added after the initial study registration in order to attach more importance to impaired glucose metabolism as a characteristic of MetS and to counteract recruitment difficulties due to the COVID-19 pandemic.”

- Line 583: “current or recent history of smoking or alcoholism (defined as...)”

Thank you for this suggestion. We defined smoking during the past 3 years prior to study enrollment as exclusion criteria since literature indicates that cardiometabolic effects of smoking decline after quitting within 2-3 years (Dobson et al. 1991, doi: 10.1016/0895-4356(91)90157-5; American Cancer Society, <https://www.cancer.org/cancer/risk-prevention/tobacco/guide-quitting-smoking/benefits-of-quitting-smoking-over-time.html>). Ex-smokers included in this study reported an average duration of 26 years since quitting smoking.

The exclusion criterion “current or recent history of alcoholism” was defined as clinical diagnosis of alcohol use disorder according to ICD-10/DSM-5 within the last three years, or self-reported problematic alcohol consumption requiring treatment with less than 12 months abstinence before study enrollment. We have included these definitions in the revised manuscript (lines 582-586):

“(6) current or recent history of smoking (defined as during the past 3 years prior to study enrollment^{70,71}) or alcoholism (defined according to ICD-10/DSM-5 within the last three years, or self-reported problematic alcohol consumption requiring treatment with less than 12 months abstinence before study enrollment)”

Results

- Thank you for clarifying how randomisation was conducted. In terms of the implications of differences between the arms:
 - Sex: According to the SAGER guidelines results should be ‘routinely presented disaggregated by sex and gender’ and the ‘potential implications of sex and gender on the study results should be discussed. If not conducted the rationale should be given’. The sample size here is too small to be stratified by sex so a comment in the methods section to this effect would fulfil this point.

Thank you for pointing out the importance of sex-disaggregated reporting according to the SAGER guidelines. Due to the limited sample size, stratification by sex was not feasible in our study. To address this, we have included a corresponding statement in the Methods section as recommended (lines 937-939).

“In line with the SAGER guidelines, sex-disaggregated analyses should be pursued; however, due to the limited sample size, stratification by sex was not conducted.”

- Are there any expected differences by sex? Re: the implications of an unbalanced distribution of sex in the arms of the study – you can't separate your results to see if men and women respond differently so you can't know if it matters that the groups are unbalanced. This is a limitation that could be included in the discussion.

Thank you for this valuable comment. We agree that the unbalanced distribution of sex between study groups is considered a limitation. We have therefore addressed this issue in the discussion section (lines 486-491).

“Finally, the unbalanced sex distribution between the study groups may have influenced the results. To confirm our results based on an RCT and to evaluate potential sex-specific differences in the response to the intervention, further studies are needed. For example, a multicenter RCT with a sample size calculation may provide the next level of evidence in clinical research as it allows for sex-specific analysis and better generalizability across different populations and reduces a potential center-specific bias.”

- TG and SBP – Thank you for your response to this – it is very clear and would be a good addition to your limitations section, specifically words to the effect of what you have here:

Furthermore, nutritional interventions might potentially have a stronger impact on higher or even pathological baseline level of clinical markers- such as SBP and TG-compared to physiological levels and this may have implications when comparing and interpreting these results.

Thank you very much for this constructive suggestion. We have incorporated this aspect into the limitations section as recommended (lines 479-482).

“Furthermore, nutritional interventions in general might potentially have a stronger impact on higher or even pathological baseline level of clinical markers compared to physiological levels, which may have implications when comparing and interpreting the results of these studies.

- Thank you also for presenting the calibration plots. we don't think you have added them to the manuscript; could you add them even as supplementary figures and a short sentence in the methods and results about calibration?

Thank you for this suggestion. We have included the calibration plots as Supplementary Figure 4 and added a corresponding description of the calibration in the revised manuscript (lines 879-889).

“Calibration analysis

Out-of-fold (OOF) predictions for each DIABLO and sPLSDA model were generated using 5-fold cross-validations with 25 repeats. For each participant, the Oats–Control score margin were taken and converted it to a probability with Platt scaling (logistic regression fit only to the OOF data). Probabilities were grouped into quantile bins; the bin mean vs. the observed oats rate with 95% Wilson intervals and overlaid a loess smoother were visualized. AUC is derived from cross-validation. Since the dashed 45° line marks perfect calibration, the majority of the calibration plots did not indicate a bias and showed that the predicted probabilities match the observed outcomes well, in particular the DIABLO models and the sPLS-DA model of the global plasma metabolomic profile, from which the main outcome of our study were derived, performed very well (Supplementary Fig. 4).”

Discussion

- Thank you for including a section on strengths and limitations– see above for suggestions of more specific examples of limitations which would improve this section further.

Thank you for this valuable comment. We agree that including more specific examples of limitations improves this section further. Following your suggestions, we have revised the section of strengths and limitations to provide more detailed information on key limitations, including the unbalanced sex distribution between the study groups, the potential implications of baseline imbalances despite randomization, and the recording of adherence to the study instruction during the follow-up period based on self-reported information (lines 475-491).

“However, our findings remain constrained by the limited sample size typical in clinical studies, making it susceptible to baseline imbalances, despite the randomized group allocation. As a result, modest effects of the nutritional interventions may have gone undetected due to large inter-individual variability and potential confounding factors. Furthermore, nutritional interventions in general might potentially have a stronger impact on higher or even pathological baseline level of clinical markers compared to physiological levels, which may have implications when comparing and interpreting the results of these studies. In addition, adherence to the study instruction during the follow-up period was self-reported, which is common practice in nutritional studies, though potential overestimation should be considered compared to the objective biomarker measures which were conducted before and after the short-term and six-week intervention. Finally, the unbalanced sex distribution between the study groups may have influenced the results. To confirm our results based on an RCT and to evaluate potential sex-specific differences in the response to the intervention, further studies are needed. For example, a multicenter RCT with a sample size calculation may provide the next level of evidence in clinical research as it allows for sex-specific analysis and better generalizability across different populations and reduces a potential center-specific bias.”

Supplementary methods

- Your response to the question of missing values in the metabolomic profiles is very helpful – perhaps include this sentence (or words to this effect) from your response in the supplementary methods (page 8) –

This is in line with the literature, reporting that missing values are quite common in large untargeted metabolomics datasets and might comprise up to 50% of the dataset (Krutkin et al. J Am Soc Mass Spectrom. 2025. doi: 10.1021/jasms.4c00434).

Thank you for this helpful suggestion. We have now incorporated the indicated information into the Supplementary Methods (page 14).

“This is in line with the literature, reporting that missing values are quite common in large untargeted metabolomics datasets and might comprise up to 50% of the dataset²³.”

Reviewer #5 (Remarks to the Author):

Thank you very much for co-reviewing our manuscript. We appreciate your efforts.

We would like to thank the authors for their responses and revisions. Most comments have been addressed, but there are a few additional comments below. We have included suggestions for text changes in the manuscript as tracked changes.

Introduction:

The rationale for why certain metabolites, particularly phenols, were chosen is still not clear, especially for readers who are not very familiar with the biological functions/importance of phenols. Despite being in the title, comments about phenols are lost within lists of other components (line 30, line 44). It would be helpful to have one sentence introducing or highlighting phenols and their biological role earlier in the introduction and one sentence on why these specific ones were chosen. Are they all the same type or have they been suggested in the literature before?

Suggestions:

Perhaps highlight phenols and what they do earlier (lines 28-31)

Oats offer an interesting and promising approach for treating MetS due to their unique composition characterized by a high fiber content, especially β -glucan, various bioactive substances including phenols (which... [insert something here to highlight what the biological role/importance of phenols is] and essential minerals and vitamins.

Then bring forward the ones you look at here somewhere (lines 39-48):

However, the underlying mechanisms of oat-induced improvements in metabolic health are not fully understood. Recent studies suggest the ability of oats to modulate the gut microbiota as a mechanism for its health-promoting effects¹³. In particular, microbially produced metabolites were ascribed a decisive role in nutrition-induced metabolic improvement¹⁴. However, it is still unknown whether metabolites produced by the microbial degradation of oats, such as phenolic compounds, play an important health-promoting role alongside short-chain fatty acids (SCFAs)¹⁵ produced by the bacterial fermentation of β -glucan. We hypothesized that the beneficial effects of oats on metabolism might be influenced by host-microbiome interactions, leading to changes in phenolic compounds such as FA, DHFA, 2-aminophenol sulfate and 2-58 acetamidophenol sulfate which have been indicated in... [insert here something on why these ones have been picked] .

Methods:

Thank you for clarifying compliance with the study instruction during follow-up. A brief comment in the limitations section that this is self-reported and not an objective biomarker measure would be important, especially since you used biomarkers to assess compliance during the interventions.

Thank you for clarifying the rationale underlying the study design choices. The following point from your response would be a good addition to the methods section: *Therefore, the follow-up period was conducted to evaluate the durability of the responses, which allowed us to draw conclusions about the optimal intervals at which a short-term, high-dose oat diet should be implemented on a regular basis.*

To be clear where your definitions deviate from the IDF we suggest:

Lines 565-576

...coupled with two of the following five criteria: (1) elevated BP (≥ 120 mmHg systolic and/or ≥ 80 mmHg diastolic), (adapted to include pre-hypertension)⁶³; [...] (adapted to the WHO definition⁶⁴). The specific threshold values for BP were chosen because prehypertension (≥ 120 mmHg systolic and/or ≥ 80 mmHg diastolic) has been associated with a significantly increased risk of cardiovascular disease^{65–68} and therefore represents a stage in which preventive dietary interventions may be more effective. The fifth criterion was added after the initial study registration in order to attach more importance to impaired glucose metabolism as a characteristic of MetS and to counteract recruitment difficulties due to the COVID-19 pandemic.

Line 583: “current or recent history of smoking or alcoholism (defined as...)”

Results:

Thank you for clarifying how randomisation was conducted. In terms of the implications of differences between the arms:

- Sex: According to the SAGER guidelines results should be ‘routinely presented disaggregated by sex and gender’ and the ‘potential implications of sex and gender on the study results should be discussed....If not conducted the rationale should be given’. The sample size here is too small to be stratified by sex so a comment in the methods section to this effect would fulfil this point.
- Are there any expected differences by sex? Re: the implications of an unbalanced distribution of sex in the arms of the study – you can’t separate your results to see if

men and women respond differently so you can't know if it matters that the groups are unbalanced. This is a limitation that could be included in the discussion.

- TG and SBP – Thank you for your response to this – it is very clear and would be a good addition to your limitations section, specifically words to the effect of what you have here:

Furthermore, nutritional interventions might potentially have a stronger impact on higher or even pathological baseline level of clinical markers—such as SBP and TG—compared to physiological levels and this may have implications when comparing and interpreting these results.

Thank you also for presenting the calibration plots. we don't think you have added them to the manuscript; could you add them even as supplementary figures and a short sentence in the methods and results about calibration?

Discussion:

Thank you for including a section on strengths and limitations— see above for suggestions of more specific examples of limitations which would improve this section further.

Supplementary methods:

Your response to the question of missing values in the metabolomic profiles is very helpful – perhaps include this sentence (or words to this effect) from your response in the supplementary methods (page 8) –

This is in line with the literature, reporting that missing values are quite common in large untargeted metabolomics datasets and might comprise up to 50% of the dataset (Krutkin et al. J Am Soc Mass Spectrom. 2025. doi: 10.1021/jasms.4c00434).